# Assessing the factors governing the ability to predict late-spring flooding in cold regions mountain basins

Vincent Vionnet[1,2], Vincent Fortin[2], Etienne Gaborit[2], Guy Roy[2], Maria Abrahamowicz[2], Nicolas Gasset[3], John W. Pomeroy[1]

[1] Centre for Hydrology, University of Saskatchewan, Saskatoon, SK, Canada
[2] Environmental Numerical Research Prediction, Environment and Climate Change Canada, Dorval, QC, Canada
[3] Environmental Numerical Prediction Development, Meteorological Service of Canada, Environment and Climate Change Canada, Dorval, QC, Canada

*Correspondence to*: Vincent Vionnet (vincent.vionnet@canada.ca)

**Abstract.** From June 19 to June 22, 2013, intense rainfall and concurrent snowmelt led to devastating floods in the Canadian Rockies, foothills and downstream areas of southern Alberta and southeastern British Columbia, Canada. Such an event is typical of late-spring floods in cold regions mountain headwater combining intense precipitation with rapid melting of late-lying snowpack and represents a challenge for hydrological forecasting systems. This study investigated the factors governing the ability to predict such an event. Three sources of uncertainty, other than the hydrological model processes and parameters, were considered: (i) the resolution of the atmospheric forcings, (ii) the snow and soil moisture Initial Conditions (ICs) and (iii) the representation of the soil texture. The Global Environmental Multi-scale hydrological modelling platform (GEM-Hydro), running at a 1-km grid spacing, was used to simulate hydrometeorological conditions in the main headwater basins of southern Alberta during this event. The GEM atmospheric model and the Canadian Precipitation Analysis (CaPA) system were combined to generate atmospheric forcing at 10, 2.5 and 1 km over Southern Alberta. Gridded estimates of snow water equivalent from the Snow Data Assimilation System (SNODAS) were used to replace the model SWE at peak snow accumulation and generate alternative snow and soil moisture ICs before the event. Two global soil texture datasets were also used. Overall twelve simulations of the flooding event were carried out. Results show that the resolution of the atmospheric forcing affected primarily the flood volume and peak flow in all river basins due to more accurate estimation of intensity and total amount of precipitation during the flooding event provided by CaPA analysis at convection-permitting scales (2.5 and 1 km). Basin-averaged snowmelt also changed with the resolution due to changes in near-surface wind and resulting turbulent fluxes contributing to snow melt. Snow ICs were the main sources of uncertainty for half of the headwater basins. Finally, the soil texture had less impact and only affected peak flow magnitude and timing for some stations. These results highlight the need to combine atmospheric forcing at convection-permitting scale with high quality snow ICs to provide accurate streamflow predictions during late-spring flood in cold region mountain river basins. The predictive improvement by inclusion of high elevation weather stations in the precipitation analysis and the need for accurate

mountain snow information suggests the necessity of integrated observation and prediction systems for forecasting extreme events in mountain river basins.

## 1 Introduction

From June 19 to 22, 2013, heavy rainfall and snowfall (at high elevations) with local amounts exceeding 300 mm fell over a broad region of the Canadian Rocky Mountains of southeastern British Columbia and southwestern Alberta, and the foothills and adjacent plains of southern Alberta, Canada (Milrad et al., 2015; Liu et al., 2016; Kochtubajda et al., 2016; Pomeroy et al., 2016a,b). This heavy precipitation resulted from moisture convergence from the Pacific and from the Canadian Prairies and US Great Plains through evapotranspiration (Milrad et al., 2015; Li et al., 2017). At high elevations, rain and then snow fell on a deeper than normal, late-lying snowpack leading to rain-on-snowmelt which enhanced runoff generation (Fang and Pomeroy 2016; Pomeroy et al., 2016a). This heavy rainfall combined with a rapidly melting alpine snowpack triggered severe flooding in the Oldman, Bow and Red Deer river basins of southern Alberta and the Elk River of southern British Columbia and impacted many communities, including Calgary, the largest city in Alberta (Pomeroy et al., 2016b). The floods led to the evacuation of 100,000 people and caused five fatalities and over $6 billion of damage to infrastructure such as roads, railways, bridges, parks and homes, making it the most expensive natural disaster in Canadian history at the time (Pomeroy et al., 2016b).

This extreme weather event presents similarities with other flooding events that occurred in regions of complex terrain. In terms of atmospheric conditions, it shares similarities with events that previously occurred in the mountains of southern Alberta (Shook, 2016), in the mountainous basins of central Europe in June 2013 (Grams et al. 2014), in the Colorado range in the United States in September 2013 (Gochis et al., 2015) or in mountainous Mediterranean areas (Ricard et al., 2012). These events were characterized by long-lasting precipitation strongly influenced by the local topography (e.g. Friedrich et al., 2016; Milrad et al, 2017) where mesoscale convective systems stalled over the foothills when moving upslope towards the mountains. In late-spring and early-summer, these heavy rainfall events can occur with a high-freezing altitude leading to a rain-on-snow (ROS) event in high-elevation areas that are still covered by snow (e.g. McCabe et al., 2007; Corripio et al., 2017). Therefore, late-spring and early-summer flooding events in mountainous terrain are unique since they potentially combine severe impacts resulting from intense precipitation from active convective systems in the foothills and enhanced runoff generation from a combination of snowmelt and rainfall-runoff in the higher elevations of the front and central ranges.

The complex spatial and temporal variability of runoff generation during late-spring and early-summer floods in mountainous terrain makes them particularly difficult to simulate for hydrological models (e.g. Corripio et al., 2017). The skill of simulated streamflow is influenced by the ability to estimate initial conditions (soil moisture, snowpack …), the quality of the meteorological forcing driving the model and the capability of the model to simulate hydrological processes

and streamflow (e.g. Hapuarachchi et al., 2011; Pagano et al., 2014). Zappa et al. (2011) studied how these three sources of uncertainty influenced flood forecasting in complex topography areas and identified the precipitation forcing (either from radar estimate or from weather forecasts) as the main source of uncertainty for peak flow events. Hydrological simulations are particularly sensitive to the spatial and temporal distribution of the precipitation input in regions of complex orography (e.g. Jasper et al., 2002; Lobligeois et al., 2014). These regions are often characterized by sparse gauge networks and pose difficulties to radar measurements (beam blockage, ground clutter; Germann et al. 2006) so that atmospheric models are often considered as an alternative solution to obtain precipitation estimate. In particular, high-resolution, convection-permitting atmospheric models have the ability to provide realistic estimation of topographically-induced vertical motions and resulting precipitation (e.g. Rasmussen et al., 2011; Pontoppidan et al., 2017). A recent review by Lundquist et al. (2019) concluded that high-resolution atmospheric models are now better able to simulate mountain seasonal precipitation than gridded precipitation products derived from in situ observation networks or/and radar estimates in complex terrain although reproducing specific convective storms remains a challenge. For the June 2013 flood in Alberta, Li et al. (2017) and Milrad et al. (2017) have shown that the Weather Research and Forecast (WRF) atmospheric model at convection-permitting resolution (3-km) demonstrated reasonable skill in simulating the spatial and temporal evolution of precipitation patterns before and during the flood.

Hydrological simulations of flooding events in complex terrain are also affected by uncertainties in the Initial Conditions (ICs). Initial soil moisture and its spatial distribution across the basin can potentially strongly impact runoff generation and resulting streamflow (e.g. Silvestro and Rebora, 2014; Edouard et al., 2018). Simulated soil moisture is influenced by soil hydraulic parameters, generally derived from soil texture through empirical equations (e.g. Cosby et al., 1984). Soil texture is usually obtained from national or global soil datasets (e.g. Dai et al. 2019). Soil databases can also directly affect the hydrologic response by inducing differences in terms of precipitation partitioning between surface runoff and infiltration. Lovat et al. (2019) quantified this effect and showed how different sources of soil texture can affect simulated discharge volume during flooding events in French Mediterranean basins with complex topography. In addition, in seasonally snow-covered basins, snowpack conditions prior to flooding events constitute another source of uncertainties. Jörg-Hess et al. (2015) and Griessinger et al. (2016) have shown that runoff prediction can benefit from information from a dedicated external snow monitoring system to obtain more realistic snow ICs. For example, the insertion of snow information close to peak snow accumulation in a snowpack model can significantly improve snowmelt simulations during spring and early summer (Revuelto et al., 2016; Hedrick et al., 2018). Pomeroy et al. (2012) and Fang et al. (2013) showed through model falsification of process algorithms and forest cover that prediction of streamflow in the Canadian Rockies depends on the model's ability to predict over-winter redistribution of snow by wind and forest canopy. Snowpack conditions (winter accumulation, melt timing) also affects the evolution of spring and summer soil moisture in mountain basins (e.g. Maurer and Bowling, 2014). For the June 2013 event in Southern Alberta, Fang and Pomeroy (2016) confirmed the importance of

antecedent soil moisture and snowpack conditions on runoff generation for a small (9.4 km²) and well-instrumented mountainous research basin.

ROS events constitute an additional challenge for hydrological modelling systems in mountainous terrain (Rössler et al., 2014; Corripio et al., 2017). Models need to provide an estimate of the extent to which snowmelt during ROS increases the magnitude of the streamflow response. At the basin scale, the amplitude of this response depends on the areal extent and snow water equivalent amount at the beginning of the ROS event (McCabe et al., 2007). Uncertainties about the physical processes controlling runoff generation during ROS also affect model performances. Turbulent heat exchanges during ROS events can provide 60 to 90% of the energy for snowmelt (e.g. Marks et al., 1998; Garvelmann et al, 2014) so that simulated snowmelt during ROS events is sensitive to input wind speed, temperature and humidity. The peculiarities of a ROS event near to summer solstice are also important. The June 2013 ROS event in southern Alberta showed slower snowmelt rates compared to those before and after the event due to reduced energy from solar irradiance not being compensated for by increased turbulent fluxes or longwave irradiance (Pomeroy et al., 2016a). Finally, initial snowpack properties such as initial liquid water content and cold content influence the timing and amount of runoff generation (Würzer et al., 2016). Rössler et al. (2014) showed that the contribution of turbulent fluxes as well as the fine-scale distribution of input precipitation controlled by the topography were needed to reconstruct the snow cover dynamics and the estimated peak flow during a vast ROS event in the Swiss Alps in 2011. Corripio et al. (2017) found that the atmospheric model WRF at convection-permitting scale coupled to an energy-balance snowpack model can be used to simulate with a certain success a late spring ROS event that occurred in the Spanish Pyrenees in June 2013. These studies were however limited to a single basin and only partially considered how the different sources of uncertainty affected the hydrological response during ROS events.

The present study investigates the factors (external to the hydrological model) governing the hydrological response of several mountainous basins during late-spring flood. It focusses on three main sources of uncertainty for a distributed hydrological model: (i) the resolution of the atmospheric forcings during the event (ii) the snowpack and soil moisture ICs before the event and (iii) the representation of soil texture. Noting that snowpack and soil moisture ICs, as well as the hydrological response to the ROS event are partially controlled by the hydrological model, this study does not aim at quantifying the uncertainty of the model processes and parameters on the ICs or the flood prediction.. The hydrological response of the distributed Global Environmental Multi-scale hydrological modelling platform (GEM-Hydro, Gaborit et al. 2017) during the June 2013 Flood in Southern Alberta was analyzed in detail. Twelve headwater basins covering the main river basins in the region were considered. A specific configuration of the Canadian atmospheric model GEM (Côté et al., 1998; Girard et al., 2014) was deployed over southern Alberta to produce atmospheric forcing at resolutions ranging from 1 to 10 km to drive GEM-Hydro. In particular, the precipitation simulated by GEM was combined with precipitation gauge measurements from all observation networks available in the region using the Canadian Precipitation Analysis (CaPA) system (Lespinas et al., 2015; Fortin et al., 2018) to generate reference quantitative precipitation estimation (QPE) products

for the flood at different resolutions. Uncertainties related to snowpack and soil moisture ICs were assessed by considering two estimates of snowpack conditions at peak snow accumulation: the default GEM-Hydro simulation and snow information from the SNOw Data Assimilation System (SNODAS; Barett, 2003). They were used in conjunction with two soil texture datasets in GEM-Hydro to generate a total of four different sets of snowpack and soil moisture ICs and their impact on flood modelling. The paper is organized as follows. Section 2.1 presents the study area and the different datasets used in our study. It also describes the GEM-Hydro modelling platform and the configurations of the different experiments carried out with this model. Section 3 evaluates the different hydrological inputs used in this study (initial snowpack conditions, QPE products) and examines the performances and the sensitivity of the resulting hydrological simulations. Section 4 contains a discussion of the main results of this study. Finally, concluding remarks are presented in Section 5.

## 2. Data and Methods

### 2.1 Study area and data

The study area covers the three main river basins in southern Alberta that were strongly impacted by the June 2013 flood: the Red Deer, Bow and Oldman river basins (Fig. 1). These rivers drain the Rocky Mountains and their foothills, flowing eastward towards the Canadian Prairies and eventually joining to form the South Saskatchewan River. Hourly streamflow time series were obtained from the National Hydrological Service of Environment and Climate Change Canada (ECCC) for 12 stations located in the headwaters of the Red Deer River (2 stations), Bow River (6 stations) and Oldman River (4 stations). These rivers were selected for the evaluation of GEM-Hydro simulations since they were strongly impacted by the flood and are not affected by regulation. The location of these stations is shown on Fig. 2 and their main characteristics are given in Table 1. Information on the soil texture (proportion of clay, sand and silt) for each basin were obtained from two different global datasets: the Global Soil Dataset for Earth System Model (GSDE; Shangguang et al. 2014) and the SoilGrid database (Hengl et al., 2017). The GSDE dataset has a resolution of 30 arcsec and combines existing regional and national soil information. The SoilGrid product has a resolution of 250 m and was obtained using digital soil mapping. Both datasets can be considered as reference datasets (Dai et al., 2019). The mean soil texture per basin is reported in Table 1, and Fig. S1 in the supplementary material shows the top 50-cm clay and sand content over southern Alberta for the GSDE and SoilGrid databases. SoilGrid provides grid-based spatially continuous estimation of soil texture, while GSDE provides estimations with abrupt changes at the boundaries of soil polygons. In both datasets, the soil contains more sand than clay for each basin (except for station 05BH015 with GSDE). SoilGrid reports generally more sand and less clay than GSDE, especially for the basins located in the central part of southern Alberta (from station 05BH015 in the north to station 05AA035 in the south; Table 1).

Precipitation data were obtained from four different networks (Fig. 1). Automatic synoptic stations (SYNOP network) maintained by Environment and Climate Change Canada and other organizations such as Alberta Agriculture and Forestry

provide a good coverage over the prairies and the forested foothills but only a sparse coverage of the Canadian Rockies, particularly in Banff National Park where the stations are mainly located in the valleys (see for example the upper Bow river basin, Fig. 1). Therefore, to fill in data gaps, precipitation data were taken from Alberta Environment and Parks because of their good coverage of the higher elevations of the Canadian Rockies headwaters. Data from high elevation mountain stations of the Canadian Rockies Hydrological Observatory (CRHO, University of Saskatchewan, https://www.usask.ca/hydrology/CRHO.php) were also used in the upper Bow River Basin. SYNOP, ABE and CRHO data consist of hourly data that were cumulated to derive 6-h precipitation amounts for CaPA (Sect. 2.3.2). Finally, stations from the American Cooperative Observer Network (COOP) reporting 6-h precipitation amounts were included. These stations are mainly located in the US and on the western side of the study area and were referred to as stations from the SHEF network (Standard Hydrometeorological Exchange Format) in Lespinas et al. (2015). They were not included in the 6-h operational version of CaPA at the time of the flood.

Information on snow conditions before the flooding event and during the winter of 2012/2013 were obtained from 11 automatic snow pillows from Alberta Environment and Parks located in southern Alberta. These stations measure hourly snow water equivalent (SWE). Additionally, outputs from the SNOw Data Assimilation System (SNODAS) from the US National Operational Hydrologic Remote Sensing Center (NOHRSC) were included in this study. SNODAS estimates various snow properties (including SWE and snow depth) by merging satellite, airborne, and ground-based snow data with a numerical simulation of snowcover (Barrett, 2003). SNODAS data are available at 1-km spatial resolution and 24-hour temporal resolution and cover continental US as well as part of Canada (up to 54° N). In southern Alberta, the snow pillows from Alberta Environment and Parks are included in SNODAS since they provide relevant information for mountain snow conditions, including those affecting the Columbia River flowing from Canada to the US. The ECCC snow analysis (Brasnett, 1999) was not considered in our study since its spatial resolution at the time of the event (10 km) was too coarse to accurately represent snow conditions in the complex topography of the Canadian Rockies.

## 2.2 The GEM-Hydro modelling platform

GEM-Hydro is a distributed hydrological modelling platform developed at ECCC for hydrological forecasting and prediction across Canada (Gaborit et al., 2017). It includes two components: (i) the GEM-Surf surface prediction system (Bernier et al. 2011) and (ii) the WATROUTE routing scheme (Kouwen, 2010). A recent experimental GEM-Hydro version also includes the Canadian Land Data Assimilation system based on satellite data (CaLDAS-Sat, Carrera et al. 2019) for real-time forecasting purposes. However, CaLDAS was not used in this study. Instead, all GEM-Hydro simulations were performed in an open-loop manner, without any surface or hydrological data assimilation, which would hinder the effects of the external factors assessed here. GEM-Hydro has been extensively evaluated over the Great Lakes basin (Gaborit et al., 2017) and WATROUTE is a component of the Water Cycle Prediction System running operationally at ECCC over the Great Lakes and St. Lawrence River basin (Durnford et al., 2018). This system provides daily operational forecasts of

streamflow and lake levels in this region. The GEM-Hydro model follows the same approach as the Weather Research and Forecasting Model hydrological modelling system (WRF-Hydro, Gochis et al., 2015) and represents surface and sub-surface hydrological processes (GEM-Surf) with runoff, lateral flow and drainage provided to a routing model (WATROUTE) to compute streamflow.

GEM-Surf simulates the evolution of six types of surfaces: land, glacier, inland water, sea, ice (over sea or lakes) and urban areas. Over land, it uses the SVS (Soil, Vegetation and Snow) land surface scheme (Alavi et al., 2016; Husain et al., 2016). SVS relies on a multiple energy budget approach to solve the energy and mass balance at the earth surface. The land tile is divided into three main covers: bare ground, low vegetation and high vegetation. Then, SVS solves independent energy budgets using a force restore approach for bare ground, vegetation, snow covering bare ground and low vegetation, and snow

below high vegetation (Husain et al., 2016). SVS uses a single-layer energy balance scheme to simulate the snowpack evolution. Water movement in the soil column is represented in SVS using a multi-layer scheme and assuming Darcian flow between the soil layers (Alavi et al., 2016). SVS simulates surface runoff as well as lateral flow, accounting for sloping surfaces (Soulis et al., 2011), and base flow from the bottom layer. WATROUTE routes surface runoff, lateral flow and base flow produced by SVS to the basin outlet. In WATROUTE, runoff and lateral flow directly feed the streams while base flow

is provided to a lower zone storage compartment, representing surficial aquifers, which releases water to the streams using a power function.

## 2.3 Model configuration and experiments

In this study, GEM-Hydro was used in a stand-alone (offline) and open-loop (no assimilation) mode to produce a hindcast of the hydrology of the June 2013 flood. The model was driven by a set of atmospheric data specifically generated for this

study, which combined short-term numerical weather forecast and precipitation analysis. These atmospheric forcings were produced at different resolutions to assess the sensitivity of GEM-Hydro to the resolution of forcing data. The impact of snowpack and soil moisture ICs and soil texture on streamflow simulation was also assessed. In the following, the main features of the atmospheric forcing data are detailed, as well as the different experiments carried out with GEM-Hydro.

### 2.3.1 Atmospheric simulations

The latest operational version of the GEM NWP model used at ECCC (Côté et al., 1998; Girard et al., 2014; Caron et al., 2015; Milbrandt et al., 2016) was used to generate up-to-date hindcasts of the atmospheric conditions during the flooding event. GEM was configured with three one-way nested domains with 10, 2.5 and 1-km grid spacing centred over the study area in southern Alberta. This nested approach has been applied in previous studies at ECCC (e.g. Vionnet et al., 2015; Leroyer et al., 2018). Two domains at the kilometre scale (2.5 and 1 km) were used since the previous studies of Li et al.

(2017) and Milrad et al. (2017) showed that models at convection-permitting resolution performed best for this event. Details of the simulation domains are listed in Table 2. Figure 3 shows the extent of the 2.5 and 1-km grids over western Canada.

Table 2 also details the physical parameterizations used in GEM at different resolutions. The model at 10-km grid spacing has the same settings as the operational version of the Regional Deterministic Prediction System (RDPS, Caron et al. 2015) used at ECCC. The 2.5 and 1-km grids used the latest configuration of the High Resolution Deterministic Prediction System (HRDPS, Milbrandt et al., 2016) which includes the P3 microphysical scheme (Predicted Particle Properties, Morrison and Milbrandt, 2015). Land surface characteristics were specified at each grid cell of each domain using different geophysical datasets described in Table 3.

GEM was used to produce 12-h meteorological forecasts 4 times per day (at 00, 06, 12 and 18 UTC) from 18 June 2013 00 UTC to 22 June 2013 12 UTC. Initial and lateral boundary conditions for the 10-km GEM model were taken every hour from the analysis and forecasts of the operational RDPS which covers most of North America. The 2.5-km (1-km) GEM model used lateral boundary conditions obtained from the 10-km (2.5-km) GEM model every 12 minutes. Model variables were stored at an hourly frequency. GEM forecasts at different resolutions are available on the Federated Research Data Repository (Vionnet et al., 2019).

### 2.3.2 Precipitation analysis

To test their influence on the hydrology of the June 2013 flooding event, as simulated by GEM-Hydro, new QPE products were generated. They rely upon the Canadian Precipitation Analysis (CaPA) system (Mahfouf et al., 2007; Lespinas et al., 2015; Fortin et al., 2018). CaPA combines precipitation observations with a background field obtained from a short-term meteorological forecast using optimal interpolation to produce a QPE on a regular grid. Precipitation observations consisted of rain gauges and radar QPEs (Fortin et al., 2015). The current operational version of CaPA at ECCC produces 24-h and 6-h QPE at 10-km resolution over a domain covering most of North America and at 2.5-km resolution over a domain covering most of Canada (Fortin et al., 2018).

Three precipitation datasets (6-h accumulation) were produced using CaPA for the June 2013 Flood (from 18 June 2013 12 UTC to 22 June 2013 12 UTC) with the aim of evaluating the impact of precipitation background resolution on the QPE products and the hydrological response simulated by GEM-Hydro. The resolution of the various precipitation background was that of the GEM atmospheric simulations described in Sect. 2.3.1 (10, 2.5 and 1 km). The forecast used to generate precipitation forcing data had a lead time between 6 and 12 hours to avoid spin-up errors. Indeed, the precipitation amounts are currently underestimated during the first few hours of the forecast. Each CaPA experiment included all the ground-based precipitation data from the SYNOP, ABE, CRHO and COOP networks (Sect. 2.1) to obtain the best density of stations in the Rockies and their foothills (Fig. 1). CaPA is designed to deal with different precipitation networks and applies a spatial consistency test to identify and remove observations with large errors from the final analysis. More details about the Quality-Control procedures in CaPA are given in Lespinas et al. (2015). Radar data were not included in the different QPE products since they underestimated precipitation amounts in this mountainous region for this event (Kochtubajda et al., 2016), in

agreement with earlier studies in other mountainous regions (e.g. Germann et al., 2006). The data from each CaPA experiment are available on the Federated Research Data Repository (Vionnet et al., 2019).

### 2.3.3 Surface and hydrological modelling

GEM-Hydro was used to simulate the evolution of the surface and hydrological conditions during and after the flooding
event (from 18 June 2013 12 UTC to 26 June 2013 12 UTC). GEM-Hydro simulations combine successive integration of GEM-Surf (including SVS) and WATROUTE. In this study, GEM-Surf ran on the same grid and used the same vegetation and topographic fields as did GEM 1 km (Tables 2 and 3). Seven layers down to 1.4 m depth were used to represent the vertical layering of soil, following Gaborit et al. (2017). The routing with WATROUTE was implemented over a 1-km grid covering the Red Deer, Bow and Oldman river basins (Fig. 1). Flow directions were derived from the HydroSHEDs
database (Lehner et al., 2008). The default (uncalibrated) version of GEM-Hydro was used in the study. Parameters in SVS were the same as in the version of SVS used for NWP (Alavi et al., 2016; Husain et al., 2016). Parameters in WATROUTE were derived using the standard procedures implemented in the operational hydrological system of ECCC (Durnford et al., 2018). Manning's roughness parameters obtained with this method were then adjusted to optimize simulated peak flow timing and amplitude (see Sect. 2 of the supplementary material) since erosion and changes in river channels were reported
for many rivers impacted by the June 2013 flood (Pomeroy et al., 2016b) so that the default routing parameters used in WATROUTE may not be suitable for this extreme event. Calibrating the rest of GEM-Hydro is a challenging task due to model computation time (Gaborit et al., 2017). The main purpose of GEM-Hydro as used by ECCC is to produce multi-scale hydrological simulations over large river basins including ungauged basins. Calibrating it locally on each gauge from upstream to downstream is not feasible in practice due to model computation time, and would violate parameter consistency
in space as well as lack robustness in time (Gaborit et al., 2017). For these reasons, GEM-Hydro was not calibrated in this study, except for the adjustment of Manning's roughness parameters.

Twelve simulations of the flood were carried out with GEM-Hydro. They differ in terms of resolution of the atmospheric driving data, precipitation forcing, ICs and soil texture. Table 4 describes their main characteristics. The atmospheric forcings required for GEM-Surf include shortwave and longwave radiation at the surface, surface pressure, air temperature,
specific humidity, precipitation, and wind. Continuous atmospheric forcings from 18 to 22 June were obtained from successive GEM forecasts at different resolutions (Table 2). For the period of 22nd to 26th June, only operational forecasts of the RDPS at 10-km resolution were used as forcings. In all instances, forecast hours 6 to 11 were used to avoid spin-up errors. For the forecasts at 10 and 2.5 km, surface pressure, air temperature, humidity, and precipitation phase were downscaled to correct for elevation differences between the grid of the forecasts and the 1-km grid used by GEM-Surf as in
Bernier et al. (2011). No correction was applied for the wind field. The different precipitation datasets produced with CaPA were used as precipitation forcing data. Six-hourly precipitation outputs from CaPA were then disaggregated into hourly accumulations in accordance with the temporal precipitation structure of the GEM forecast.

ICs for the surface variables on 18 June 2013 12 UTC were taken from four different 1-km GEM-Surf experiments. Each experiment was initialized on June 1[st], 2012 using RDPS 10-km surface fields interpolated to the 1-km grid and then integrated for more than one year until 18 June 2013 12 UTC to allow a sufficient spin-up of surface variables, especially soil moisture. The hourly meteorological forcings for this one-year spin-up period were taken from successive RDPS forecasts at 10-km resolution issued 4 times per day (00, 06, 12 and 18 UTC). Similar to all other experiments, 6-11 forecast hours were used. The meteorological forcings were then downscaled to the 1-km GEM-Surf grid using the method described in Bernier et al. (2011). Following Carrera et al. (2010), precipitation was taken from the RDPS from 1 November 2012 to 30 April 2013 and from the operational version of CaPA at 10 km for the rest of the period. CaPA underestimates solid precipitation amounts in mountainous areas (Schirmer and Jamieson, 2015), since few reliable precipitation gauges are available at high altitude (Fortin et al. 2018) and gauge observations affected by wind-induced undercatch of winter precipitation (e.g. Nešpor and Sevruk, 1999) can be still used in CaPA despite the QC procedure supposed to remove them from the analysis (Lespinas et al. 2015). Two GEM-Surf simulations used the GSDE soil database and the two remaining used the SoilGrid database (Sect. 2.1).

For each pair of GEM-Surf simulations with a given soil database, two configurations were used for the snow and soil conditions: an Open Loop (*OPL*) configuration and a second configuration using snow information from a dedicated external snow prediction system: the SNODAS system (configuration *SND*). In the first configuration, *OPL*, simulated snow conditions evolved in response to the atmospheric forcing without constraint from observations. Using a similar setup, Carrera et al. (2010) and Separovic et al. (2014) have shown that GEM-Surf tends to underestimate snow water equivalent on the ground (SWE) since the 10-km forcing from RDPS does not resolve local topographic snowfall enhancement. For this reason, the second configuration, *SND*, used estimated SNODAS SWE to correct GEM-Surf modelled SWE over the region (Sect. 2.1). The correction was carried out on May 1[st] 2013, close to peak SWE to adjust for over-winter bias in accumulated precipitation and to allow for the adjustment of soil moisture in the period prior to the flooding event. On May 1[st] 2013, simulated snow depth in GEM-Surf was used to estimate SWE using simulated snow density, and this SWE was adjusted to match SNODAS SWE. After insertion on May 1[st], the *SND* simulation kept running until 18 June 2013 12 UTC using the same atmospheric forcing as simulation *OPL*. The *OPL* and *SND* simulations provide two estimates of snow and soil moisture ICs at the beginning of the flooding event (18 June 2013). Note that the simulations with the different soil texture databases will also result in different soil moisture ICs.

## 3. Results

### 3.1 Quantitative precipitation estimation

Figure 4 shows the maps of cumulative precipitation over the region during the 3-day flooding event obtained with CaPA at different resolutions as well as the differences between the CaPA experiments. In addition, Figure 5 gives the average total

precipitation for the 12 headwater basins used for the evaluation of hydrological simulations (Table 1). The three QPE products show similar precipitation patterns, in agreement with the fact that they include the same station networks. Regions with cumulative precipitation depths exceeding 200 mm are found on the foothills of the Rockies, mainly in the Bow River Basin. In *CaPA 1 km* and *CaPA 2.5 km*, local maxima were estimated to have exceeded 250 mm in the upper Elbow, Highwood and Red Deer River (Fig. 4b and c) with average precipitation above 200 mm for these river basins (Fig. 5). The minima of total precipitation for each dataset is obtained for the basin of the Bow River at Banff (station 05BB001) which is located on the leeside of the first significant topographic barriers on the eastern side of the Canadian Rockies (Fig. 2). The spatial pattern of these three QPE products differ from the operational version of CaPA at the time of the flood (not shown), which presented an underestimation of cumulative precipitation on the foothills of the Rockies, in particular in the upper Elbow and Highwood river basins. This operational precipitation analysis at the time of the flooding event did not include stations from the CRHO, COOP and ABE networks. The resolution of the precipitation background affects the precipitation analysis as shown on Fig. 4 and 5. Compared to *CaPA 10 km,* additional details associated with the influence of the topography on the precipitation background are seen at 2.5 km (Fig. 4d) and at 1 km (Fig. 4e) in areas of complex terrain, especially in the Bow River and Red Deer River basins. The patterns of precipitation differences between the CaPA experiments at kilometric scale and the one at 10-km closely follow topography contours in these regions. Larger precipitation amounts are found in areas of high-elevation whereas valleys tend to receive less precipitation due to the smoothing of the topography in the precipitation background at 10 km. Overall, *CaPA 1 km* provided the largest estimates of total precipitation for all the river basins located in the Red Deer River and Bow River basins (Fig. 5), with *CaPA 2.5* km providing the second largest estimate for seven basins out of eight. Results are less systematic for the Oldman River basin with a generally similar estimate of total precipitation among the three datasets.

Table 5 provides a detailed evaluation of the different QPE products against SYNOP, CRHO, COOP and ABE station observations. A "leave-one-out" cross validation method was used, where stations are removed one by one and an analysis value is estimated at their location using stations in the vicinity, following Lespinas et al. (2015). This analysis value was then compared to that observed. Table 5 presents the performances of the QPE products during two phases of the life cycle of the flooding event as identified by Kochtubajda et al. (2016) and Li et al. (2017): a convective period until 20 June 12 UTC associated with an intense lightning activity followed by a stratiform period until the end of the flood event. Overall, the three QPE products agree well with the observations for the three main basins. *CaPA 10 km* had a consistent negative bias mainly due to an underestimation of precipitation during the convective phase for the Oldman and Bow River basins. The precipitation background from the convective-permitting version of GEM at 1 km provided the best performances (lower RMSE, and higher correlation coefficient) during the convective period for all three basins. However, it suffered from reduced performances during the stratiform phase with a tendency to overestimate precipitation on the Oldman and Bow River basins. This explained why the overall best performances were obtained with CaPA 2.5 km despite limited performances for the Oldman and Red Deer Rivers during the convective phase. These results are consistent with previous

studies by Li et al (2017) and Milrad et al. (2017) that showed that atmospheric models at convection-permitting resolution performed best for this event. As detailed by Milrad et al. (2017), this is mainly due to (i) an improved representation of the orographic ascent that contributed to the magnitude of the extreme rainfall and (ii) an anchoring and increasing duration of the precipitation on the Eastern side of the Rockies. It should be noted that CaPA at 2.5 km and 1 km are still subject to localization errors of the precipitation maxima in the background from GEM at convection-permitting resolution which may affect the results of the leave-one out evaluation. *CaPA 10 km* is potentially less affected due to a smoother precipitation background (e.g. Clark et al., 2016).

Improved performance in estimating cumulative precipitation during the flood is itself insufficient for accurate hydrological simulations, and the precipitation forcing should also reproduce the sub-daily precipitation amounts well (e.g. Singh, 1997). Figure 6 uses quantile-quantile (QQ) plots to compare the distribution of observed and analyzed precipitation for three datasets (*CaPA 10 km*, *CaPA 2.5 km* and *CaPA 1.0 km*) and for the three main river basins in southern Alberta. As in Table 5, analyzed precipitation is derived from the leave-one out method. The concordance correlation coefficient (Lawrence and Lin 1989), *RC*, is used to estimate the agreement between analyzed and observed 6-h precipitation amounts. *RC* ranges from -1 to 1. 1 (-1) indicates perfect concordance (discordance) whereas 0 indicates no correlation. Over the Bow River Basin, all datasets reproduced the distribution of 6-h precipitation amounts well, except for amounts above 60 mm. For this basin, the best performance was obtained with *CaPA 1.0 km* (especially over 40 mm). The same conclusion can be made for the Oldman River Basin where *CaPA 10 km* and *CaPA 2.5 km* underestimated the 6-h precipitation depths greater than 35 mm. Finally, the three versions of CaPA had similar performance over the Red Deer River Basin, where no values greater than 40 mm were observed. Overall, Fig. 6 illustrates the added value of precipitation analysis at the kilometre scale for better capturing the intensity of extreme mountain precipitation that may be important for hydrological simulations.

### 3.2 Snow and soil moisture initial conditions

Figures 7a and 7b compare simulated initial SWE on May 1st 2013 from the experiments *OPL* and *SND*, date at which SNODAS SWE was inserted into the *SND* simulation (see Sect. 2.3.3). Both simulations present similar large scale snowcover patterns with snow absence in the prairies extending north-eastward of the study area and in the low elevation Columbia River Valley of British Columbia and its tributaries. However, the simulations differ substantially in their estimation of SWE. The *OPL* simulation shows less SWE than does *SND* in the mountain headwaters of the Bow, Oldman and Red Deer rivers basins. A comparison of *OPL* simulations with SWE observations from ABE's snow pillows (Fig. 7c) reveals a general underestimation of SWE (except in the upper Red Deer River Basin). *SND* presents a better agreement with snow pillow SWE observations (Fig. 7d), except for the upper Elbow and Highwood river basins. Such agreement was expected since SWE observed by ABE snow pillows is included in the SNODAS analysis. The insertion of SNODAS SWE in the *SND* simulation strongly modified the high mountain snowpack conditions prior to the flooding event as shown on Fig. 7e and 7f Figure 8 summarizes the basin-averaged initial SWE for the 12 mountainous basins considered in this study.

Very little snow remained in the *OPL* simulations in the headwater basins of the Oldman and Bow River basins, which is inconsistent with the presence of high mountain snow reported prior to the flood (Liu et al., 2016; Pomeroy et al., 2016a, b). On the other hand, the *SND* simulations predicted the persistence of snowpacks at high elevations prior to the event, mainly in the Upper Bow River Basin (detailed topographical heights are given in Fig. 1) for basins 05BB001, 05BJ0010 and 05BL019 which is consistent with observations in the region. A better agreement is found between the *OPL* and *SND* simulations for the upper part of the Red Deer basin (stations 05CA004 and 05CA009). As expected, the initial SWE prior to the flooding event is not sensitive to the choice of the soil database (Fig. 8).

Figure 8 also details the basin-averaged initial soil saturation over the top 50-cm of the soil column for the 12 mountainous basins considered in this study. In all experiments, initial soil saturation shows a general tendency to decrease from the north (Red Deer River) to the south (Oldman River) of the study area. A good agreement between the experiments is found for the Red Deer River basin (stations 05CA004 and 05CA009), where snow is present in all experiments. For the rest of the basins, simulations using the SoilGrid database systematically provided lower estimates of soil saturation compared to simulations using the GSDE database due to a coarser soil texture in SoilGrid (Table 1). The insertion of SNODAS SWE in the *SND* simulations affected initial soil saturation as well and led to higher soil saturation. The largest differences between the simulations *OPL* and *SND* were found for basins of the Bow River at Banff (station 05BB001) and the Highwood River (station 05BL019), where simulated snowpack differs substantially as well. However, differences in initial soil saturation between simulations *OPL* and *SND* were also found for basins where difference in SWE prior to the flooding event were low (stations 05AA008 and 05AA022 for example). This illustrates the indirect impact of inserting SNODAS close to peak snow accumulation; modifying the amount of snow available for melt during the period prior to the flood and eventually leading to differences in soil saturation at the time of the flooding event.

### 3.3 Hydrological simulations

Initial snow and soil conditions from four GEM-Hydro simulations were used in combination with the atmospheric forcing and the QPE products at different resolutions to provide initial and boundary conditions for twelve hydrological simulations of the flooding event over 18 June to 25 June 2013 (Table 4). These simulations were evaluated using discharge measurements from twelve headwater basins of the Red Deer, Bow and Oldman rivers (Fig. 2 and Table 1) Three error metrics were used to evaluate the performances of all twelve hydrological simulations of the flood: (i) the percent bias, *PBIAS*, (ii) the relative difference of peak flow amplitude, $\Delta Q_{max}$, and (iii) the Nash efficiency coefficient, *NSE* (see Appendix A for their definition). Figure 9 shows the distributions of these error metrics for the different experiments. Error metrics for each station and each GEM-Hydro experiment are also shown on Fig. S3 in the supplementary material.

Figure 9a shows that all simulations using the *OPL_GSD* and *OPL_SLG* configurations present an underestimation of the flood discharge volume for most of the gauging stations. For these configurations, increasing the horizontal resolution of the precipitation analysis and the atmospheric forcing from 10 to 2.5 km contributed to reducing the overall negative *PBIAS* for

discharge at most of the river gauging stations and also improved the estimation of peak flow magnitude and *NSE* distribution (Fig. 9b and c). The atmospheric forcing at 1km brought additional improvements with for example a reduction of median PBIAS from -26% to -13% and an increase in NSE from 0.28 to 0.5 when considering *1km_OPL_GSD* instead of *10km_OPL_GSD*. The general underestimation of flood volume using the *OPL_GSD* or *OPL_SLG* configurations may be explained by the underestimation of initial SWE in the basin headwaters in the *OPL* configurations as shown in Fig. 7. For this reason, the *SND* configuration, which incorporates SNODAS SWE to replace the modelled peak snow accumulation, was considered as an alternative to obtain snow and soil moisture ICs. Figure 9 shows that this substitution had a large impact on hydrological simulations and improved the overall estimation of flood volume for any atmospheric forcing and soil database. However, it led to a general tendency to overestimate peak flow amplitude (especially with the atmospheric forcing at 2.5 and 1 km) and increased the spread of model performance in terms of *NSE*. Finally, changing the soil database from GDSE to SoilGrid systematically decreased the flood volume and the peak flow magnitude, consistent with the lower estimates of soil saturation obtained with SoilGrid at the beginning of the event (Fig. 8). The use of SoilGrid also reduced the spread of *NSE* and generally improved the median *NSE* compared to GSDE.

The sensitivity of the hydrological responses to the different sources of uncertainty was quantified for each hydrometric station and each error metric using the method described in Appendix B. Results are shown on Fig. 10, and Fig. 11 details hydrographs simulated at four hydrometric stations for selected GEM-Hydro simulations. The sensitivity to the resolution of the atmospheric forcing is found at most of the stations (Fig. 10). It is ranked among the two main sources of uncertainty at all twelve stations for *PBIAS*, ten stations for the difference in peak flow amplitude and at nine stations for *NSE*. In particular, stations located in the Red Deer River basin (05CA004 and 05CA009) are both more sensitive to the resolution of the atmospheric forcing than the other sources of uncertainty. Figures 11a and 11b illustrate how an increase in the resolution of the atmospheric forcing can modify the simulation of the flood dynamics. At these two stations, using an atmospheric forcing at 1 km instead of 10 km for given ICs (*SND_GSD* or *SND_SLG*) improved the estimation of the peak flow values. The soil database is ranked third among the sources of uncertainty on the flood discharge volume at ten out of twelve stations. Its influence increases for the simulation of the peak flow amplitude and for the Nash-Sutcliffe efficiency but it never appears as the main source of uncertainty, except for the station 05BH015 for *NSE*. As mentioned previously, the choice of the soil database impacts the peak flow amplitude with SoilGrid systematically leading to lower peak flow values compared to GSDE, especially for stations 05AB041 (Fig. 11a) and 05BH015 (Fig. 11b). The flood dynamics are also impacted with a higher response time with SoilGrid for some stations (05BH015 and 05BL012 for example; Fig. 11b and 11c) because SoilGrid is associated with lower estimates of initial soil saturation and a coarser soil texture than GSDE (Fig. 8 and Table 1). The sensitivity to the soil texture was very similar when using the configurations *OPL* and *SND* despite higher initial soil saturation in configurations *SND*.

The sensitivity to snow and soil moisture ICs varies greatly from one station to another (Fig. 10) and is associated with different ICs for SWE and soil moisture in configurations *OPL* and *SND* (Fig. 8). The sensitivity can be very low at stations

such as 05BH015 or 05AB041 which were characterized with no snow and similar soil saturation on 18 June 2013 in *OPL* and *SND*. The sensitivity remains low for stations where snow is present in both configurations (stations 05CA004, 05CA009 and 05BG010). The snow ICs become the main source of sensitivity for all scores for stations where almost no snow is present in *OPL* contrary to *SND*. Figures 11c and 11d illustrates the hydrographs simulated at two of these stations.

Configuration *SND* led to an increase in flood discharge volume and peak flow but did not modify the streamflow timing at the onset of the flood. Finally, stations 05BL019 and 05BB001 present a very large sensitivity to the snow ICs, in agreement with the two very different estimates of initial SWE in *OPL* and *SND* (Fig. 8). Figure 12 compares the evolution of basin-averaged SWE and streamflow for four GEM-Hydro simulations at these two stations. Figure 12a shows that *SND* experiments overestimated the peak flow and the flood discharge volume at station 05BL019. This basin experienced both

intense precipitation and rapid snowmelt on 20 June 2013. It should be noted that the SNODAS analysis overestimated SWE close to peak accumulation at the snow pillow located in the headwaters of the Highwood River Basin (Fig. 7). The Bow River at Banff (05BB001) was characterized by a steady increase in streamflow on 20 and 21 June 2013 that was partially captured by the *OPL* experiments. *SND* experiments led to a substantial overestimation of flood volume at this station. Streamflow was also overestimated prior to the flooding event in the *SND* experiment (Fig. S4 in supplementary material),

suggesting that the insertion of SNODAS SWE led to a strong SWE overestimation for this basin before and during the flooding event.

Figure 12 shows that the decrease in basin-averaged SWE between 18 and 25 June 2013 changed with the initial resolution of the atmospheric driving data. It reached -53% and -44% in the *1.0km_SND_GSD* simulation compared to only -40% and -35% in the *10km_SND_GSD* simulation for station 05BL019 and 05BB001, respectively. To better understand these

differences, Figure 13 illustrates the contributions of the energy balance terms to the overall energy input to the snow cover during the flooding event (expressed in millimetres melt equivalent, mm m.e.), referring to the energy required to conduct the phase change of ice to liquid water as in Würzer at al. (2016). Three energy balance terms were considered: net radiation, advective energy brought by rain and the net turbulent fluxes (latent and sensible heat). The calculations were carried out for the eight basins where snow was initially present in configuration *SND* (Fig. 8). Figure 13 shows a consistent response

across all basins: the atmospheric forcing at 10 km led to lower energy inputs to the snow cover during the flooding event than the atmospheric forcing at 2.5 and 1 km. This is due to a low contribution of the turbulent fluxes to the total simulated energy input using an initial 10-km forcing whereas turbulent fluxes provide most of the simulated energy input when using forcing at 2.5 and 1 km resolution. The change in the magnitude of the turbulent fluxes is explained by a change in the 10-m wind speed used to drive GEM-Hydro (Fig. 13, Bottom). At 10 km, wind speeds were taken from a configuration of GEM

using (i) an effective aerodynamic roughness length depending on vegetation type and on the subgrid-scale orography (Zadra et al., 2008) and (ii) a parameterization of low-level blocking (Palmer et al., 1986). Both physical parameterizations reduced the near-surface wind speed in regions of complex orography. In addition, no downscaling accounting for the topography was applied to the 10-km wind when interpolating it on the 1-km GEM-Hydro grid. At 2.5 km and 1 km, these two

parameterizations were not used and the aerodynamic roughness length was only function of the vegetation type. This led to larger wind speed, higher turbulent fluxes and finally higher total snowmelt at the basin scale when using atmospheric forcing at kilometric scale.

## 4. Discussion

This paper has examined the sensitivity to different external factors of the hydrological response simulated by the GEM-Hydro hydrological modelling platform during the June 2013 Flood in southern Alberta. The term "external" refers to the fact that the sources of uncertainty studied here do not include the model internal processes or parameters, but are rather due to sources of uncertainty outside of the hydrologic model, namely atmospheric forcing resolution, snow amount at peak accumulation and soil texture database. The simulation of this extreme event offers insights into the key external factors
governing the prediction skill of late spring flooding events in mountainous regions. The twelve main unregulated headwater basins in southern Alberta were considered to include basins characterized by contrasted soil and snow ICs prior to the flooding event as well as different total precipitation amount during the flood.

This study used three sets of atmospheric forcings at different horizontal resolutions (10, 2.5 and 1 km) to assess how forcing
resolution influences GEM-Hydro simulations. In particular, three QPE products were generated using the Canadian Precipitation Analysis (CaPA) system with different horizontal resolutions of the GEM precipitation background. Increasing the resolution of the GEM precipitation background from 10 km to kilometre scales (2.5 and 1 km) improved the accuracy of the final QPE products in terms of cumulative precipitation as well as the intensity of extreme precipitation during the flooding event. Improvements were mostly found during the convective phase of the flooding event. Streamflow simulated
by GEM-Hydro benefited from these improvements in forcing data, both in terms of flood discharge volume and peak flow magnitude. The largest difference was found in moving from 10-km QPE to the 2.5 km QPE which invoked precipitation forcing data from a convection-permitting NWP system. Li et al. (2017) and Milrad et al. (2017) have shown that atmospheric models at such resolution were required to capture the spatial and temporal variability of extreme precipitation during the June 2013 flooding event. This has also been shown in other mountainous regions (e.g. Richard et al., 2007;
Weusthoff et al., 2010; Rasmussen et al., 2011, Lundquist et al., 2019). The results shown here confirm that improving the realism of the precipitation model in complex terrain also improves the precipitation analysis. This shows the potential to improve hydrological forecasting by using the latest operational version of CaPA running at 2.5 km over Canada (Fortin et al., 2018). On the other hand, only marginal improvements in the intensity of extreme precipitation and slight degradation of cumulative precipitation predictions were found when increasing the resolution of the precipitation model from 2.5 km to 1
km. These findings are in agreement with Pontoppidan et al. (2017) who showed only minor improvements from 3 km to 1 km in atmospheric simulations of an intense precipitation event in Western Norway. Further evaluations are required in mountainous terrain to clearly assess the benefits and limitations of a 1-km precipitation analysis. Finally, it should be noted

that this study used precipitation analysis at different resolutions instead of precipitation forecasts to drive GEM-Hydro in hindcast mode. This strategy was used to keep relatively constant the uncertainty associated with the precipitation forcing as in Lin et al. (2018) and to avoid the increase in precipitation errors with increasing lead time (Zappa et al., 2011). In another context, convection-permitting meteorological ensemble prediction systems could have been used to sample the uncertainty associated with precipitation forecasts (e.g. Vincendon et al., 2011), but this falls beyond the scope of this study.

Additional sensitivity to the resolution of the atmospheric forcing was found in basins that were initially covered by snow. In these basins, rain falling on a late-lying snowpack at high elevations and concomitant humid weather conditions induced rain-on-snowmelt and enhanced runoff generation from rainfall alone, such that there was more runoff than storm precipitation (Fang and Pomeroy, 2016). GEM-Hydro simulations showed that the increase in runoff generation during ROS depends on the resolution of the atmospheric forcing. When using forcing at 2.5 km and 1 km, turbulent fluxes were the main energy source for snowmelt for all basins as found in previous studies for other ROS events (e.g. Marks et al., 1998; Würzer et al., 2016; Corripio et al., 2017), though lack of consideration of forest canopy dampening of turbulent fluxes in GEM-Hydro may have overestimated this effect (Pomeroy et al., 2012). At 10-km resolution the snowpack energy balance was dominated by net radiation since the contribution of turbulent fluxes was lower than at 2.5 km and 1 km due to lower wind speed at 10-km grid spacing associated with the physical parameterizations of orographic blocking and orographic roughness used in GEM at 10 km (Palmer et al. 1986; Zadra et al., 2008). Pomeroy et al. (2016a) showed that net longwave radiation, sensible and latent heat flux and advected energy were the main energy sources for snowmelt in the alpine zone of a small mountainous basin during the June 2013 Flood as a result of cold air and low wind speed. Forest canopy would further reduce the turbulent fluxes (e.g. Marks et al., 1998; Reba et al., 2012). This suggests that the role of the turbulent fluxes found in the GEM-Hydro simulations driven by the atmospheric forcing at 2.5 and 1 km grid-spacing may have been overestimated. Nonetheless, the impact of the resolution of the atmospheric forcing on the snowmelt dynamics during a ROS event highlighted in this study is expected to be found for many ROS events due to the general substantial contribution of turbulent fluxes to snowmelt rates during ROS and their dependence on wind speed (e.g. Marks et al., 1998; Gravelman et al., 2014).

Differences in snow and soil moisture ICs resulting from uncertainties on peak snow accumulation were the main source of uncertainty for estimating flood discharge volume from half of the headwater basins considered in this study. This highlights the role of antecedent SWE conditions on the simulated hydrological response during ROS events (McCabe et al., 2007). Two configurations of GEM-Surf were considered to obtain snow ICs. The first set of hydrological simulations used snow ICs taken from an open-loop simulation of GEM-Surf and presented a systematic underestimation of flood volume in GEM-Hydro simulations with all atmospheric forcing and soil texture databases. This is mainly due to the absence of snow in the high elevation headwater basins at the time of the flood, which resulted from a strong and systematic underestimation of winter snow accumulation. These results are consistent with the earlier findings of Carrera et al. (2010) and Separovic et al. (2014) and are mainly explained by an underestimation of mountain winter precipitation simulated by GEM at 10-km grid

spacing (Schirmer and Jamieson, 2015). The insertion of SWE data from a dedicated snow monitoring system was considered as an alternative to the Open Loop simulation as in Jörg Hess et al. (2015). The SWE from the SNODAS operational snow analysis system (Barrett, 2003) near the time of peak snow accumulation was inserted to correct the negative bias in winter snow accumulation and to obtain more realistic snow and soil moisture ICs at high elevation. GEM-

Hydro simulations including SNODAS SWE showed contrasting results. Overall, it improved the estimation of the flood volume and peak flow when combined with atmospheric forcing at 2.5 and 1 km. However, the large overestimation of flood volume found at the stations located in the headwaters of the Bow and Highwood River basins suggests an overestimation of basin-average SWE prior to the flood in these basins. This is in agreement with Teufel et al. (2017) who found an overestimation of daily simulated streamflow when using snow ICs from SNODAS in their hydrological simulations of the

June 2013 flood. In a recent study, Lv et al. (2019) reported a strong overestimation of SNODAS SWE in sub-alpine forest elevations in the Canadian Rockies due to a lack of representation of canopy interception and sublimation losses in the SNODAS snow model. They have also found a poor agreement in open alpine environments due to missing wind-induced snow redistribution processes in SNODAS. Therefore, the large sensitivity of the hydrological responses associated with snow ICs found in this study results from very uncertain snowpack conditions. This is explained by the absence of a

reference distributed snow analysis product in the Canadian Rockies. The sensitivity to snow ICs may be lower in other mountainous areas where such products are available such as the Swiss Alps (Jörg Hess et al., 2015; Griessinger et al., 2016) or the Sierra Nevada Mountains in California (Hedrick et al., 2018).

Two databases were also considered to assess the influence of the soil texture on the hydrological response. These databases

were used not only during the event, but also over the one-year spin-up period before the event, and therefore also led to differences in terms of soil moisture ICs. The flood discharge volume was slightly impacted at most of the hydrometric stations due to different estimates of initial soil saturation resulting from differences in soil texture. Soil texture also impacted the streamflow timing at the beginning of the flood at individual stations in agreement with Anquetin et al. (2010) who showed that the soil texture can influence the first phase of floods until soil saturation is reached. However, overall,

these differences in simulated flood discharge volume were less important than the differences found with the different atmospheric forcings and snow ICs. Similarly, Lovat et al. (2019) found that the soil database has less influence than the spatial resolution of the hydrological model for flash flood modelling in snow-free Mediterranean basins with complex topography. In their study, a higher resolution of the hydrological model was associated with an improved representation of topography, soil and land data, with a fixed resolution of atmospheric forcings. This is the opposite as the present study

which assessed the impact of the resolution of the atmospheric forcings, using a fixed resolution for the hydrological model. Therefore, further studies are needed to better assess the impact of soil database on the hydrological response of basins with complex topography in regard to the other sources of uncertainty.

GEM-Hydro presents limitations that potentially affected the hydrological responses simulated by the model. In particular, Manning's parameters were specifically adjusted for this event. This adjustment improved the estimation of peak flow

timing and magnitude for all hydrological simulations but it is not expected that it changed the relative importance of the factors governing the flood prediction skills at the different hydrometric stations. The sensitivity of the simulated hydrological response was also potentially impacted by limitations in the SVS land surface scheme for application to mountain and cold-region hydrology. SVS uses a 1-layer snowpack scheme to simulate the snow cover evolution over bare ground and low vegetation and below high-vegetation. Due to simplification in the representation of water flow through snowpack, such a scheme will be challenged by rain-on-snow events (Würzer et al. 2016) and by the occurrence of preferential flow pathways, layering and refreezing in snowpack (Leroux and Pomeroy, 2017). SVS uses a simple canopy turbulence scheme that may lead to inaccurate estimation of turbulent fluxes in dense evergreen forest (e.g. Reba et al., 2012) resulting in inaccurate estimate of snowmelt during ROS event (Marks et al., 1998). SVS also does not include the effects of slope on incoming radiation fluxes (e.g. Oliphant et al., 2003; Pomeroy et al., 2003) that influence snowmelt runoff generation in mountainous basin (DeBeer and Pomeroy, 2017; Brauchli et al., 2017). In addition, SVS does not currently simulate soil freezing and its strong impact on infiltration to frozen soils (e.g. Gray et al., 2001). Pomeroy et al. (2016b) and Teufel et al. (2017) suggested that the presence of frozen soil at high elevations and in valley bottoms modified snowmelt infiltration and contributed to the runoff generation during the June 2013 flooding event. Finally, the formulation used in SVS and other land surface schemes to simulate water movements in soil does not allow for macropore water movement which limits their ability to simulate the fast response of headwater basins during flooding events (Gaharari et al., 2019).

## 5. Conclusions

This study aims to assess the external factors governing the hydrological prediction skill of late spring floods in cold regions mountain basin. This type of flood can combine heavy precipitation from meso-scale convective systems influenced by the topography with rapid rain-on-snowmelt of late-lying snowpacks. Three main sources of uncertainty were considered in this study: (i) resolution of the atmospheric forcing, (ii) the snowpack and soil ICs and (iii) the soil texture. Toward this goal, a 1-km resolution configuration of the GEM-Hydro modelling platform was used to simulate, in hindcast mode, the hydro-meteorological conditions during the June 2013 flood in southern Alberta, Canada. The GEM atmospheric model was used in combination with the CaPA precipitation analysis system to generate several atmospheric forcings and QPE products for the flood at different horizontal resolutions (10, 2.5 and 1 km). In addition to the atmospheric forcings at different resolution, two sets of initial snow and soil conditions as well as two different soil texture databases were considered to assess their impact on hydrological simulations in the twelve main headwater basins of Southern Alberta. The model performances were evaluated in terms of flood volume discharge and peak flow estimation. Results were analyzed at each station to determine the sensitivity of model performances to the different sources of uncertainty.

The major conclusions of this study are summarized below.

- The resolution of the atmospheric driving data influenced model performances at most of the hydrometric stations, both in terms of flood discharge volume and peak flow magnitude. This is primary due to the improvements in QPE accuracy, both in terms of cumulative and extreme values of precipitation when using precipitation background from an atmospheric model at convection-permitting resolution (2.5 and 1 km). This study did not assess the uncertainty associated with precipitation forecast from deterministic or ensemble NWP systems.

- Simulated snowmelt during this ROS event was also impacted by the resolution of the atmospheric forcing in basins that were initially covered by snow. Indeed, significantly larger wind speeds were simulated at 2.5 and 1 km compared to the 10-km forcing resulting in larger turbulent fluxes and subsequent snow melt. These differences in wind speed resulted from the physical parameterizations of sub-grid topography on resolved wind speed used in the atmospheric model at 10-km grid spacing. These results highlight the importance of the wind speed forcing when estimating melt rates in ROS events and more generally the need of knowledge of the underlying atmospheric models providing driving data to hydrological models.

- Uncertainties in snow ICs at high-elevation strongly impacted the simulated hydrological response. Hydrological simulations starting with almost no snowpack at high-elevations led to a systematic underestimation of flood volume and peak flow. The insertion of SNODAS SWE data near the time of peak snow accumulation resulted in initial conditions with substantial snowpacks and coverage at high-elevation. It also modified soil moisture ICs. The use of SNOWDAS SWE led to contrasting abilities to simulate flood discharge volumes and a consistent overestimation in the mountainous part of the basins when combined with the most accurate QPE. These results highlight the importance of accurate snow ICs for the prediction of spring and early summer floods and shows the need for more accurate snow information in mountainous terrain from dedicated external snow monitoring systems.

- Soil texture databases influenced the initial soil saturation prior to the flooding event. However, simulated discharge was generally less affected by differences in the soil texture databases than differences in the resolution of the atmospheric forcing or in the snow ICs.

The results of this study demonstrate the potential benefit of integrated observation and prediction systems for mountain flood forecasting – a greater number of high elevation weather stations improved the prediction and it would have been improved further by a more accurate snow analysis product. These conclusions need to be considered with caution since only twelve headwater basins were analyzed in this study for one late-spring flood event. In particular, further studies are required to evaluate better the impact of the resolution of the atmospheric forcing on the simulated snowmelt dynamic during ROS events. Other mountainous areas with better snow monitoring systems than the Canadian Rockies could also be considered to refine the assessment about the impact of snow ICs. The on-going implementation of GEM-Hydro as a hydrological forecasting system in Western Canada and the development of a new version of SVS to better simulate the snow and frozen ground processes impacting cold regions hydrology will allow further evaluations, taking into account uncertainties associated with model processes and parameters.

**Data availability**

The CaPA and GEM data at different resolutions are publicly available on the Federated Research Data Repository (Vionnet et al., 2019). Ground station data from the ABE, CRHO and COOP networks are available through dedicated web-portals: (i) ABE: https://agriculture.alberta.ca/acis/alberta-weather-data-viewer.jsp, (ii) CRHO: http://giws.usask.ca/meta/, (iii) COOP:

https://www.ncdc.noaa.gov/data-access/land-based-station-data/land-based-datasets/cooperative-observer-network-coop.
Daily discharge data are available on the website of Canadian National Hydrological Service (https://wateroffice.ec.gc.ca/) Hourly discharge data for the flooding event were obtained through a specific request to the Canadian National Hydrological Service.

**Appendix A. Scores for objective evaluation of hydrological simulations**

Several metrics are used in the paper to compare the hydrographs simulated by GEM-Hydro in a given configuration to a reference observed hydrograph. The following notations are used:

- $N$ the total number of reference discharge data
- $Q_i^{obs}$ the hourly observed discharge for time $i$ $(i=1 \ ... \ N)$ and
- $Q_m^{obs}$ the mean value of the hourly discharge period over the full time period.

- $Q_{max}^{obs}$ the amplitude of the observed peak flow
- $Q_i^{ref}$ the hourly simulated discharge for time $i$ $(i=1 \ ... \ N)$
- $Q_{max}^{sim}$ the amplitude of the simulated peak flow

The difference of amplitude of the peak flow (%) is defined as:

$$\Delta Q_{max} = 100.\frac{Q_{max}^{sim} - Q_{max}^{obs}}{Q_{max}^{obs}}$$

A negative (positive) $\Delta Q_{max}$ value denotes an under- (over-)estimatation of the peak flow amplitude. The percent bias (*PBIAS*) was then used to assess the simulation's overall water budget fit during the flooding event. It is defined as:

$$PBIAS = \ 100.\frac{\sum_{i=1}^{N}(Q_i^{sim} - Q_i^{obs})}{\sum_{i=1}^{N} Q_i^{obs}}$$

A negative (positive) *PBIAS* value denotes a general tendency to under- (over-)estimate streamflow (discharge) volume during the flooding event. Finally, the Nash-Sutcliffe efficiency (*NSE,* Nash and Sutcliffe, 1990) was considered. NSE is a

normalized statistic that indicates the overall fits between a simulated and an observed hydrographs. NSE is adimensional with the value of 1 meaning a perfect match between the observation and the simulation. It is written as:

$$NSE = 1 - \frac{\sum_{i=1}^{N}\left(Q_i^{sim} - Q_i^{obs}\right)^2}{\sum_{i=1}^{N}\left(Q_i^{sim} - Q_m^{obs}\right)^2}$$

## Appendix B: Sensitivity metrics

Three sources of uncertainties were considered in this study: (i) the resolution of the atmospheric forcing (ii) the initial snow conditions and (iii) the soil texture database. The sensitivity of hydrological responses to these sources of uncertainties was quantified for each score (see Appendix A) and each hydrometric station listed in Table 1. In the following, $S_{i,f,s,t}$ represents a given score at station $i$ for a given forcing $f$, a given initial snow condition $s$ and a given soil texture database $t$. $i$ ranges from 1 to 12 whereas $f$, $s$ and $t$ have 2 possible values each. For the atmospheric forcing, the resolutions of 10 km and 1 km were considered. $\bar{S}_{i,f}$ represents the median of the score for station $i$ and forcing $f$ for all possible combinations of $s$ and $t$ (4 in total). The same definitions were used for $\bar{S}_{i,s}$ and $\bar{S}_{i,t}$. Finally, the sensitivity to the atmospheric forcing for a given station and a given score was defined as: $S_f = \left| \bar{S}_{i,f=1} - \bar{S}_{i,f=2} \right|$. The same definition were used for $S_n$ and $S_t$. The sensitivity metrics have the same units as the score considered.

## Authors contribution

VV, VF and JP designed the study. VV ran the numerical experiments with GEM, CaPA, and GEM-Hydro and was responsible for the preparation of the manuscript. GR designed the CaPA experiments. VF, NG and EG contributed to the preparation of the experiments with GEM-Hydro and the analysis of the results. MA provided the latest version of the land-surface scheme SVS and contributed to the preparation of the SVS configuration used in this paper. All authors contributed to the preparation of the manuscript.

## Competing interest

The authors declare that they have no conflict of interest.

## Acknowledgements

This study was supported by the Global Water Future programme funded by the Canada First Research Excellence Fund. Support from the Canada Research Chairs, NSERC, Alberta Innovates, Canada Foundation for Innovation and the Canada Excellence Research Chair in Water Security is gratefully acknowledged. Milena Dimitrijevic (ECCC), Manon Faucher (ECCC) and Dorothy Durnford (ECCC) are thanked for their helpful contribution with GEM and GEM-Hydro. Special thanks to Nic Wayand (Centre for Hydrology, USask) and Samantha Hussey (ECCC) for providing precipitation, snow and discharge data. We also thank three anonymous reviewers for their suggestions to improve the quality of the manuscript.

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

**Table 1. Characteristics of the hydrometric stations used for model evaluation and mean soil texture fraction (top 50-cm) per basin from the GSDE and SoilGrid (SLG) databases.**

| Station Code | Station Name | Main River | Drainage Area (km$^2$) | Sand (%) GSDE | Sand (%) SLG | Clay (%) GSDE | Clay (%) SLG |
|---|---|---|---|---|---|---|---|
| 05CA004 | Red Deer River Above Panther River | Red Deer River | 941 | 44.5 | 48.3 | 13.0 | 14.1 |
| 05CA009 | Red Deer River Below Burnt Timber Creek | Red Deer River | 2246 | 42.6 | 47.6 | 15.1 | 15.8 |
| 05BB001 | Bow River at Banff | Bow River | 2210 | 43.4 | 48.3 | 13.6 | 12.2 |
| 05BG010 | Ghost River Above Waiparous Creek | Bow River | 484 | 47.5 | 47.8 | 19.4 | 15.7 |
| 05BH015 | Jumpingpound Creek at Township Road | Bow River | 474 | 27.1 | 44.4 | 29.8 | 19.3 |
| 05BJ010 | Elbow River at Sarcee Bridge | Bow River | 1189 | 35.1 | 45.7 | 26.1 | 17.7 |
| 05BL012 | Sheep River at Okotoks | Bow River | 1494 | 31.1 | 42.4 | 29.0 | 20.1 |
| 05BL019 | Highwood River at Diebel's Ranch | Bow River | 774 | 44.3 | 46.9 | 18.6 | 14.6 |
| 05AB041 | Willow Creek at Oxly Ranch | Oldman River | 833 | 34.2 | 39.9 | 30.8 | 21.1 |
| 05AA035 | Oldman River at Range Road | Oldman River | 1835 | 39.0 | 44.9 | 23.7 | 16.8 |
| 05AA008 | Crowsnest River near Franck | Oldman River | 403 | 47.7 | 45.2 | 17.1 | 15.3 |
| 05AA022 | Castle River near Beavers Mines | Oldman River | 820 | 41.3 | 44.5 | 21.2 | 15.3 |

5 **Table 2. Summary of GEM model configuration and references for the different physical parameterizations used in the model. The centre of the 3 domains is located at 51.44 N and -114.0 W.**

| Category | Model | GEM 10 km | GEM 2.5 km | GEM 1 km |
|---|---|---|---|---|
| Horizontal | Grid spacing (km) | 10.0 | 2.5 | 1.0 |
| | Grid size | 360 × 360 | 540 × 540 | 768 × 768 |
| | Domain (km) | 3600 × 3600 | 1350 × 1350 | 768 × 768 |
| Vertical | Levels numbers | 80 | 62 | 62 |
| | Lowest levels (m) | 40 m (momentum), 20 m (thermodynamics) | | |
| Time | Time step (s) | 240 | 60 | 30 |
| | Update interval (min) | 60 | 12 | 12 |
| Physics | Land Surface | ISBA (Belair et al., 2003a,b) | | |
| | Planet Boundary Layer | MoisTKE (Belair et al., 2005) | | |
| | Deep convection | Kain and Fritsh (1990, 1993)[a] | | None |
| | Shallow convection | Kuo-transient scheme (Belair et al., 2005) | | |
| | Cloud Microphysics | Sundqvist (1978) | P3 (Morrison and Milbrandt, 2015) | |

[a]parameter settings for KF scheme depend on model resolution

**Table 3. Geophysical databases used to generate surface fields for the GEM experiments at different resolutions.**

| Field | Database | Resolution | Experiments |
|---|---|---|---|
| Orography | U.S. Geological Survey Global 30 arc s elevation dataset (https://lta.cr.usgs.gov/GTOPO30) | 30 arc s | GEM 10 km |
| | Canadian Digital Elevation Data (CDED) over Canada (http://ftp.geogratis.gc.ca/pub/nrcan_rncan/elevation/cdem_mnec/doc/CDEM_product_specs.pdf) | 25 m | GEM 2.5 and 1 km |
| | Shuttle Radar Topography Mission (SRTM) over USA (http://srtm.csi.cgiar.org/) | 90 m | GEM 2.5 and 1 km |
| Soil texture | Global Soil Dataset for Use in Earth System Models (GSDE) (http://globalchange.bnu.edu.cn/research/soilw) | 30 arc s | All |
| Land Cover | ESA CCI LC Global Map (European Space Agency Climate Change Initiative Land Cover; https://www.esa-landcover-cci.org/) | 300 m | All |

**Table 4. Main characteristics of hydrological simulations carried out with GEM-Hydro.**

| Experiment | Atmospheric forcing | Precipitation analysis | Snow insertion | Soil texture |
|---|---|---|---|---|
| 10km_OPL_GSD | GEM 10 km | CaPA 10 km | OPL | GSDE |
| 10km_OPL_SLG | GEM 10 km | CaPA 10 km | OPL | SoilGrid |
| 10km_SND_GSD | GEM 10 km | CaPA 10 km | SND | GSDE |
| 10km_SND_SLG | GEM 10 km | CaPA 10 km | SND | SoilGrid |
| 2.5km_OPL_GSD | GEM 2.5 km | CaPA 2.5 km | OPL | GSDE |
| 2.5km_OPL_SLG | GEM 2.5 km | CaPA 2.5 km | OPL | SoilGrid |
| 2.5km_SND_GSD | GEM 2.5 km | CaPA 2.5 km | SND | GSDE |
| 2.5km_SND_SLG | GEM 2.5 km | CaPA 2.5 km | SND | SoilGrid |
| 1.0km_OPL_GSD | GEM 1.0 km | CaPA 1.0 km | OPL | GSDE |
| 1.0km_OPL_SLG | GEM 1.0 km | CaPA 1.0 km | OPL | SoilGrid |
| 1.0km_SND_GSD | GEM 1.0 km | CaPA 1.0 km | SND | GSDE |
| 1.0km_SND_SLG | GEM 1.0 km | CaPA 1.0 km | SND | SoilGrid |

**Table 5. Errors metrics (Bias in mm, Root Mean Square Error (RMSE) in mm and Pearson correlation coefficient, R) for accumulated precipitation estimated by the leave-one out method for 3 CaPA experiments. Three river basins are considered as well as three time periods: a convective period (19 June 12 UTC to 20 June 12 UTC), a stratiform period (20 June 12 UTC to 22 June 12 UTC) and the whole period. Avg. Obs. represents the averaged observed cumulated precipitation (in mm).**

| Catchment | Period | Avg. Obs. | CaPA 10 km | | | Capa 2.5 km | | | CaPA 1 km | | |
|---|---|---|---|---|---|---|---|---|---|---|---|
| | | | Bias | RMSE | R | Bias | RMSE | R | Bias | RMSE | R |
| Red Deer River | Convective | 25 | -1.0 | 10.0 | 0.91 | -1.1 | 11.3 | 0.87 | **-0.5** | **8.4** | **0.93** |
| | Stratiform | 63 | -4.9 | **17.6** | **0.90** | -4.2 | 18.9 | 0.87 | **-1.0** | 25.4 | 0.79 |
| | All | 88 | -5.9 | **24.4** | **0.92** | -5.3 | 26.5 | 0.89 | **-1.5** | 30.6 | 0.87 |
| Bow River | Convective | 75 | -4.8 | 20.5 | 0.89 | **-0.4** | 19.4 | 0.89 | **-0.4** | **15.3** | **0.93** |
| | Stratiform | 95 | **-2.4** | 23.4 | 0.84 | 5.1 | **22.4** | **0.87** | 4.1 | 23.6 | 0.86 |
| | All | 170 | -7.2 | 38.2 | 0.88 | 4.7 | **33.2** | **0.91** | **3.7** | 35.0 | 0.90 |
| Oldman River | Convective | 80 | -5.4 | 26.8 | 0.65 | **-0.4** | 26.7 | 0.62 | 2.9 | **19.9** | **0.8** |
| | Stratiform | 20 | **-0.5** | **5.1** | **0.91** | 1.0 | 6.2 | 0.89 | 2.3 | 7.2 | 0.85 |
| | All | 100 | -5.9 | 27.9 | 0.75 | **0.5** | 26.5 | 0.75 | 5.3 | **21.5** | **0.85** |
| All | Convective | 65 | -4.1 | 21.2 | 0.87 | **-0.6** | 20.9 | 0.87 | 0.7 | **15.9** | **0.92** |
| | Stratiform | 61 | -2.3 | 17.8 | **0.93** | 1.4 | **17.4** | **0.93** | 2.3 | 20.0 | 0.92 |
| | All | 126 | -6.4 | 31.8 | 0.91 | **0.8** | **29.4** | **0.92** | 3.0 | 29.7 | **0.92** |

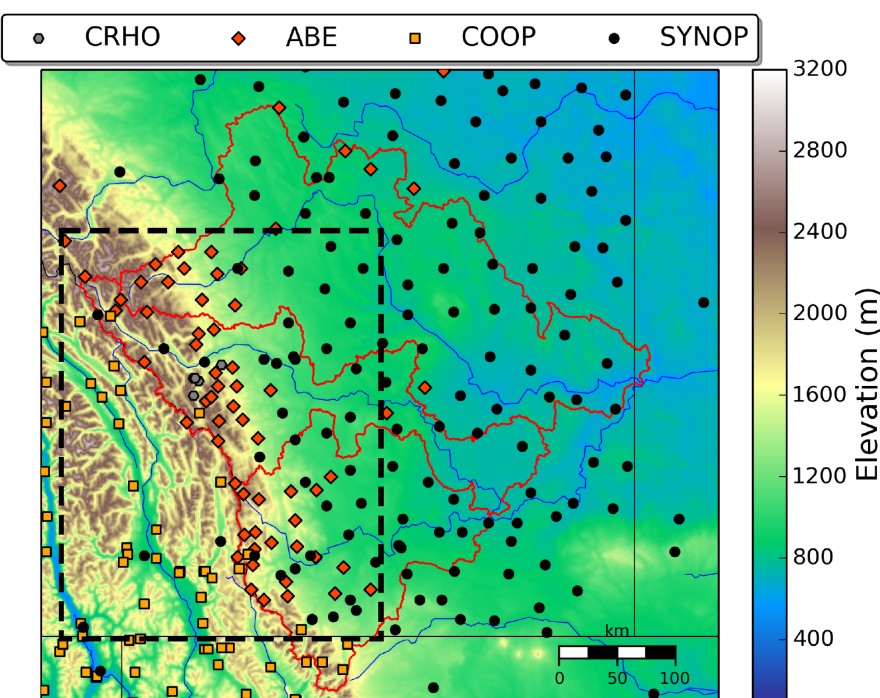

**Figure 1: Map of southern Alberta showing the topography at 1-km resolution and locations of precipitation stations from different networks: USask's Canadian Rockies Hydrological Observatory (CRHO), Alberta Environment and Parks (ABE), the American meteorological cooperative network (COOP) and automatic synoptic stations (SYNOP) maintained by Environment and Climate Change Canada and other organizations. The red lines represent the Red Deer (north), Bow (middle) and Oldman (south) river basins. The dashed black box highlights the location of the region shown on Fig. 2.**

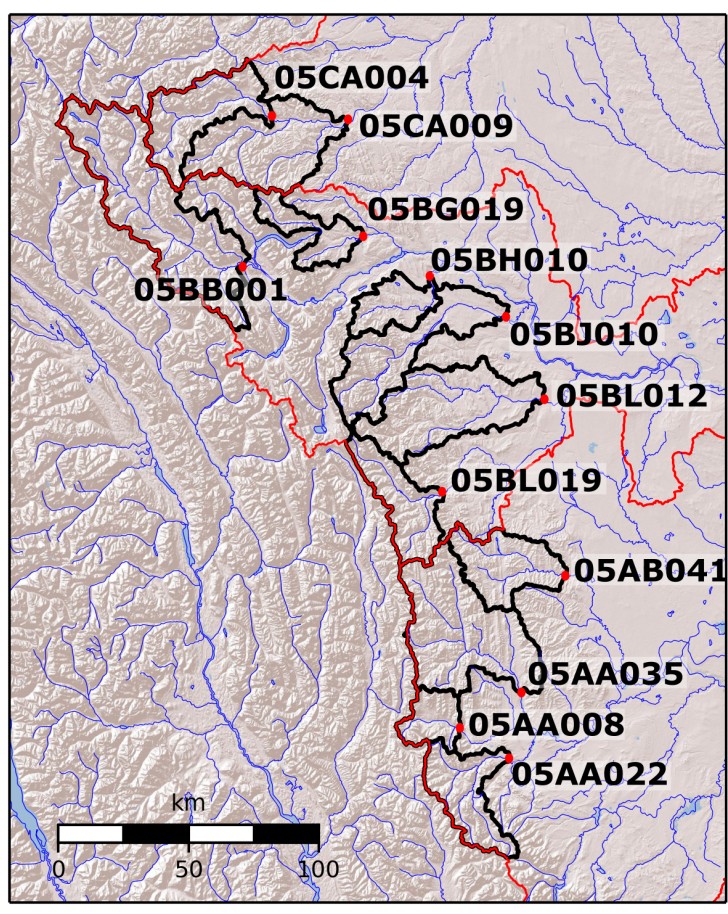

5    **Figure 2: Map showing the locations of the river gauges used in this study and the corresponding river basins (black lines). The red lines represent the Red Deer (north), Bow (middle) and Oldman (south) River basins. The general location of this area is shown on Fig. 1.**

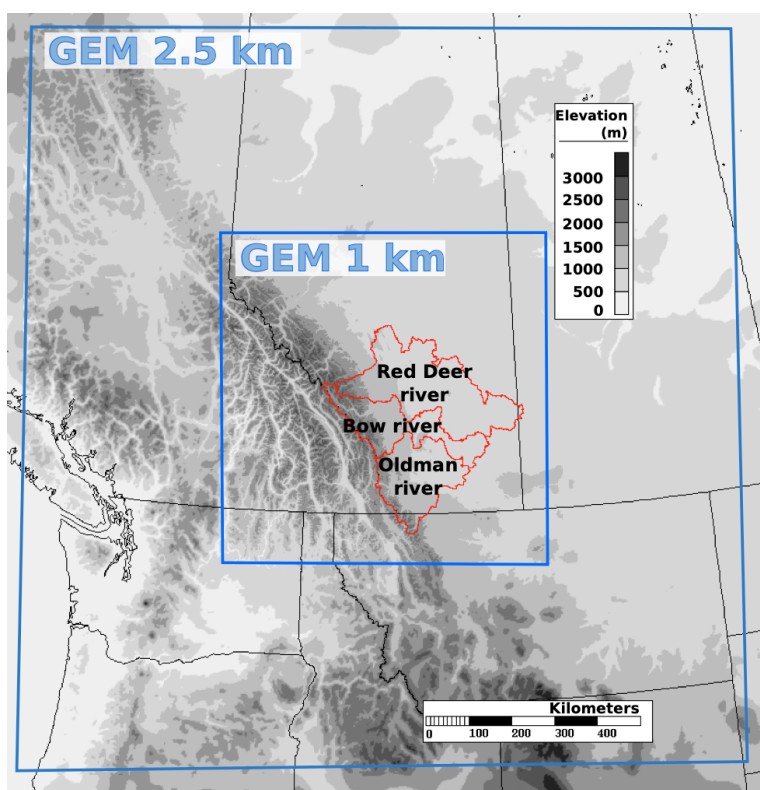

**Figure 3. Location of the 2.5-km and 1-km GEM computational grids with elevation shown in shades of grey. The red lines represent the boundaries of the main river basins in southern Alberta.**

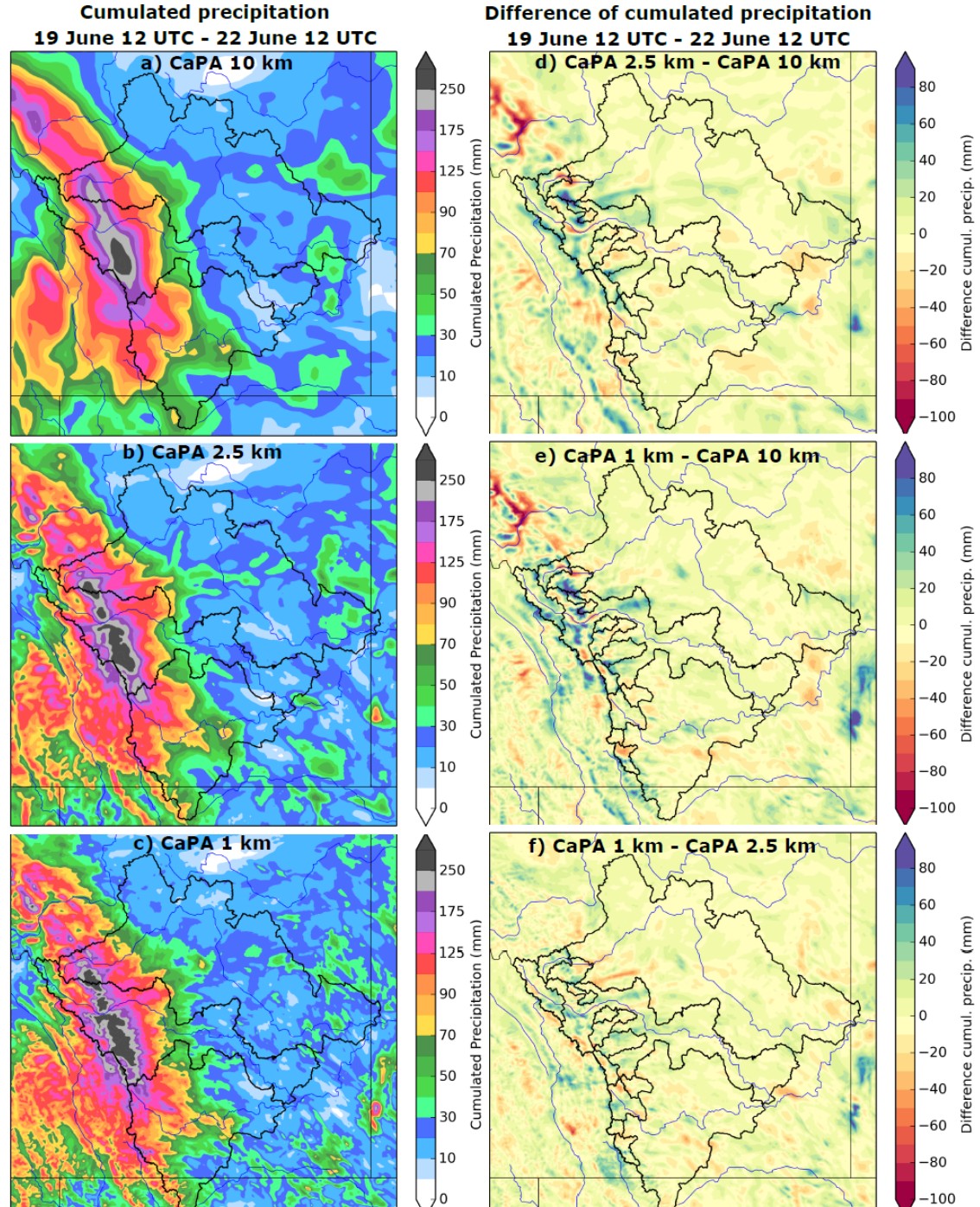

**Figure 4. (Left) Accumulated precipitation from 19 June 12 UTC to 22 June 12 UTC obtained with CaPA at (a) 10 km, (b) 2.5 km and (c) 1 km. (Right) Differences between the CaPA experiments. The black lines on the left figures represent the Bow, Oldman and Red Deer river basins (see Fig. 1 for general location). In addition, the right figures show the locations of the 12 basins considered for the evaluation of hydrological simulations (Fig. 2).**

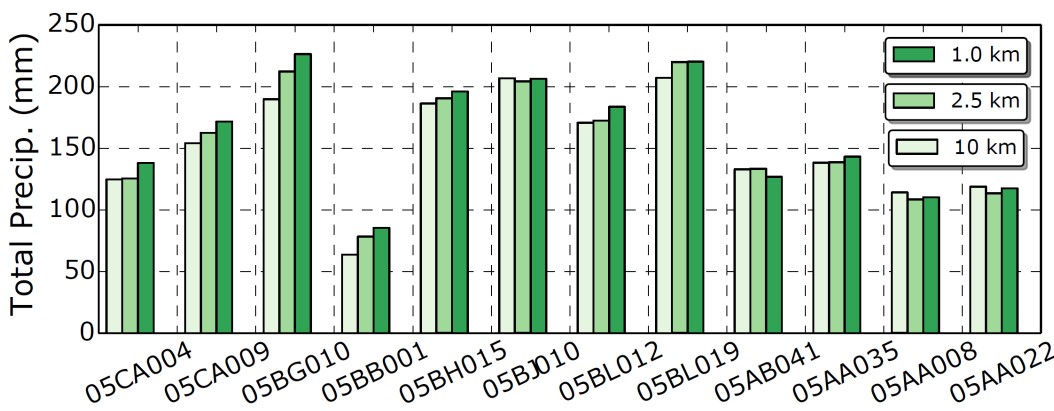

**Figure 5. Accumulated precipitation from 19 June 12 UTC to 22 June 12 UTC averaged over the 12 basins shown on Fig 2. for the CaPA experiments at different resolutions.**

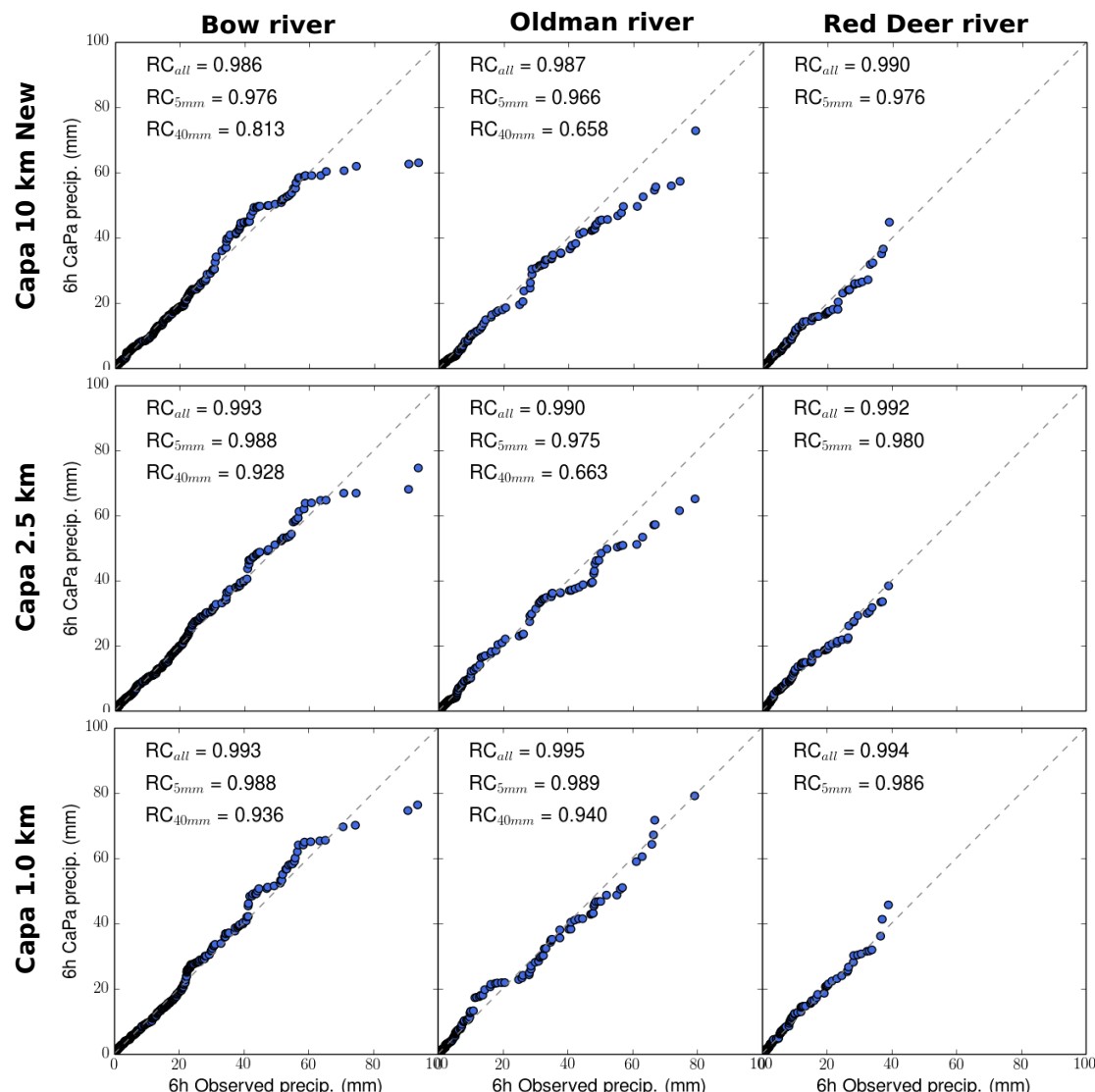

**Figure 6. Quantile-quantile (QQ) plots between 6-h observed precipitation at CRHO, COOP and ABE stations and 6-h precipitation estimated by the leave-one out method for the different analysis: CaPA 10 km New (Top), CaPA 2.5 km (Middle) and CaPA 1 km (Bottom). The QQ plots are computed from 19 June 12 UTC to 22 June 12 UTC for the 3 main river basins in the region: Bow, Oldman and Red Deer. RC represents the concordance correlation coefficient (Lawrence and Lin, 1989) computed for different categories of 6-h precipitation: (i) all, (ii) above 5 mm and (iii) above 40 mm (see text for more details).**

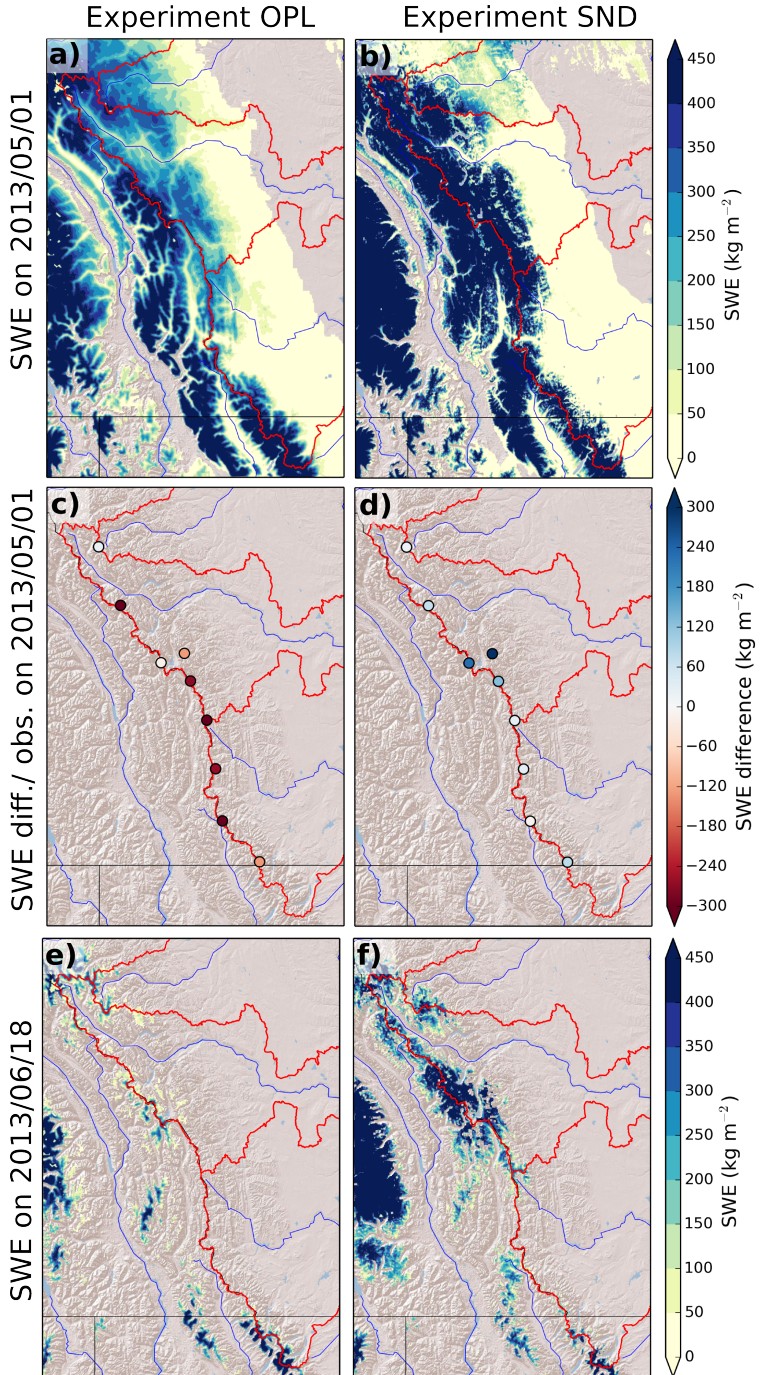

**Figure7. (Top) Snow water equivalent (SWE) on 1 May 2013 for two categories of experiments: Open Loop (*OPL*) without insertion of SNODAS and (b) with insertion of SNODAS (*SND*). (Middle) SWE difference with observation from ABE snow pillows on 1 May 2013 for *OPL* (c) and *SND* (d). (Bottom) SWE on 18 June 2013 for *OPL* (e) and *SND* (f). The red lines on each figure represent the boundaries of the Bow, Oldman and Red Deer river basins (see Fig. 1 for general location).**

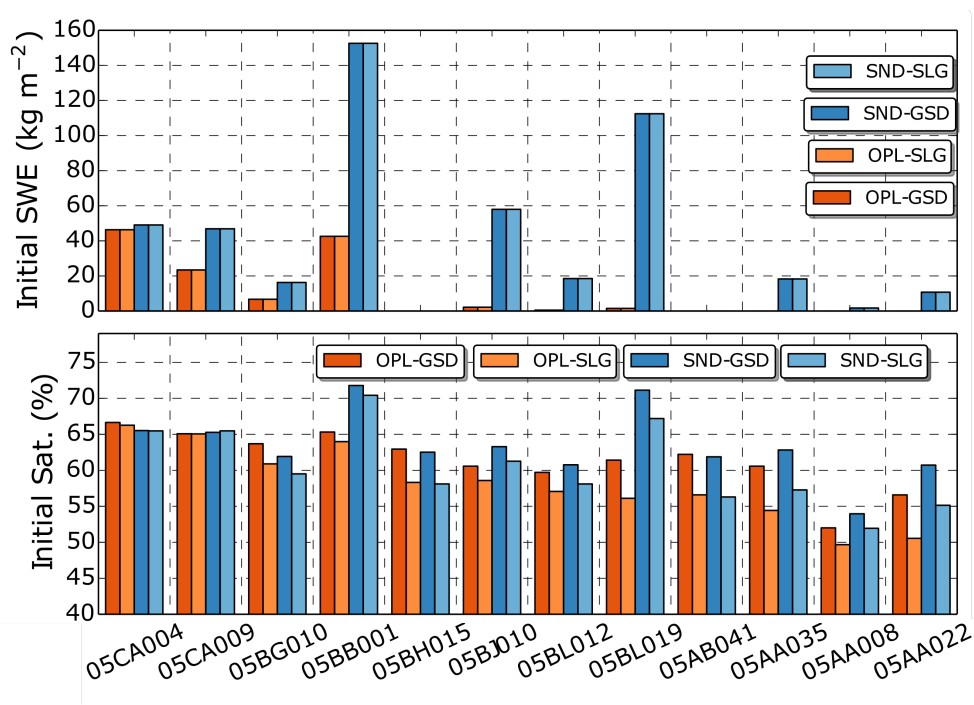

**Figure 8: (Top) Basin-averaged initial SWE and (Bottom) basin-averaged initial soil saturation for the top 50 cm of soil column on 18 June 2013 12 UTC for the 12 basins shown on Fig 2. Results from four GEM-Hydro simulations providing initial conditions are shown. Their characteristics are given in Sect. 2.3.3.**

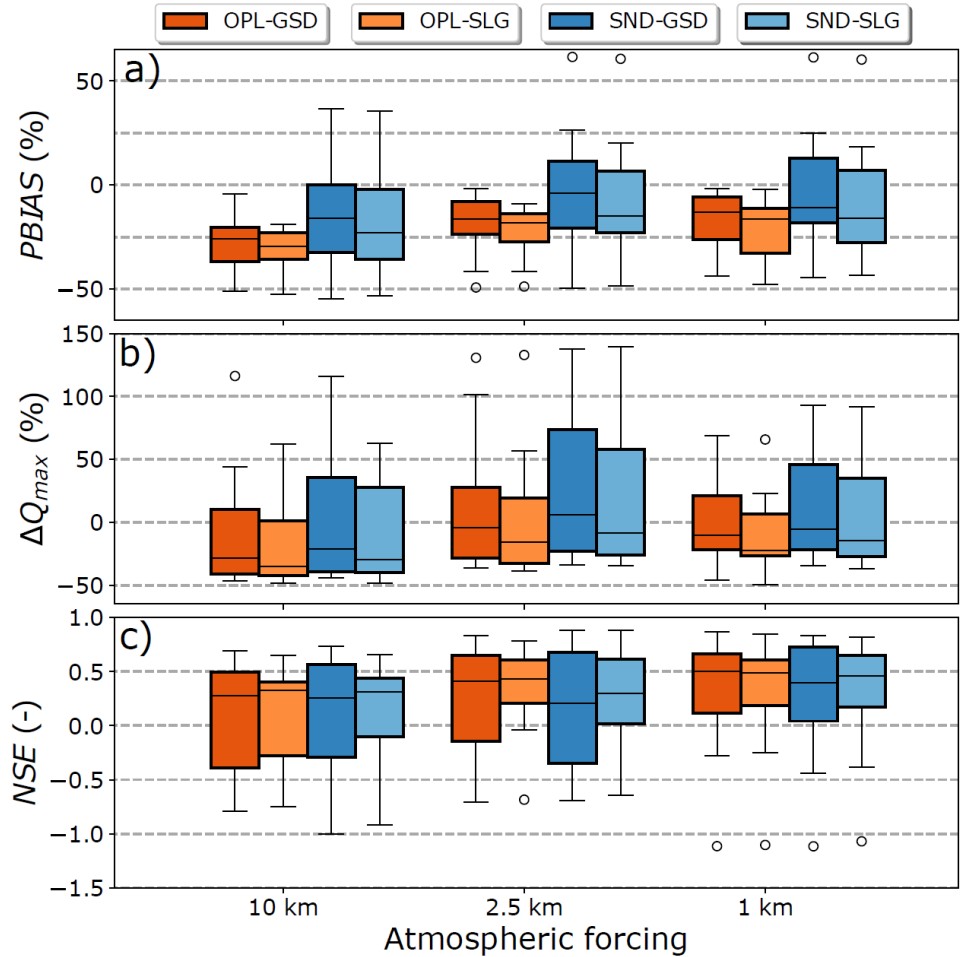

**Figure 9: Distributions of error metrics for the different GEM-Hydro simulations of the June 2013 flood as a function of the resolution of the atmospheric forcing: (a) percent bias, (b) relative difference of peak flow amplitude and (c) Nash-Sutcliffe efficiency. The colours of the boxplots depends on the different soil and snow ICS and on the soil texture database. Error metrics were computed from 20 June 0 UTC to 25 June 0 UTC. Their definition is given in Appendix A.**

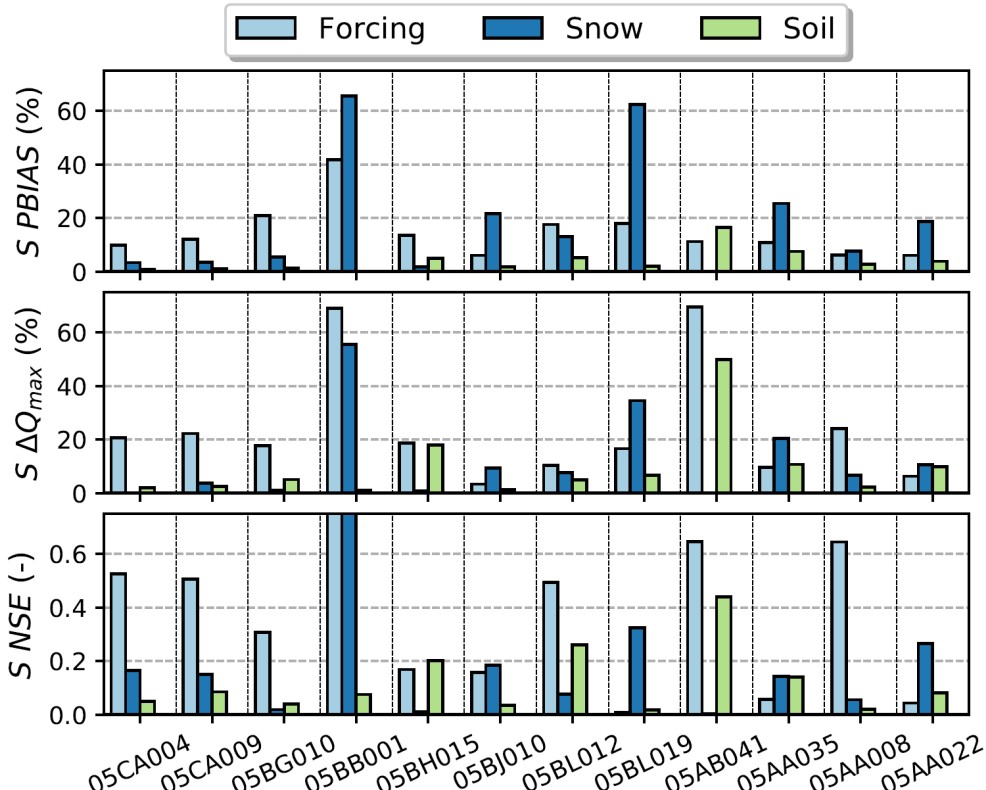

**Figure 10: Sensitivity of the error metrics to the three sources of uncertainty considered in this study. The computation of the sensitivity metrics is described in Appendix B. The atmospheric forcings at 10 and 1 km were considered to obtain the sensitivity to the resolution of the atmospheric forcing. Note that the maximal sensitivity for station 05BB001 for *NSE* is not displayed on the bottom plot.**

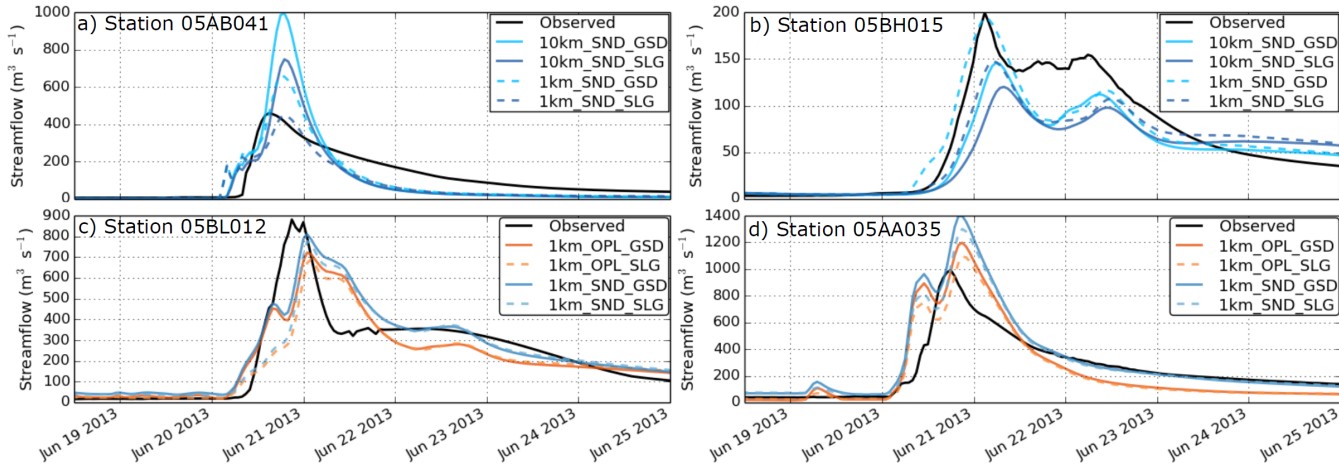

**Figure 11: Observed and simulated streamflow at four hydrometric stations. Graphs (a) and (b) illustrate the combined sensitivity to the resolution of the atmospheric forcing and the soil database whereas graphs (c) and (d) illustrate the sensitivity to the soil database and initial snow conditions. Note the different *y* axis on each graph.**

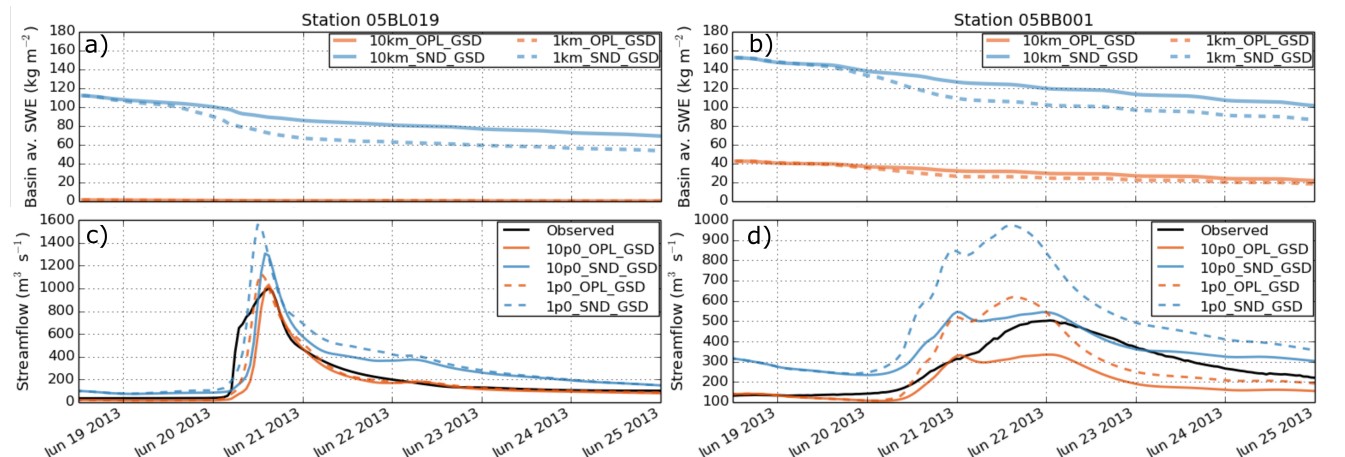

**Figure 12: Snow and streamflow simulations at two hydrological stations: station 05BL019 (Left) and station 05BB001 (Right). (a) & (b): Basin-averaged SWE for four GEM-Hydro simulations; (c) & (d) Observed and simulated streamflow. Note the different *y* axis on each graph.**

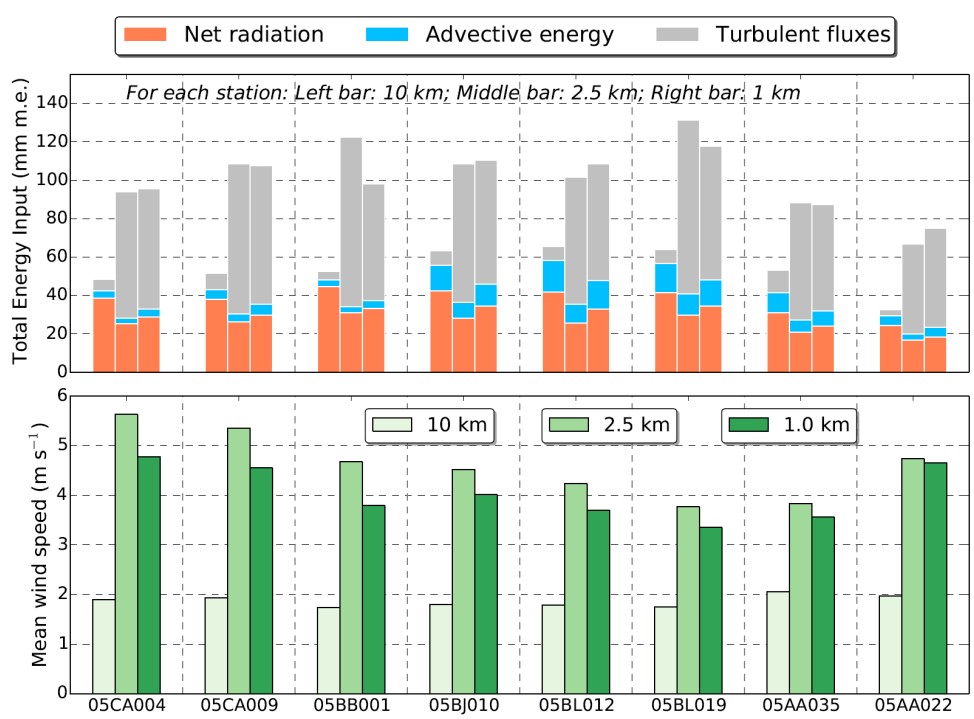

**Figure 13: (Top) Contribution of the different energy balance terms to the total averaged energy input to the snowpack (expressed in millimetre melt equivalent) during the flooding event simulated using atmospheric forcing at different resolutions (Left: 10 km; Middle: 2.5 km; Right: 1 km) and spatially averaged over all the snow-covered pixels of different sub-basins. Hydrological simulations in configuration *SND_GSD* were considered. (Bottom) Mean wind speed simulated using atmospheric forcing at different resolutions averaged over all the snow-covered pixels of different sub-basins and the whole flooding event.**