# Peer review of "Assessing the factors governing the ability to predict late-spring flooding in cold regions mountain basins"

_Hydrology and Earth System Sciences, 2019_

## Referee Comment (RC1) · Anonymous Referee #1 · 26 Jun 2019

General comments: This paper investigates the current modelling techniques available for flood forecasting and the associated required inputs. The evaluation was performed on a well documented significant flood in Canada in 2013, which resulted in large losses and damages in the affected area. The topography of the study area is very complex and addresses the limitation of the current flood prediction system used at ECCC. The material presented would benefit the modelling community working on refining and adjusting models to better predict floods in similar conditions and in complex terrain. In general the paper well written and relatively clear.

Specific comments: On page 8, OPL and SND are introduced, its not clear what they

mean. On page 9 and 10 section 3.2 , could the authors give some more details for the cumulative underestimation of the stations. Also some possible explanation why going from a 10km Capa grid to 2.5km grid provided better results. Why does the Red Deer basin have a decrease in cumulative precipitation. How much does convection potentially play in these underestimates. On page 12, it seems that the river routing is a large unknown, in these simulations, are there ways to determine the actual routing for this event and apply it to the simulation. Is this even feasible? What is the frequency of reporting of the precipitation stations?

Technical corrections: Page,3 line 20. Use "Another objective" rather than "Another main objective" Page 8, what does OPL and SND stand for. Page 10, line 1. Remove "in", should read foothills of the Red Deer Basin. Page 10, line 11. "in particular in", remove the second "in". Page 10, line 20, use analyzed instead of analysed. Page 11, line 17. "with to that " is awkward. Page 12, line 14. Remove space between 59 and %. Page 12, line 30 . Remove space between number signs. Page 12, line 33. Comma after however. Page 25, line29. "Radar data was" Page 14, line 12. Theses should be These. Page 15, line 13, similarly should be similar. Page 15, line 26. Calibration of these parameters. Page 15 Lines 1-10. This sentence is very long and the entire section doe snot flow well. Consider rewriting to make the flow better. Page 17 Line15. remove "are" to read forecasting systems can be expected. Also remove SVS from the sentence.

---

## Referee Comment (RC2) · Anonymous Referee #2 · 9 Aug 2019

Dear authors,

Your study is a case study (a nice one, I admit). I am personally not very interested in such studies, where actually only the data owner and model operators learn something about their specific data chain in a single event in their region. With some adjustments it might fit in NHESS. For HESS I see too limited added information for the community.

Where is the true novelty? You use well citable tools and data for a specific event. Their present combination might be novel, but surely not original (different resolution, different initial conditions, different density of stations).

Where is the solid statistic?

[Figure]

See the attached document for additional comments.

Please also note the supplement to this comment:
https://www.hydrol-earth-syst-sci-discuss.net/hess-2019-152/hess-2019-152-RC2-supplement.pdf

---

## Referee Comment (RC3) · Anonymous Referee #3 · 14 Aug 2019

This paper examined the capability of a hydrological model (GEM-Hydro) in simulating the June 2013 flood event in Alberta, Canada. In particular, three sub-basins in the Bow River basin were selected to assess the impacts of spatial resolution, precipitation gauge density, and initial snow conditions on model ability in reproducing the flow volumes. Also, the model sensitivity to Manning coefficients in capturing the peak flow was investigated.

General Comments:

The objective of this paper is straightforward and this paper is well-written, easy to follow and well-structured. However, the creditability of the study has been reduced

because the study is highly localized and reads more like a report of an application of a hydrological model to a specific flood event. There are little knowledge gain for the community. In general, two major concerns are needed to be addressed in this paper:

**1) Novelty of the study**

While the study would be a great contribution to the development of a Canadian hydrological forecasting system, it might not be novel enough as a scientific contribution for the international community. The effects of different spatial resolutions on model simulations have been previously and heavily studied in different hydrological models and it is well expected that finer resolution could provide better simulations because of its ability in capturing the fine-scale hydro-meteorological processes. It is also expected that the inclusion of additional information (e.g. increasing network density, inclusion of SWE information) would improve model performance because of the data-driven nature of the sophisticated models nowadays. Lastly, the Manning coefficients (both channel and floodplain) are well known to be one of the most sensitive routing parameters in any hydrological/hydraulic models. Adjusting such parameter will definitely improve the model ability in matching the peak flow timing and magnitude.

Furthermore, the Discussion section of the study did not provide innovative insights on modelling of extreme flood events. Although the Discussion section was well-written and fully supported by references, the major findings of this study were merely a confirmation of what had been shown in many previous studies (as repeatedly mentioned by the authors). Therefore, there is a lack of true novelty and scientific contribution in this study. The authors should vigorously address this critical issue by providing a better discussion on what new knowledge and information the international community could learn from this study.

2) Lack of connection with other worldwide extreme flood events

The Introduction section was again well-written and the rationale of the study was well presented, however, it was highly focused on the description of the June 2013 flood
event and the previous works that were related to the 2013 flooding. The literature review did not discuss any research related to other worldwide flood events and any modelling works that address the current challenges of the modelling community in dealing with extreme flood induced by rain-on-snow events. This makes the study highly localized and event specific. The authors should provide a boarder discussion on similar research conducted in other regions, modelling strategy used in simulating such kind of flood events, and the research gaps this study could fill in for advancing the knowledge of the community.

Specific Comments:

P1L28-29: How could the results of this study guide the development of the hydrological forecasting system worldwide?

P4L10: Could the authors comment on the consistency of these four different networks?

P5L3-4: What are the drainage areas of these river basins?

P5L6-7: Could the authors provide the basic information of these 10 stations (e.g. station name, drainage area, name of tributary)?

P10L10-12: It would be better to provide the performance measures (Bias, RMSE, R2) of each basin in a table.

P10L13-14: Could the authors explain how the negative bias was removed by increasing the spatial resolution to 2.5km and 1km? Figures 7c) and 7d) do not show an obvious removal of the negative bias, instead, the points are still scattered above and below the 1:1 line.

P10L15-16: Could the authors explain why there were overestimations of cumulative precipitation using CaPA 1.0km?

P11L1-2: While the analysis on the effects of spatial resolution, gauge network den-
sity, and initial snow condition were conducted across the Oldman, Bow, and Red Deer River basins, the hydrological simulations were evaluated on Jumpingpound Creek, Elbow River and Highwood River, which are all located within the Bow River basin. I think such selection of hydrological simulations could not fully reveal the impacts of those factors controlling the flood dynamics. A better experimental design could be selecting one or two headwater sub-basins from each river basin (Oldman, Bow, and Red Deer). The results could potentially provide more information than the current setting (10 stations all within the Bow River basin) especially when different responses were witnessed after the inclusion of additional information (precipitation and/or SWE) across the three basins (e.g. consistently underestimation of cumulative precipitation plus overestimation of SWE in the Bow River, a mixture of over-and under-estimation of cumulative precipitation in the Red Deer, and underestimation of cumulative precipitation plus fairly accurate SWE estimates in the Oldman River). I wonder why the authors only focused on hydrological simulations within the Bow River basin.

P15L30-P16L15: This study did not examine the model structure and the process representation in the model at all. Providing a list of potential reasons that might affect the model performance here becomes irrelevant unless concrete proof and result analysis are given to show the underlying causal relationship. This sub-section should be better re-written.

P17L19-20: What about the ground station data from the ABE, CHRO, and COOP networks? Are they publicly available to the international readers?

Remarks:

P2L4-5, 10, and 12: could the authors check this reference please? Is this reference the correct one that the authors intended to use? I guess it should be Pomeroy et al. 2016a. Please correct me if I am wrong.

P8L18: should be "similar to" not "similarly to"
P8L24 and 27: please spell out the full name first before using the abbreviation

P8L33: missing full stop after "(18 June 2013)"

P12L5: should it be "Fig. 10c, f and Fig. 11c, f" instead of "Fig. 10d, f and Fig. 11d,f"?

P12L23: better use "hypothesis" instead of "assumption"

P17L15: delete "are"

Figures 3 and 4: better use "boundaries" instead of "limits"

Figures 10-12: it is a bit misleading to use filled area to represent the results because essentially there are only two simulations using initial conditions from OPL and SND. They might not necessarily represent the upper and lower limits of the model performance, especially when more different initial conditions are used.

**HESSD**

---

## Author Comment (AC1) · 1 Oct 2019

Answer to Reviewer 1 HESS-2019-152

We thank Reviewer 1 for his comments. We provide here our responses to his comments, including the plan to revise the manuscript in response to reviewer comments. The original review comments are in normal black font while our answers appear in blue font.

**General comments:**

This paper investigates the current modelling techniques available for flood forecasting and the associated required inputs. The evaluation was performed on a well documented significant flood in Canada in 2013, which resulted in large losses and damages in the affected area. The topography of the study area is very complex and addresses the limitation of the current flood prediction system used at ECCC. The material presented would benefit the modelling community working on refining and adjusting models to better predict floods in similar conditions and in complex terrain. In general the paper well written and relatively clear.

**Specific comments:**

On page 8, OPL and SND are introduced, its not clear what they mean.

"OPL" refers to the Open Loop GEM-Hydro simulation from June 2012 to June 2013 used to derive the initial land surface conditions. "SND" refers to the simulation with insertion of SNODAS SWE on 1$^{st}$ May 2013 near peak snow accumulation. Experiment SND provides another estimation of land surface conditions prior to the flooding event. These acronyms will be more clearly defined in the revised manuscript.

On page 9 and 10 section 3.2, could the authors give some more details for the cumulative underestimation of the stations.

The revised manuscript will include a more detailed evaluation of the different CaPA experiments, both spatially and temporally. As suggested by Reviewer 3, a table containing the performances metrics (Bias, RMSE, R2) for each CaPA experiment and each main basin (Oldman, Bow and Red Deer Rivers) will be added. The performances metrics will be also computed for the whole event and separately for the convective and stratiform periods as defined in Kochtubajda et al. (2016) and Li et al. (2017). Following the general comment of Reviewers 2 and 3, the CaPA experiment at 10 km using the default stations (*CaPA 10km Def*) that was characterised by a large underestimation of precipitation in the mountains and the foothills will be not used anymore in the paper.

Also some possible explanation why going from a 10km Capa grid to 2.5km grid provided better results.

Improvements in the precipitation analysis from 10 km to 2.5 km resulted from the improvement in the precipitation background provided by the Canadian NWP system GEM. GEM at 2.5 km resolves explicitly part of the convection with the cloud microphysical scheme P3 whereas all the convection is parameterised in GEM at 10 km. Previous studies by Li et al. (2017) and Milrad et al. (2017) showed that atmospheric models at convection-permitting resolution performed best for this event. As detailed by Milrad et al. (2017), this is mainly due to (i) an improved representation of the orographic ascent that contributed to the magnitude of the extreme rainfall (ii) an anchoring and increasing duration of the precipitation on the Eastern side of the Rockies. These explanations will be clearly given in the revised manuscript.

Why does the Red Deer basin have a decrease in cumulative precipitation.

Fig. 6 in the initial manuscript shows a decrease in cumulative precipitation for an area of the Red Deer basin for each CaPA experiment including the additional stations compared to the analysis with the default stations (*CaPA 10km Def*). This is explained by an overestimation of precipitation in this area for *CaPA 10km Def* (Fig 5. b). The additional stations in the new precipitations analysis (see their

location on Fig. 1 in the initial manuscript) corrected this overestimation leading to a decrease of precipitation in this area. We will include a better explanation in the revised manuscript.

How much does convection potentially play in these underestimates.
We thank Reviewer 1 for this comment. Indeed, the extreme rainfall event of June 2013 in Alberta consisted of 2 stages: a convective period followed by a stratiform period (Liu et al., 2016; Kochtubajda et al. 2016; Li et al., 2017). The convection was associated with an intense lightning activity and substantial rainfall accumulation, triggering high flows in the eastern slopes the Rockies. In the revised manuscript, we will evaluate separately the performance of the different CaPA experiments during the convective and the stratiform period. As mentioned earlier, table containing the performances metrics (Bias, RMSE, R2) for each CaPA experiment and each main basin (Oldman, Bow and Red Deer Rivers) will be added. The performances metrics will be computed for the whole event and as well as for the convective and stratiform periods.

On page 12, it seems that the river routing is a large unknown, in these simulations, are there ways to determine the actual routing for this event and apply it to the simulation. Is this even feasible?
As mentioned by Reviewer 1, the river routing strongly influenced the timing of the peak flow simulated by GEM-Hydro for this extreme flood. Determining the routing parameters for this specific event is challenging since major changes occurred in the geometry of the riverbeds so that parameters based on historical data may not be suitable for this extreme event. Routing parameters could be potentially determined for this event for river sections located in between two hydrometric stations. The observed discharge at the upstream station could be used as an input of the routing scheme and a calibration of the routing parameters could be carried out using the discharge of the downstream station for model evaluation. This would still require an estimation of the water input to river in between the two stations. This method would be restricted to certain portions of the river and could not be used for the headwaters of the catchments. Therefore, we will still used in the revised manuscript a correction factor applied to the default Manning coefficient to optimise overall peak magnitude and timing.

What is the frequency of reporting of the precipitation stations?
Precipitation stations used in this study were reporting at an hourly frequency. 6-h hour cumulated precipitation have then been derived from these hourly data to be used as input of the precipitation analysis. We will mention clearly the frequency of reporting of the stations in the revised manuscript.

**Technical corrections:**

We will include the below technical comments in the revised version of the manuscript.

Page,3 line 20. Use "Another objective" rather than "Another main objective"

Page 8, what does OPL and SND stand for.

Page 10, line 1. Remove "in", should read foothills of the Red Deer Basin.

Page 10, line 11. "in particular in", remove the second "in".

Page 10, line 20, use analyzed instead of analysed. "

Page 11, line 17. "with to that " is awkward.

Page 12, line 14. Remove space between 59 and %.

Page 12, line 30 . Remove space between number signs.

Page 12, line 33. Comma after however.

Page 25, line29. "Radar data was" "

Page 14, line 12. Theses should be These.

Page 15, line 13, similarly should be similar.

Page 15, line 26. Calibration of these parameters.

Page 15 Lines 1-10. This sentence is very long and the entire section doe snot flow well. Consider rewriting to make the flow better.

Page 17 Line15. remove "are" to read forecasting systems can be expected. Also remove SVS from the sentence.

---

## Author Comment (AC2) · 1 Oct 2019

**Answer to Reviewer 2 HESS-2019-152**

We thank Reviewer 2 for his detailed comments. We provide here our responses to his comments, including the plan to revise the manuscript in response to reviewer comments. The original review comments are in normal black font while our answers appear in blue font.

**General Comments**

Dear authors,

Your study is a case study (a nice one, I admit). I am personally not very interested in such studies, where actually only the data owner and model operators learn something about their specific data chain in a single event in their region. With some adjustments it might fit in NHESS. For HESS I see too limited added information for the community.

Where is the true novelty? You use well citable tools and data for a specific event. Their present combination might be novel, but surely not original (different resolution, different initial conditions, different density of stations).

We thank Reviewer 2 for this comment. Substantial changes will be made to the initial paper to show how this case study of an extreme flood in mountainous terrain can bring a scientific contribution to the international community.

First of all, some of the results that were presented in the initial manuscript will be removed since they correspond to expected results with a lack of real novelty as detailed Reviewers 2 and 3. We will remove from the manuscript:
- the precipitation analysis at 10 km using the default stations (*CaPA 10 km Def*). This precipitation analysis used the same stations as the operational analysis at the time of the event but its quality is mediocre due the low density of stations in mountainous terrain. In the revised manuscript, we will only keep the precipitation analysis generated with all the precipitation data for each horizontal resolution (10, 2.5, 1 km). Theses different precipitation data will be presented as our best estimation of the precipitation at each horizontal resolution.
- the results corresponding to the different Manning options that were described in the section *Results*. We will describe the selection of the correction factor applied to the default values of the Manning coefficients in the section *Methods*. See below our answer to the specific comment on this question.

Removing these two components of the initial manuscript will allow us to focus the revised manuscript on the factors governing the predictions skill of extreme late-spring flood in mountainous terrain. The three **scientific questions** of the **revised manuscript** will be as follows:
1. How does the resolution of the atmospheric forcing influence the predictability of the hydrological response during extreme late-spring flood in mountainous terrain?
2. At kilometric scale, can the rain/snow partitioning from an advanced cloud microphysical scheme improve the predictability of the hydrological response during this kind of event?
3. How does the sensitivity of the hydrological response to the resolution of the atmospheric forcing and to the phase partitioning method compare to (i) the sensitivity to the initial snow conditions and (ii) the uncertainty in soil datasets?

An ensemble of hydrological simulations will be used to answer these questions:
- three atmospheric forcings will be used at 10, 2.5 and 1 km as in the initial manuscript. At each resolution, the precipitation forcing will consist of the CaPA analysis including all the stations available in the region to obtain our best estimate of precipitation at a given resolution.

- for each atmospheric forcing, the same two initial snow and soil conditions will be used as in the initial manuscript.
- finally for each atmospheric forcing and initial snow/soil conditions, two different soil database will be considered: the Global Soil Dataset for Earth System Model (GSDE, (Shangguan et al., 2014) and the Soilgrid dataset (Hengl et al., 2017). These 2 database can be considered as reference soil databases for Earth system models (Dai et al, 2019).
- for the atmospheric forcing at 1 km, for each soil dataset and initial snow conditions, three hydrological simulations will be carried out with GEM-Hydro using different phase partitioning methods (PPM): (i) a constant threshold depending on air temperature (0°C, the default value in GEM-Hydro), (ii) a more advanced PPM combining temperature and relative humidity (Harder and Pomeroy, 2013) and (iii) the direct precipitation phase from the P3 cloud microphysical model running in GEM at 1 km.

These hydrological simulations will be evaluated at a new set of 12 unregulated stations located in the headwaters of the Red Deer, Bow and Oldman River basins (see below our answer to the next general comment regarding the statistic). This new set of the stations will made our analysis less localised and will allow us to derive more general conclusions useful for the international community.

For example, we will add and discuss in the revised manuscript results on the influence of the resolution of the atmospheric forcing on the simulation of the snow melt dynamic during the rain-on-snow event at each sub-basin where snow was initially present. Figure 1 shows that the contributions of the turbulent fluxes to the total energy input to the snowpack is larger with the atmospheric forcing at 2.5 and 1 km than at 10 km. This result is consistent across all the sub-basins and results from larger wind speed at 2.5 and 1 km than at 10 km. It illustrates that the importance of the wind field downscaling on simulated turbulent fluxes and resulting snowmelt. To our knowledge, a quantification of these effects on the hydrological response during extreme floods has never been proposed in previous studies and constitutes an original contribution of our study.

[Figure]

*Figure 1: Contribution of energy balance terms to the total averaged energy input to the snowpack (expressed in millimetre melt equivalent) during the flooding event simulated using atmospheric forcing at different resolutions (Left: 10 km; Middle: 2.5 km; Right: 1 km) for different sub-basins. The location of the sub-basins is shown on Fig. 2 below.*

Following these substantial changes, we will propose a new title for the paper: *Assessing the factors governing prediction skill of extreme late-spring flood in mountainous terrain*. The section *Discussion* of the revised manuscript will be rewritten to better highlight the scientific contribution of this study.

Where is the solid statistic?

An improved experimental design will be used in the revised manuscript. 12 unregulated stations located in the headwaters of the Red Deer (2 stations), Bow (6 stations) and Oldman (4 stations) River basins will be selected for the evaluation of hydrological simulations. Among these 12 stations, 4 stations were used in the initial manuscript. The location of these stations is given on the map below. In total, the river basins of these 12 stations cover 12700 km² and were characterised by different total precipitation amount, initial snowpack conditions and soil characteristics. For each of this stations, hydrological simulations will be evaluated in terms of flood volume, peak magnitude and timing. This will provide a more solid evaluation framework to assess the factors governing prediction skill of extreme late-spring flood in mountainous terrain.

[Figure]

*Figure 2. Location of the 12 hydrometric stations proposed for the evaluation of hydrological simulations in the revised manuscript. The black lines represent the limits of the catchments associated with each station. The orange polygon delineates the location of the catchments that were used for model evaluation in the initial manuscript.*

**Comments from the pdf**

**Main points:**
- Your study is a case study. Fits in my opinion better in NHESS than in HESS. I am personally not very interested in such studies, where actually only the data owner and model operators learn something about their specific data chain in a single event.

See above our answer to this comment.

- Where is the true novelty? You use well citable tools and data for a specific event. Their present combination might be novel, but surely not original (different resolution, different initial conditions, different density of stations).
See above our answer to this comment.

-Figures 10-12 suggest a range based on only two member. This is not adequate.
We thank Reviewer 2 for this comment. The filled area will be systematically removed from the figures showing hydrographs in the revised manuscript to avoid any mis-interpretation of the results.

**Introduction**
The whole introduction reviews previous analysis of the here analyzed event and introduces the specific models and data used by the authors.
Please correct me, but I don't see any link to current challenges in hydrological modelling of rain-on-snow events, current methods in simulation of floods and previous similar approaches in other areas.
What is the knowledge gap here?
The fact that "Despite its severe hydrological consequences, this extreme weather event has received little attention from a hydrological modelling point of view."?
Is the question "cam my model cope with this event" still a relevant one in the times where model assessment relies on multi-model and multi ensemble input for periods of several years?

The Introduction in the revised manuscript will be fully rewritten to put our study in a general context and better highlight the current challenges in modelling of extreme floods in complex terrain and downstream regions.

We will first include a general paragraph on extreme flooding events in mountainous terrain mentioning previous severe events such as the flood in the mountainous catchments of central Europe in June 2013 (Grams et al., 2014), the Colorado Flood in September 2013 (Gochis et al., 2015) or previous flooding events in the mountains of Western Canada (Buttle et al., 2016). These events were characterised by large amount of rainfall strongly influenced by the local topography (e.g. Friedrich et al., 2016). In late spring, these heavy rainfall can occur with high-freezing level leading to a rain-on-snow (ROS) event in areas that are still covered by snow (Mc Cabe et al., 2007; Corripio and Lopez-Moreno, 2017). The June 2013 Flood in Alberta corresponds to this type of event (Pomeroy et al., 2016). We will then briefly detail the main hydro-meteorological features of this extreme event.

We will then write paragraphs on the modelling strategies that are used to predict these extreme events (e.g. Hapuarachchi et al., 2011; Pagano et al. , 2014). In particular, we will discuss the main source of uncertainties for operational flood forecasting in complex terrain (Mascaro et al., 2010, Zappa et al., 2011). We will highlight the importance of the spatial and temporal accuracy of the rainfall forcing (e.g. Jasper et al., 2002; Vincendon et al., 2011, Lobligeois et al., 2014). The influence of initial soil moisture and snow conditions will be discussed as well (e.g. Anquetin et al., 2010; Silvestro and Rebora, 2010; Edouard et al. 2018). In particular, we will mention the benefit for runoff prediction of using snow information from a dedicated external snow monitoring system (Jörg-Hess et al., 2015; Griessinger et al., 2016). Finally, the uncertainties associated with the soil dataset will be mentioned (Lovat et al., 2019).

A specific paragraph presenting the challenges of ROS modelling will then be written. We will mention past studies focusing on hydrological prediction of specific rain-on-snow events (Rössler et al., 2014, Corripio and Lopez-Moreno, 2017) and the main conclusions from these studies. In particular, we will insist on the importance of an accurate estimation (i) the areal extent and snow water equivalent at the beginning of the ROS (McCabe et al., 2007) (ii) the contribution of the turbulent fluxes to total snow melt during ROS events (e.g. Marks et al., 1998; Garvelmann et al., 2014; Würzer et al., 2016) and (iii) the evolution of the rain/snow partitioning during the event (Jasper et al., 2002) and the potential benefit of the phase partitioning from cloud microphysical schemes implemented in atmospheric models (Harpold et al., 2017). We will also mention the influence of the initial snowpack properties at the beginning of the ROS (e.g. Würzer et al., 2016).

We will finally describe the scientific objectives of this paper and give an overview of the modelling strategy.

The list of the references mentioned here is given at the end of this document.

P 2 L 27: Pagano, T. C., Wood, A. W., Ramos, M. H., Cloke, H. L., Pappenberger, F., Clark, M. P., ... & Verkade, J. S. (2014). Challenges of operational river forecasting. Journal of Hydrometeorology, 15(4), 1692-1707.
As described in our previous answer, this reference will be added to the introduction when describing the general context of your study.

**Text**
P4. L4L Are these data available only for the event or also for some training periods before?
Measurements of precipitations and snow water equivalent are available both for this event and for other time periods before and after the flooding event. Daily discharge data are available through the Canadian Water Office website and cover the periods of operation of the different hydrometric stations. Finally, hourly discharge data for the flooding event were obtained through a specific request to the Canadian Water Office. Only hourly discharge data were used in our study since we focused on the short-term hydrological response during the flooding event. We will improve the description of the data availability in the revised manuscript.

P5 L1-2 Large literature on this: E.g.
Jonas, T., Marty, C., & Magnusson, J. (2009). Estimating the snow water equivalent from snow depth measurements in the Swiss Alps. Journal of Hydrology, 378, 161-167. https://doi.org/10.1016/j.jhydrol.2009.09.021
We fully agree with Reviewer 2 that methods such the one proposed by Jonas et al. (2009) could be used to obtain an estimation of SWE from the ECCC snow depth analysis, even if this method was initially developed in the Swiss Alps and may present limitations in the Canadian Rockies. The low resolution of the ECCC snow analysis (10 km) is the main reason why we did not use this product in our study. We will make this point more clear in the revised manuscript.

P6 L15-16 These are operational runs or ad-hoc hindcasts?
All the GEM simulations used to drive the hydrological simulations are ad-hoc hindcasts that were specifically generated for this study using the operational versions of the GEM atmospheric model available in June 2018. We will clarify this point in the revised manuscript.

P 7 L 7-9 Large set on international literature on this.
E.g.: Germann, U., Galli, G., Boscacci, M., & Bolliger, M. (2006). Radar precipitation measurement in a mountainous region. Quarterly Journal of the Royal Meteorological Society: A journal of the atmospheric sciences, applied meteorology and physical oceanography, 132(618), 1669-1692.
The reference provided in the initial manuscript was published in an international journal and was specific to the June 2013 flooding event. We will consider adding additional references from the international literature in the revised manuscript.

P 8 L1 I accept your argumentations. This is for me a reason to ask nevertheless for simulation of other (smaller) events in order to evaluate the non-randomness of your analyses.
We do not plan to add the simulations of additional events in the revised manuscript. Indeed, this work has required the generation of new GEM hindcasts of the June 2013 Flood at different resolutions. This cannot be easily extended to other events. Instead, a new set of 12 stations will be used for model evaluation in the revised manuscript (see our answer above to the second general comment).. This new set of stations will consist of the main unregulated stations located in the headwaters of the Oldman, Bow and Red Deer Rivers basins. We believe that this new selection of stations will allow us to derive less localised conclusion and improve the quality of our analysis.

P 8 L 23: Sevruk, B. (1983). Correction of measured precipitation in the Alps using the water equivalent of new snow. Hydrology Research, 14(2), 49-58.

Nešpor, V., & Sevruk, B. (1999). Estimation of wind-induced error of rainfall gauge measurements using a numerical simulation. Journal of Atmospheric and Oceanic Technology, 16(4), 450-464.

Savina, M., Schäppi, B., Molnar, P., Burlando, P., & Sevruk, B. (2012). Comparison of a tipping-bucket and electronic weighing precipitation gage for snowfall. Atmospheric Research, 103, 45-51.

We will add these references in the revised manuscript.

P 8 L 28-29

This study also uses alternatively row model conditions and SWE estimations at model initialization in the context of runoff and SWE forecasting in mountainous terrain. SWE estimation based on observations improve the forecast of discharge volume

Jörg-Hess, S., Griessinger, N., & Zappa, M. (2015). Probabilistic forecasts of snow water equivalent and runoff in mountainous areas. Journal of Hydrometeorology, 16(5), 2169-2186. https://doi.org/10.1175/JHM-D-14-0193.1

P9 L3: Figure 10 in  Jörg-Hess, S., Griessinger, N., & Zappa, M. (2015). Probabilistic forecasts of snow water equivalent and runoff in mountainous areas. Journal of Hydrometeorology, 16(5), 2169-2186. https://doi.org/10.1175/JHM-D-14-0193.1

We thank Reviewer 2 for this very interesting reference. It will be added in the introduction as well as in the discussion of the revised manuscript. The impact of the insertion of SNODAS SWE on the simulation of the flood volume will be discussed in regards to the results obtained by Jörg-Hess et al. (2015) on the forecast of runoff volume in the Swiss mountains.

P 10 L3-5 : Is such behavior to be expected also for less severe events?

An overall improvement of quantitive precipitation estimation with atmospheric models running at convective-resolving resolution compared to lower resolution models has been obtained in many studies in complex terrain (Rasmussen et al., 2011; Gutmann et al., 2012; Lundquist et al., 2019). Therefore, we estimate that similar conclusions are expected for less severe events as well.   ß

- Gutmann, E., Rasmussen, R., Liu, C., Ikeda, K., Gochis, D., Clark,M., et al. (2012). A comparison of statistical and dynamical downscaling of winter precipitation over complex terrain. J. Clim. 25, 262–281. doi: 10.1175/2011JCLI4109.1

- Lundquist, J., Hughes, M., Gutmann, E., & Kapnick, S. (2019). Our skill in modeling mountain rain and snow is bypassing the skill of our observational networks. Bulletin of the American Meteorological Society, (2019).

- Rasmussen, R., Liu, C., Ikeda, K., Gochis, D., Yates, D., Chen, F., et al. (2011). High-resolution coupled climate runoff simulations of seasonal snowfall over Colorado: a process study of current and warmer climate. J. Clim. 24, 3015– 3048. doi: 10.1175/2010JCLI3985.1

P 10 L 6 Red Deer seem to be also fine at 10 km standard application. Why?

CaPA 10 km Def shows improved performances for the Red Deer River basin compared to the Bow and Oldman Rivers basins as shown on Fig. 7a of the initial manuscript. This can be explained by the fact that most of the precipitation over the Red Deer basin fell between 20 June 06 UTC and 21 June 00 UTC, mostly after the transition from convective to stratiform precipitation that occurred between 06 and 08 UTC on 20 June (Kochtubajda et al., 2016; Li et al., 2017). GEM at 10 km provides a more accurate precipitation background during the stratiform period than during the convective period. However, a more detailed look at Fig. 5 of the initial manuscript shows that CaPA 10 km Def  presents an underestimation of precipitation on the eastern side of the headwaters of the Red Deer River and an overestimation in the western side of the headwaters. This results from an error in the location of the precipitation in the 10-km background that was not corrected by the analysis since no station was used in the headwaters of the Red Deer River by CaPA 10 km Def (see Fig. 1 of the initial manuscript). The additional stations used in CaPA 10 km New corrected this error in the location of the precipitations (Fig. 6d of the initial manuscript), leading to improved performances of the analysis for the Red Deer Basin (Fig. 7b of the initial manuscript).

A table containing the performances metrics (Bias, RMSE, R2) for each CaPA experiment and each main basin will be added to the revised manuscript as suggested by Reviewer 3. In addition, the performances metrics will be computed for the whole event and computed separately for the convective and stratiform periods as defined in Kochtubajda et al. (2016) and Li et al. (2017). This will allow a better explanation of the performances of the CaPA experiments for the different basins.

P 10 L 17-18: Citation?
We will add the following references to the revised manuscript:
- Bracken, L. J., Cox, N. J. and Shannon, J. (2008), The relationship between rainfall inputs and flood generation in south–east Spain. *Hydrol. Process.*, 22: 683-696. doi:10.1002/hyp.6641
- Huang, Y., Bárdossy, A., and Zhang, K.: Sensitivity of hydrological models to temporal and spatial resolutions of rainfall data, Hydrol. Earth Syst. Sci., 23, 2647–2663, https://doi.org/10.5194/hess-23-2647-2019, 2019.

P 11 L 7 : Nice Figure, but difficult to see the black lines from the barcharts to the gauges.
Thanks for this comment. The bar charts will be modified in the revised manuscript to improve the readability of the figure. Note that the figure will be modified as well since new hydrometric stations will be used for model evaluation (see our answer to the first general comments).

P 11 L 9: So, 10km_Def always profits from SND instead of OPT. The other forcings partly.
The atmospheric forcing *10km_Def* suffers from a systematic underestimation of precipitation which is systematically and partially compensated by the increase in initial SWE when using *SND* as the initial snow condition. The other precipitation forcing benefits from additional stations which greatly improves their quality in the headwaters, removing the underestimation of precipitation and removing the main source of uncertainty of the hydrological simulations. Therefore, hydrological simulations using these improved atmospheric forcing are more sensitive to error in the initial snow conditions; in particular in the headwaters where SND tends to overestimate the initial SWE conditions and lead to overestimated flood volume (see station 05BL019 for example).
The different source of uncertainties and there relative importance will be better described in the revised version of the paper, as mentioned above in our answer to the first general comment.

P 11 L 23: Same for at least two other creeks
These two creeks correspond to the hydrometric stations 05BL027 and 05BL023. No snow was present in the drainage basins of these 2 stations on 18 June 2013. Note that these two stations will be removed from the analysis in the revised manuscript.

P12 L 1-2: I don't see the point in filling the areas between OPT and SND in panels c) and f) of both Figures. You suggest here a range like for probabilistic forecasts with only two members.
Same in Figure 12.
See above our answer to the 3rd general comment.

P 12 L 26-27 : We see the results, but we don't see the numbers for all Manning options. Table would be useful
We thank Reviewer 2 for this comment. In the revised manuscript, the results corresponding to the different Manning options will be removed from the section *Results*. We will describe the selection of the correction factor applied to the default values of the Manning coefficients in the section *Methods*. Indeed, as mentioned by Reviewers 2 and 3, the adjustment of the Manning coefficient does not bring novelty to our study. Therefore, we decided to present it as a calibration step in the revised manuscript. The correction factor selected at this calibration step will then be used in the rest of the analysis.
The figures below shows the impact of the correction factor on the distribution of model errors in terms of peak magnitude and timing. Results are given for all the hydrometric stations and for 4 hydrological simulations: *2.5km_OPL*, *2.5km_SND*, *1.0km_OPL* and *1.0km_SND*. These figures will be added to the supplementary material of the revised manuscript

[Figure]

*Distribution of model errors in terms of peak timing (Left) and magnitude (Right) for different correction factors applied to the default Manning coefficients.*

This new presentation of the Manning options will leave more room in the section *Results* for additional results that will strengthen the paper as described above. The challenges associated with the choice of the routing parameter for such extreme event will be kept in the discussion.

P 14 L 32: Is this really the first citation to a international study?

The paper by Bernhardt and Schulz (2010) describes the SnowSlide scheme which can be considered in the snow hydrology community as one of the reference schemes that simulate gravitational snow redistribution in a hydrological model. However, it is clear that it is not the first time that the importance of gravitational snow redistribution for mountain snow hydrology is mentioned in a paper. In the revised manuscript, we will add the reference to the one of the earlier studies on this topic (Blösch and Kirnabauer, 1992).

*Blösch, G. and Kirnbauer, R. (1992), An analysis of snow cover patterns in a small alpine catchment. Hydrol. Process., 6: 99-109. doi:10.1002/hyp.3360060109*

List of references that will included in the introduction (not exhaustive):

Anquetin, S., Braud, I., Vannier, O., Viallet, P., Boudevillain, B., Creutin, J. D., & Manus, C. (2010). Sensitivity of the hydrological response to the variability of rainfall fields and soils for the Gard 2002 flash-flood event. *Journal of hydrology*, *394*(1-2), 134-147.

Buttle, J. M., Allen, D. M., Caissie, D., Davison, B., Hayashi, M., Peters, D. L., ... & Whitfield, P. H. (2016). Flood processes in Canada: Regional and special aspects. *Canadian Water Resources Journal/Revue canadienne des ressources hydriques*, *41*(1-2), 7-30.

Corripio, J., & López-Moreno, J. (2017). Analysis and Predictability of the Hydrological Response of Mountain Catchments to Heavy Rain on Snow Events: A Case Study in the Spanish Pyrenees. *Hydrology*, *4*(2), 20.

[revised manuscript text omitted]

---

## Author Comment (AC3) · 1 Oct 2019

**Answer to Reviewer 3 HESS-2019-152**

We thank Reviewer 3 for his insightful comments. We provide here our responses to his comments, including the plan to revise the manuscript in response to reviewer comments. The original review comments are in normal black font while our answers appear in blue font.

**General Comments**

This paper examined the capability of a hydrological model (GEM-Hydro) in simulating the June 2013 flood event in Alberta, Canada. In particular, three sub-basins in the Bow River basin were selected to assess the impacts of spatial resolution, precipitation gauge density, and initial snow conditions on model ability in reproducing the flow volumes. Also, the model sensitivity to Manning coefficients in capturing the peak flow was investigated.

General Comments:
The objective of this paper is straightforward and this paper is well-written, easy to follow and well-structured. However, the creditability of the study has been reduced because the study is highly localized and reads more like a report of an application of a hydrological model to a specific flood event. There are little knowledge gain for the community. In general, two major concerns are needed to be addressed in this paper:

1) Novelty of the study

While the study would be a great contribution to the development of a Canadian hydrological forecasting system, it might not be novel enough as a scientific contribution for the international community. The effects of different spatial resolutions on model simulations have been previously and heavily studied in different hydrological models and it is well expected that finer resolution could provide better simulations because of its ability in capturing the fine-scale hydro-meteorological processes. It is also expected that the inclusion of additional information (e.g. increasing network density, inclusion of SWE information) would improve model performance because of the data-driven nature of the sophisticated models nowadays. Lastly, the Manning coefficients (both channel and floodplain) are well known to be one of the most sensitive routing parameters in any hydrological/ hydraulic models. Adjusting such parameter will definitely improve the model ability in matching the peak flow timing and magnitude.

Furthermore, the Discussion section of the study did not provide innovative insights on modelling of extreme flood events. Although the Discussion section was well-written and fully supported by references, the major findings of this study were merely a confirmation of what had been shown in many previous studies (as repeatedly mentioned by the authors). Therefore, there is a lack of true novelty and scientific contribution in this study. The authors should vigorously address this critical issue by providing a better discussion on what new knowledge and information the international community could learn from this study.

We thank Reviewer 3 for this general comment. Significant changes will be made in the revised manuscript to make sure that this study bring enough novel contributions to the international community. These changes are detailed below.

First of all, some of the results that were presented in the initial manuscript will be removed since they correspond to expected results with a lack of real novelty as detailed above by Reviewer 3. We will remove from the manuscript:
  - the precipitation analysis at 10 km using the default stations (*CaPA 10 km Def*). This precipitation analysis used the same stations as the operational analysis at the time of the event but its quality is mediocre due the low density of the stations in complex terrain. In the revised manuscript, we will only keep the precipitation analysis generated with all the precipitation data for each horizontal resolution (10, 2.5, 1 km). Theses different precipitation data will be presented as our best estimation of the precipitation at each horizontal resolution.

- the results corresponding to the different Manning options that were described in the section *Results*. We will describe the selection of the correction factor applied to the default values of the Manning coefficients in the section *Methods*. Indeed, as mentioned by Reviewers 2 and 3, the adjustment of the Manning coefficient does not bring novelty to our study. Therefore, we decided to present it as a calibration step. The correction factor selected at this calibration step will then be used in the rest of the analysis. Figures describing the impact of the correction factor will be added to the supplementary material of the revised manuscript. The challenges associated with the choice of the routing parameter for such extreme event will be kept in the section *Discussion*.

Removing these two components of the initial manuscript will allow us to focus the revised manuscript on the factors governing the predictions skill of extreme late-spring flood in mountainous terrain. The three **scientific questions** of the **revised manuscript** will be as follows:

1. How does the resolution of the atmospheric forcing influence the predictability of the hydrological response during extreme late-spring flood in mountainous terrain?
2. At kilometric scale, can the rain/snow partitioning from an advanced cloud microphysical scheme improve the predictability of the hydrological response during this kind of event?
3. How does the sensitivity of the hydrological response to the resolution of the atmospheric forcing and to the phase partitioning method compare to (i) the sensitivity to the initial snow conditions and (ii) the uncertainty in soil datasets?

An ensemble of hydrological simulations will be used to answer these questions:

- three atmospheric forcings will be used at 10, 2.5 and 1 km as in the initial manuscript. At each resolution, the precipitation forcing will consist of the CaPA analysis including all the stations available in the region to obtain our best estimate of precipitation at a given resolution.
- for each atmospheric forcing, the same two initial snow and soil conditions will be used as in the initial manuscript.
- finally for each atmospheric forcing and initial snow/soil conditions, two different soil database will be considered: the Global Soil Dataset for Earth System Model (GSDE, (Shangguan et al., 2014) and the Soilgrid dataset (Hengl et al., 2017). These 2 database can be considered as reference soil databases for Earth system models (Dai et al, 2019).
- for the atmospheric forcing at 1 km, for each soil dataset and initial snow conditions, three hydrological simulations will be carried out with GEM-Hydro using different phase partitioning methods (PPM): (i) a constant threshold depending on air temperature (0°C, the default value in GEM-Hydro), (ii) a more advanced PPM combining temperature and relative humidity (Harder and Pomeroy, 2013) and (iii) the direct precipitation phase from the P3 cloud microphysical model running in GEM at 1 km.

As suggested by Reviewer 3 in a specific comment below, the evaluation of the hydrological simulations will be carried out at **12 unregulated stations** located in the **headwaters** of the **Red Deer** (2 stations), **Bow** (6 stations) and **Oldman** (4 stations) River basins. Among these 12 stations, 4 stations were used in the initial manuscript. The location of these stations is given on Figure 2 below and their main characteristics are reported in Table 1 below. In total, the river basins of these 12 stations cover 12700 $km^2$ and are characterised by different soil characteristics, total precipitation amount during the flooding event and initial snowpack conditions prior to the flood. We strongly believe that this new set of the stations will made our analysis less localised and will allow us to derive more general conclusions useful for the international community.

For example, we will add and discuss in the revised manuscript results on the influence of the resolution of the atmospheric forcing on the simulation of the melt dynamic during the rain-on-snow event at each sub-basin where snow was initially present. Figure 1 shows that the contributions of the turbulent fluxes to the total energy input to the snowpack is larger with the atmospheric forcing at 2.5 and 1 km than at 10 km. This result is consistent across all the sub-basins and results from larger wind speed at 2.5 and 1 km than at 10 km. It illustrates that the importance of the wind field downscaling on

simulated turbulent fluxes and resulting snowmelt. To our knowledge, a quantification of these effects on the hydrological response during extreme floods has never been proposed in previous studies and constitutes an original contribution of our study.

[Figure]

*Figure 1: Contribution of energy balance terms to the total averaged energy input to the snowpack (expressed in millimetre melt equivalent) during the flooding event simulated using atmospheric forcing at different resolutions (Left: 10 km; Middle: 2.5 km; Right: 1 km) for different sub-basins. The location of the sub-basins is shown on Fig. 2 below.*

Following these substantial changes, we will propose a new title for the paper: *Assessing the factors governing prediction skill of extreme late-spring flood in mountainous terrain*. The section *Discussion* of the revised manuscript will be rewritten to better highlight the scientific contribution of this study.

2) Lack of connection with other worldwide extreme flood events
The Introduction section was again well-written and the rationale of the study was well presented, however, it was highly focused on the description of the June 2013 flood event and the previous works that were related to the 2013 flooding. The literature review did not discuss any research related to other worldwide flood events and any modelling works that address the current challenges of the modelling community in dealing with extreme flood induced by rain-on-snow events. This makes the study highly localized and event specific. The authors should provide a boarder discussion on similar research conducted in other regions, modelling strategy used in simulating such kind of flood events, and the research gaps this study could fill in for advancing the knowledge of the community.

The Introduction in the revised manuscript will be fully rewritten to put our study in a general context.

We will first include a general paragraph on extreme flooding events in mountainous terrain mentioning previous severe events such as the flood in the mountainous catchments of central Europe in June 2013 (Grams et al., 2014), the Colorado Flood in September 2013 (Gochis et al., 2015) or previous flooding events in the mountains of Western Canada (Buttle et al., 2016). These events were characterised by large amount of rainfall strongly influenced by the local topography (e.g. Friedrich et al., 2016). In late spring, these heavy rainfall can occur with high-freezing level leading to a rain-on-snow (ROS) event in areas that are still covered by snow (Mc Cabe et al., 2007; Corripio and Lopez-Moreno, 2017). The June 2013 Flood in Alberta corresponds to this type of event (Pomeroy et al., 2016). We will then briefly detail the main hydro-meteorological features of this extreme event.

We will then write paragraphs on the modelling strategies that are used to predict these extreme events (e.g. Hapuarachchi et al., 2011; Pagano et al. , 2014). In particular, we will discuss the main source of uncertainties for operational flood forecasting in complex terrain (Mascaro et al., 2010, Zappa et al., 2011). We will highlight the importance of the spatial and temporal accuracy of the rainfall forcing (e.g. Jasper et al., 2002; Vincendon et al., 2011, Lobligeois et al., 2014). The influence of initial soil

moisture and snow conditions will be discussed as well (e.g. Anquetin et al., 2010 , Silvestro and Rebora, 2010; Edouard et al. 2018). In particular, we will mention the benefit for runoff prediction of using snow information from a dedicated external snow monitoring system (Jörg-Hess et al., 2015; Griessinger et al., 2016). Finally, the uncertainties associated with the soil dataset will be mentioned (Lovat et al., 2019).

A specific paragraph presenting the challenges of ROS modelling will then be written. We will mention past studies focusing on specific hydrological prediction of rain-on-snow events (Rössler et al., 2014, Corripio and Lopez-Moreno, 2017) and the main conclusions from these studies. In particular, we will insist on the importance of an accurate estimation (i) the areal extent and snow water equivalent at the beginning of the ROS (McCabe et al., 2007) (ii) the contribution of the turbulent fluxes to total snow melt during ROS events (e.g. Marks et al., 1998; Garvelmann et al., 2014; Würzer et al., 2016) and (iii) the evolution of the rain/snow partitioning during the event (Jasper et al., 2002) and the potential benefit of the phase partitioning from cloud microphysical schemes implemented in atmospheric models (Harpold et al., 2017). We will also mention the influence of the initial snowpack properties at the beginning of the ROS (e.g. Würzer et al., 2016).

We will finally describe the scientific objectives of this paper and give an overview of the modelling strategy.

The list of the references mentioned here is given at the end of this document.

Specific Comments:
P1L28-29: How could the results of this study guide the development of the hydrological forecasting system worldwide?
Our answers to the two above general comments show how our study can contribute to the improvement of the general knowledge on flood modelling in complex terrain. Therefore, we believe that our results are interesting for the international community working in this subject. The Abstract will be re-written to better highlight the general contribution of our study.

P4L10: Could the authors comment on the consistency of these four different networks?
These four different meteorological have been deployed for different purposes. For example, the SYNOP and ABE stations are well distributed across Alberta to provide a permanent monitoring of weather conditions across the province. On the other hand, CRHO stations are distributed around targeted mountainous catchments and provide very valuable information in poorly observed areas by the other networks. CaPA has been designed to deal with different precipitation networks and insure the consistency of the precipitation data when they are included in the analysis. A spatial consistency test is applied in the Quality-control (QC) procedures of CaPA to identify and remove observations with large errors from the final analysis. The QC procedures are detailed in Lespinas et al. (2015). We will add in the revised manuscript a comment on the consistency of the different networks and mention the QC procedures used in CaPA.

P5L3-4: What are the drainage areas of these river basins?
The answer to this question is given in Table 1 shown below.

P5L6-7: Could the authors provide the basic information of these 10 stations (e.g. station name, drainage area, name of tributary)?
As mentioned above in our answer to the first of the general comments and as proposed by Reviewer 3 in one of his following comment, a new set of 12 stations will be used for model evaluation in the revised manuscript. This new set of stations will consist of the main unregulated stations located in the headwaters of the Oldman, Bow and Red Deer Rivers basins. In the revised manuscript, we will include a table that provides the main information for the hydrometric stations used for model evaluation. A preliminary version of this table is given below and a map showing the location of the stations is shown in our answer to one of the following specific comments. (Figure 2).

*Table 1. Characteristics of the hydrometric stations proposed for model evaluation in the revised manuscript*

| Station Code | Station Name | Drainage Area (km²) | Included in the initial paper |
|---|---|---|---|
| 05CA004 | Red Deer River Above Panther River | 941 | No |
| 05CA009 | Red Deer River Below Burnt Timber Creek | 2246 | No |
| 05BB001 | Bow River at Banff | 2210 | No |
| 05BG010 | Ghost River Above Waiparous Creek | 484 | No |
| 05BH015 | Jumpingpound Creek at Township Road | 474 | Yes |
| 05BJ010 | Elbow River at Sarcee Bridge | 1189 | Yes |
| 05BL012 | Sheep River at Okotoks | 1494 | Yes |
| 05BL019 | Highwood River at Diebel's Ranch | 774 | Yes |
| 05AB041 | Willow Creek at Oxly Ranch | 833 | No |
| 05AA035 | Oldman River at Range Road | 1835 | No |
| 05AA008 | Crowsnest River near Franck | 403 | No |
| 05AA022 | Castle River near Beavers Mines | 820 | No |

P10L10-12: It would be better to provide the performance measures (Bias, RMSE, R2) of each basin in a table.

A table containing the performances measures for each CaPA experiment and each main basin will be added to the revised manuscript.

P10L13-14: Could the authors explain how the negative bias was removed by increasing the spatial resolution to 2.5km and 1km? Figures 7c) and 7d) do not show an obvious removal of the negative bias, instead, the points are still scattered above and below the 1:1 line.

Improvements in the precipitation analysis from 10 km to 2.5 km and 1 km resulted from improvements in the precipitation background provided by the Canadian NWP system GEM. GEM at 2.5 km and 1 km resolves explicitly part of the convection with the cloud microphysical scheme P3 whereas all the convection is parameterised in GEM at 10 km. Previous studies by Li et al. (2017) and Milrad et al. (2017) showed that atmospheric models at convection-permitting resolution performed best for this event. As detailed by Milrad et al. (2017), this is mainly due to (i) an improved representation of the orographic ascent that contributed to the magnitude of the extreme rainfall (ii) an anchoring and increasing duration of the precipitation on the Eastern side of the Rockies. Compared to the analysis at 10 km, the bias has been improved at 2.5 and 1 km since the points are better centred around the 1:1 line. However, as mentioned by Reviewer 3, these points are still scattered around this line with leads to similar RMSE. Better explanations will be given in the revised version of the manuscript.

P10L15-16: Could the authors explain why there were overestimations of cumulative precipitation using CaPA 1.0km?

GEM at 1 km explicitly simulated most of the convection with its cloud microphysical scheme. This led to the explicit representation of organised high-precipitation cells during the convective stage of this extreme rainfall event. However, the exact location of these precipitation cells was not necessarily well captured by the atmospheric model. Therefore, GEM at 1 km simulated unrealistically highaccumulation at some locations that were not systematically corrected by CaPA during the leave-one-out evaluation. More details will be added in the revised manuscript.

P11L1-2: While the analysis on the effects of spatial resolution, gauge network density, and initial snow condition were conducted across the Oldman, Bow, and Red Deer River basins, the hydrological simulations were evaluated on Jumpingpound Creek, Elbow River and Highwood River, which are all located within the Bow River basin. I think such selection of hydrological simulations could not fully reveal the impacts of those factors controlling the flood dynamics. A better experimental design could be selecting one or two headwater sub-basins from each river basin (Oldman, Bow, and Red Deer). The results could potentially provide more information than the current setting (10 stations all within the Bow River basin) especially when different responses were witnessed after the inclusion of additional information (precipitation and/or SWE) across the three basins (e.g. consistently underestimation of cumulative precipitation plus overestimation of SWE in the Bow River, a mixture of over-and under-estimation of cumulative precipitation in the Red Deer, and underestimation of cumulative precipitation plus fairly accurate SWE estimates in the Oldman River). I wonder why the authors only focused on hydrological simulations within the Bow River basin.

Only hydrometric stations within the Bow Rover basin were selected in the initial manuscript since we focused our evaluation of hydrological simulations over one of the area that was the most severely impacted during the flooding event and that was monitored by a relatively dense network of hydrometric stations. We agree with Reviewer 3 that this selection of stations limited our ability to analyse the impact of the different factors governing the flood dynamics.

Following this comment, a better experimental design will be used in the revised manuscript. 12 unregulated stations located in the headwaters of the Red Deer (2 stations), Bow (6 stations) and Oldman (4 stations) River basins will be selected for the evaluation of hydrological simulations. Among these 12 stations, 4 stations were used in the initial manuscript. The location of these stations is given on the map below and their main characteristics are reported in the table on the previous page. In total, the river basins of these 12 stations cover 12700 km² and were characterised by different total

[Figure]

*Figure 2. Location of the 12 hydrometric stations proposed for the evaluation of hydrological simulations in the revised manuscript. The black lines represent the limits of the catchments associated with each station. The orange polygon delineates the area that was studied in the initial*

precipitation amount, initial snowpack conditions and soil characteristics. We strongly believe that this new set of stations, not restricted to the Bow river basin, will improve the credibility of our study and bring interesting discussions on the main factors controlling the flow dynamics for contrasted regions.

P15L30-P16L15: This study did not examine the model structure and the process representation in the model at all. Providing a list of potential reasons that might affect the model performance here becomes irrelevant unless concrete proof and result analysis are given to show the underlying causal relationship. This sub-section should be better re-written.
We will carefully review this sub-section and rewrite it to be more consistent with the results shown in our study. We will keep in the revised manuscript a part of the discussion where the limitations of the land surface scheme SVS for mountain hydrology are mentioned.

P17L19-20: What about the ground station data from the ABE, CHRO, and COOP networks? Are they publicly available to the international readers?
All the data from the ABE, CRHO and COOP networks are available to the international readers through requests on dedicated web-portals. Data from ABE stations can be downloaded at: https://agriculture.alberta.ca/acis/alberta-weather-data-viewer.jsp. CHRO data are available via the web portal: http://giws.usask.ca/meta/. Finally COOP data are available at https://www.ncdc.noaa.gov/data-access/land-based-station-data/land-based-datasets/cooperative-observer-network-coop . The section *Data availability* of the revised manuscript will include these links.

Remarks:
P2L4-5, 10, and 12: could the authors check this reference please? Is this reference the correct one that the authors intended to use? I guess it should be Pomeroy et al. 2016a. Please correct me if I am wrong.
Indeed we were using the wrong reference. The correct one is Pomeroy et al (2016b) and we will make the correction in the revised manuscript.
*Pomeroy, J. W., R. E. Stewart, and P. H. Whitfield. The 2013 flood event in the South Saskatchewan and Elk River basins: Causes, assessment and damages. Can. Water Resour.. J. 41 (1-2), ,https://doi.org/10.1080/07011784.2015.1089190, 2016b*

We will incorporate the below corrections in the revised manuscript.

P8L18: should be "similar to" not "similarly to"

P8L24 and 27: please spell out the full name first before using the abbreviation

P8L33: missing full stop after "(18 June 2013)"

P12L5: should it be "Fig. 10c, f and Fig. 11c, f" instead of " Fig. 10d, f and Fig. 11d,f"?

P12L23: better use "hypothesis" instead of "assumption"

P17L15: delete "are"

Figures 3 and 4: better use "boundaries" instead of "limits"
The legend of the two figures will be modified according to this comment.

Figures 10-12: it is a bit misleading to use filled area to represent the results because essentially there are only two simulations using initial conditions from OPL and SND. They might not necessarily represent the upper and lower limits of the model performance, especially when more different initial conditions are used.
We agree with Reviewer 3 and the filled area will be systematically removed from the figures showing hydrographs in the revised manuscript.

List of references that will included in the introduction (not exhaustive):

Anquetin, S., Braud, I., Vannier, O., Viallet, P., Boudevillain, B., Creutin, J. D., & Manus, C. (2010). Sensitivity of the hydrological response to the variability of rainfall fields and soils for the Gard 2002 flash-flood event. *Journal of hydrology*, *394*(1-2), 134-147.

Buttle, J. M., Allen, D. M., Caissie, D., Davison, B., Hayashi, M., Peters, D. L., ... & Whitfield, P. H. (2016). Flood processes in Canada: Regional and special aspects. *Canadian Water Resources Journal/Revue canadienne des ressources hydriques*, *41*(1-2), 7-30.

Corripio, J., & López-Moreno, J. (2017). Analysis and Predictability of the Hydrological Response of Mountain Catchments to Heavy Rain on Snow Events: A Case Study in the Spanish Pyrenees. *Hydrology*, *4*(2), 20.

[revised manuscript text omitted]

Würzer, S., Jonas, T., Wever, N., & Lehning, M. (2016). Influence of initial snowpack properties on runoff formation during rain-on-snow events. *Journal of hydrometeorology*, *17*(6), 1801-1815.

Zappa, M., Jaun, S., Germann, U., Walser, A., & Fundel, F. (2011). Superposition of three sources of uncertainties in operational flood forecasting chains. *Atmospheric Research*, *100*(2-3), 246-262.

---

## Author Response (AR1)

Vincent Vionnet
Environnement and Climate Change Canada, Dorval, Canada
Tel.: +1 438 366 0148
Email: vincent.vionnet@canada.ca

December 20[th], 2019

Dear HESS Editor,

Please find enclosed a revised version of the manuscript HESS-2019-151. As requested by the three reviewers and as promised in our initial responses submitted on October 1[st], we made significant changes to the paper to improve its quality and make it relevant for the international hydrological community. We propose a new title for the revised manuscript: *Assessing the factors governing the ability to predict late-spring flooding in cold regions mountain basins.*

Based on the reviewers comments, new hydrological simulations using different soil texture databases were added. Overall, our revised study considers three sources of uncertainty for late-spring flood in mountain basins: (i) the resolution of the atmospheric forcings, (ii) the snow and soil moisture initial conditions and (iii) the representation of the soil texture. We used a new set of 12 unregulated stations covering in the headwaters of the Red Deer, Bow and Oldman River basins for model evaluation. To our knowledge, this is the first time these three source of uncertainties are considered simultaneously for a ROS event in several mountain basins. This allowed to derive conclusions on the key factors for the prediction of late-spring flood in mountain basins.

The manuscript has been heavily rewritten. In particular, we totally modified the introduction to present our study in a more general context. New results (including new figures) are presented in Section 3.3 that contains the evaluation of the hydrological simulations and the sensitivity of the hydrological responses. Based on these new results, the discussion section and the conclusions were modified. Our changes to the initial manuscript are detailed in the responses to the reviewers and in a version of the manuscript in track-change mode.

Note that the revised version of the manuscript does not include sensitivity tests to the phase portioning of precipitation contrary to what we proposed in our initial responses submitted on October 1[st]. Indeed, we realized that these tests would increase the size of the manuscript too much.

Thank you in advance for taking this new version of our paper into consideration,

Sincerely yours,

Vincent Vionnet
and co-authors.

Answer to Reviewer 1 HESS-2019-152

We thank Reviewer 1 for his comments. We provide here our responses to his comments and describe how we address his comments in the revised manuscript. The original reviewer comments are in normal black font while our answers appear in blue font. The lines numbers given below refer to the line number in the version of revised manuscript without track-change mode.

**General comments:**
This paper investigates the current modelling techniques available for flood forecasting and the associated required inputs. The evaluation was performed on a well documented significant flood in Canada in 2013, which resulted in large losses and damages in the affected area. The topography of the study area is very complex and addresses the limitation of the current flood prediction system used at ECCC. The material presented would benefit the modelling community working on refining and adjusting models to better predict floods in similar conditions and in complex terrain. In general the paper well written and relatively clear.

**Specific comments:**

On page 8, OPL and SND are introduced, its not clear what they mean.
"OPL" refers to the configuration of GEM-Hydro that ran in Open Loop mode from June 2012 to June 2013 and that was used to derive the initial land surface conditions. "SND" refers to the configuration of GEM-Hydro with insertion of SNODAS SWE on 1$^{st}$ May 2013 near peak snow accumulation. Configuration SND provided another estimation of land surface conditions prior to the flooding event. These acronyms are more clearly defined in the revised manuscript (P 10 L 12-13).

On page 9 and 10 section 3.2, could the authors give some more details for the cumulative underestimation of the stations.
The revised manuscript includes a more detailed evaluation of the different CaPA experiments, both spatially and temporally. As suggested by Reviewer 3, a table containing the performances metrics (Bias, RMSE, R2) for each CaPA experiment and each main basin (Oldman, Bow and Red Deer Rivers) has been added (Table 5 in the revised manuscript). To provide a more detailed assessment of the performances of the different precipitations forcing, the performances metrics have been computed for the whole event and separately for the convective and stratiform periods as defined in Kochtubajda et al. (2016) and Li et al. (2017). Table 5 shows in particular that the underestimation of precipitation was found for CaPA at 10 km during the convective phase for the Oldman and Bow River basins. More details are given in the text of the revised manuscript (P 11 L 18 to P 12 L 4).
Note that following the general comments of Reviewers 2 and 3, the CaPA experiment at 10 km using the default stations (*CaPA 10km Def* in the initial manuscript) that was characterised by a large underestimation of precipitation in the mountains and the foothills is not used anymore in the revised version of the paper.

Also some possible explanation why going from a 10km Capa grid to 2.5km grid provided better results.
Improvements in the precipitation analysis from 10 km to 2.5 km resulted from the improvement in the precipitation background provided by the Canadian NWP system GEM. GEM at 2.5 km resolves explicitly part of the convection with the cloud microphysical scheme P3 whereas all the convection is parameterised in GEM at 10 km. Previous studies by Li et al. (2017) and Milrad et al. (2017) showed that atmospheric models at convection-permitting resolution performed best for this event. As detailed by Milrad et al. (2017), this is mainly due to (i) an improved representation of the orographic ascent that contributed to the magnitude of the extreme rainfall (ii) an anchoring and increasing duration of the precipitation on the Eastern side of the Rockies. These explanations are clearly given in the revised manuscript. (P 11 L 30 to P 12 L 1)

Why does the Red Deer basin have a decrease in cumulative precipitation.
Figure 6 in the initial manuscript showed a decrease in cumulative precipitation for an area of the Red Deer basin for each CaPA experiment including the additional stations compared to the analysis with the default stations (*CaPA 10km Def*). This was explained by an overestimation of precipitation in this area for *CaPA 10km Def* (Fig 5. b in the initial manuscript). The additional stations in the new precipitations analysis (see their location on Fig. 1 in the initial manuscript) corrected this overestimation leading to a decrease of precipitation in this area. Since *CaPA 10km Def* is not used anymore in the revised manuscript, we did not add additional explanations regarding the decrease in total precipitation for the Red Deer basin in the revised manuscript.

How much does convection potentially play in these underestimates.
We thank Reviewer 1 for this comment. Indeed, the heavy precipitation event of June 2013 in Alberta consisted of 2 stages: a convective period followed by a stratiform period (Liu et al., 2016; Kochtubajda et al. 2016; Li et al., 2017). The convection was associated with an intense lightning activity and substantial rainfall accumulation (particularly for the Bow and Oldman rivers), triggering high flows in the eastern slopes the Rockies. In the revised manuscript, we evaluated separately the performances of the different CaPA experiments during the convective and the stratiform periods as well as for the whole event (see Table 5 of the revised manuscript). Results shows that the underestimation in *CaPA 10 km* was mainly found during the convective periods for the Oldman and Bow Rivers basins and during the stratiform period for the Red Deer River basins. *CaPA2.5 km* benefitting from a precipitation background at convection-permitting scale provided the best and nearly-unbiased estimate of precipitation during the flooding events. These results are detailed in Sect. 3.1 in the revised manuscript (P 11 L 18 to P 12 L 4).

On page 12, it seems that the river routing is a large unknown, in these simulations, are there ways to determine the actual routing for this event and apply it to the simulation. Is this even feasible?
As mentioned by Reviewer 1, the river routing strongly influenced the timing of the peak flow simulated by GEM-Hydro for this extreme flood. Determining the routing parameters for this specific event is challenging since major changes occurred in the geometry of the riverbeds so that parameters based on historical data may not be suitable for this extreme event. Routing parameters could be potentially determined for this event for river sections located in between two hydrometric stations. The observed discharge at the upstream station could be used as an input of the routing scheme and a calibration of the routing parameters could be carried out using the discharge of the downstream station for model evaluation. This would still require an estimation of the water input to river in between the two stations. This method would be restricted to certain portions of the river and could not be used for the headwaters of the catchments. Therefore, we still used in the revised manuscript a correction factor applied to the default Manning coefficient to optimise the overall peak magnitude and timing. The adjustment of the Manning coefficients is now presented in Sect. 2 in the supplementary material of the revised manuscript.
In addition, many river reaches in the region include hydroelectric dams and reservoirs whose storage affect streamflow at Calgary and elsewhere. The hydrometric stations used in this study are not affected by regulation but we note that far more than a simple adjustment of Manning's coefficient needs to be done to estimate river flows in this heavily managed region.

What is the frequency of reporting of the precipitation stations?
Precipitation stations used in this study were reporting at an hourly frequency for the SYNOP, CRHO and ABE networks. 6-h hour cumulated precipitation have then been derived from these hourly data to be used as input of the precipitation analysis. The frequency of reporting of the stations for each network is mentioned in the revised manuscript (P. 6 L. 3-4).

**Technical corrections:**

Page,3 line 20. Use "Another objective" rather than "Another main objective" This sentence was removed from the revised manuscript.

Page 8, what does OPL and SND stand for. The meaning of OPL and SND is more clearly defined at (P 10 L 12-13) of the revised manuscript (see also our answer above to the first specific comment).

Page 10, line 1. Remove "in", should read foothills of the Red Deer Basin. " This sentence was removed from the revised manuscript.

Page 10, line 11. "in particular in", remove the second "in". " This sentence was removed from the revised manuscript.

Page 10, line 20, use analyzed instead of analysed. Correction included here and in the rest of the text.

Page 11, line 17. "with to that " is awkward. " This sentence was removed from the revised manuscript.

Page 12, line 14. Remove space between 59 and %. " This sentence was removed from the revised manuscript.

Page 12, line 30 . Remove space between number signs. This sentence was removed from the revised manuscript.

Page 12, line 33. Comma after however. This sentence was removed from the revised manuscript.

Page 25, line29. "Radar data was" This sentence was removed from the revised manuscript.

Page 14, line 12. Theses should be These. Correction included

Page 15, line 13, similarly should be similar. Correction included

Page 15, line 26. Calibration of these parameters. " This sentence was removed from the revised manuscript.

Page 15 Lines 1-10. This sentence is very long and the entire section doe snot flow well. Consider rewriting to make the flow better. These sentences have been simplified in the revised manuscript.

Page 17 Line15. remove "are" to read forecasting systems can be expected. Also remove SVS from the sentence. This sentence was removed from the revised manuscript.

**Answer to Reviewer 2 HESS-2019-152**

We thank Reviewer 2 for his detailed comments. We provide here our responses to his comments and describe how we address his comments in the revised manuscript. The original review comments are in normal black font while our answers appear in blue font. The lines numbers given below refer to the line numbers in the version of revised manuscript without track-change mode.

**General Comments**

Dear authors,

Your study is a case study (a nice one, I admit). I am personally not very interested in such studies, where actually only the data owner and model operators learn something about their specific data chain in a single event in their region. With some adjustments it might fit in NHESS. For HESS I see too limited added information for the community.

Where is the true novelty? You use well citable tools and data for a specific event. Their present combination might be novel, but surely not original (different resolution, different initial conditions, different density of stations).

We thank Reviewer 2 for this comment. We made significant changes in the revised manuscript to show how this case study of an extreme flood in mountainous terrain is a valuable scientific contribution for the international hydrological community.

First, some of the results that were presented in the initial manuscript were removed in the revised manuscript since they correspond to expected results with a lack of real novelty for the international community as detailed by Reviewer 3. Therefore, we removed from the manuscript:

- the precipitation analysis at 10 km using the default stations (*CaPA 10 km Def).* This precipitation analysis used the same stations as the operational analysis at the time of the event but its quality is mediocre due the low density of the stations in complex terrain. In the revised manuscript, we only kept the precipitation analysis generated with all the station data for each horizontal resolution (10, 2.5, 1 km). Theses different precipitation data are now used in the revised manuscript as our best estimate of the precipitation at each horizontal resolution.
- the results corresponding to the different Manning options that were described in the section Results. The selection of the correction factor applied to the default values of the Manning coefficients is now detailed in the supplementary material and mentioned in the section Methods (P. 9 L 11-14). See below our answer to the specific comment on this question.

Removing these two components of the initial manuscript allowed us to focus the revised manuscript on the factors governing the predictions skill of late-spring flood in mountain basins. A new title for the revised manuscript is proposed ***Assessing the factors governing the ability to predict late-spring flooding in cold regions mountain basins***.

Late-spring flood in mountain basin represents a challenge for hydrological forecasting systems since they can combine intense precipitation with rapid melting of late-lying snowpack. Three sources of uncertainty, other than the hydrologic model processes and parameters, are considered in the revised manuscript: (i) the resolution of the atmospheric forcings, (ii) the snow and soil moisture initial conditions and (iii) the representation of the soil texture. For this purpose, an ensemble of twelve hydrological simulations with GEM-Hydro is used in the revised manuscript:

- three atmospheric forcings are used at 10, 2.5 and 1 km as in the initial manuscript. At each resolution, the precipitation forcing consists of the CaPA analysis including all the stations available in the region to obtain the best estimate of precipitation at a given resolution.
- for each atmospheric forcing, the same two initial snow and soil conditions are used as in the initial manuscript: the default GEM-Hydro simulation and snow information from the SNOw Data Assimilation System (SNODAS) that were used to modify peak snow accumulation in the GEM-Hydro simulation.
- finally for each atmospheric forcing and initial snow/soil conditions, two different soil database are considered: the Global Soil Dataset for Earth System Model (GSDE, (Shangguan et al., 2014) and the SoilGrid dataset (Hengl et al., 2017). These 2 database can be considered as reference soil databases for Earth system models (Dai et al, 2019).

These hydrological simulations were evaluated at a new set of 12 unregulated stations located in the headwaters of the Red Deer, Bow and Oldman River basins (see below our answer to the next general comment regarding the statistic). This new set of the stations allowed use to propose a detailed analysis of the sensitivity of the hydrological response to the three sources of uncertainty considered in this study. Their relative importance was compared for the different hydrometric stations, representing a large variety of headwater catchments. To our knowledge, this is the first time these three source of uncertainties are considered simultaneously for a ROS event in several mountain basins.

Major changes have been made in the text and in the figures compared to the revised manuscript. The results section has been extensively re-written:

- Section 3.1 evaluates the different precipitation forcings and a new Table (Table 5) contains detailed performance metrics. A new figure (Fig. 5) details the average accumulated precipitation over the 12 basins of interest.
- Section 3.2 quantifies the impact of the insertion of SNODAS data on the snowpack and soil moisture initial conditions prior to the flooding event. A new figure (Fig 8) gives the initial SWE and soil saturation for the different set of initial conditions considered in this study.
- Section 3.3 contains the evaluation of the hydrological simulations and presents the sensitivity of the hydrological response to the different source of uncertainties. It has been total re-written compared to the initial manuscript and new figures are used:
  o Figure 9 presents the distribution of error metrics for the different hydrological simulations considered in this study. Then,
  o Figure 10 quantifies the sensitivity of the error metrics to the 3 sources of uncertainty considered in this study
  o Figures. 11 and 12 provide example of hydrographs at different hydrometric stations to illustrate the variability of the sensibility of the hydrological responses across the different headwater basins considered in this work.
  o Figure 13 presents an analysis of the surface energy balance of the snowpack simulated by GEM-Hydro and its sensitivity to the resolution of the atmospheric forcing.

Based on these new results, the discussion section and the conclusions were modified accordingly.

Where is the solid statistic?

An improved experimental design is now used in the revised manuscript. 12 unregulated stations located in the headwaters of the Red Deer (2 stations), Bow (6 stations) and Oldman (4 stations) rivers basins were selected for the evaluation of hydrological simulations. Among these 12 stations, 4 stations were used in the initial manuscript. The location of these stations is given on the map below. In total, the river basins of these 12 stations cover 12700 km$^2$ and were characterised by different total precipitation amount, initial snowpack conditions and soil characteristics. For each of these stations, hydrological simulations were

evaluated using three error metrics: the difference of amplitude of the peak flow (%), the percent bias (%) and the Nash-Sutcliffe efficiency (-). The distribution of the error metrics for each configuration of GEM is shown on Fig. 9 of the revised manuscript and is discussed in the text (Sect. 3.3 of the revised manuscript, P 13 L 28 to P 14 L 10). The sensitivity of the error metrics to the three sources of uncertainty considered in this study is then detailed in Fig. 10 of the revised manuscript. In addition, the scores for each hydrometric station and each hydrological simulation are given in Fig. S3 of the supplementary material.

[Figure]

*Figure 1. Location of the 12 hydrometric stations used for the evaluation of hydrological simulations in the revised manuscript. The black lines represent the limits of the basins associated with each station. The orange polygon delineates the location of the basins that were used for model evaluation in the initial manuscript.*

**Comments from the pdf**

**Main points:**
- Your study is a case study. Fits in my opinion better in NHESS than in HESS. I am personally not very interested in such studies, where actually only the data owner and model operators learn something about their specific data chain in a single event.

See above our answer to this comment.

- Where is the true novelty? You use well citable tools and data for a specific event. Their present combination might be novel, but surely not original (different resolution, different initial conditions, different density of stations).
See above our answer to this comment.

-Figures 10-12 suggest a range based on only two member. This is not adequate.
We thank Reviewer 2 for this comment. The filled area was systematically removed from the figures showing hydrographs in the revised manuscript to avoid any mis-interpretation of the results. See Figures 11 and 12 of the revised manuscript.

**Introduction**
The whole introduction reviews previous analysis of the here analyzed event and introduces the specific models and data used by the authors.
Please correct me, but I don't see any link to current challenges in hydrological modelling of rain-on-snow events, current methods in simulation of floods and previous similar approaches in other areas.
What is the knowledge gap here?
The fact that "Despite its severe hydrological consequences, this extreme weather event has received little attention from a hydrological modelling point of view."?
Is the question "cam my model cope with this event" still a relevant one in the times where model assessment relies on multi-model and multi ensemble input for periods of several years?

The Introduction in the revised manuscript has been fully rewritten to put the study in a general context and to make it less localized. It is organised as follows:
- The 1st paragraph details the meteorological and hydrological conditions during the June 2013 flood in Alberta and details previous works on this extreme event.
- The 2nd paragraph mentions similarities between this event and several past flooding events in mountain basins and identifies the specificity of late-spring flooding events that can combine intense precipitation with rapid melting of late-lying snowpack.
- The 3rd paragraph gives an overview of the main source of uncertainties for operational flood forecasting in complex terrain and highlight the importance of the spatial and temporal accuracy of the rainfall forcing.
- The 4th paragraph details the uncertainties associated with initial soil moisture and the impact of the representation of the soil texture. It also mentions the influence of snowpack initial conditions and the benefit for runoff prediction of using snow information from dedicated external snow monitoring systems.
- The 5th paragraph describes the specific challenges associated with modelling of rain-on-snow event and refers to past studies focusing on hydrological prediction of specific rain-on-snow events.
- The 6th paragraph defines the scientific objective of this study and gives overview of the modelling strategy.

With this new structure, numerous new references have been added to the revised introduction.

P 2 L 27: Pagano, T. C., Wood, A. W., Ramos, M. H., Cloke, H. L., Pappenberger, F., Clark, M. P., ... & Verkade, J. S. (2014). Challenges of operational river forecasting. Journal of Hydrometeorology, 15(4), 1692-1707.
This reference is mentioned in the introduction of the revised manuscript. (P 2 L 32).

**Text**

P4. L4L Are these data available only for the event or also for some training periods before?

Measurements of precipitations and snow water equivalent are available both for this event and for other time periods before and after the flooding event. Daily discharge data are available through the National Hydrological Service website and cover the periods of operation of the different hydrometric stations. Finally, hourly discharge data for the flooding event were obtained through a specific request to the National Hydrological Service. Mostly hourly discharge data were used in our study since we focused on the short-term hydrological response during the flooding event. Daily discharge data were used to produce Fig. S4 in the supplementary material. The description of the data availability is now improved in the revised manuscript (P20 L 30 to P 21 L 4)

P5 L1-2 Large literature on this: E.g.
Jonas, T., Marty, C., & Magnusson, J. (2009). Estimating the snow water equivalent from snow depth measurements in the Swiss Alps. Journal of Hydrology, 378, 161-167. https://doi.org/10.1016/j.jhydrol.2009.09.021

We fully agree with Reviewer 2 that methods such as the one proposed by Jonas et al. (2009) could be used to obtain an estimation of SWE from the ECCC snow depth analysis, even if this method was initially developed in the Swiss Alps and may present limitations in the Canadian Rockies. The low resolution of the ECCC operational snow analysis (10 km) at the time of the event was the main reason why we did not use this product in our study. Therefore, we removed from the revised manuscript the sentence: "*Moreover, the ECCC snow analysis only contains information on snow depth and therefore lacks the crucial information that SWE is for hydrology*".

P6 L15-16 These are operational runs or ad-hoc hindcasts?

All the GEM simulations used to drive the hydrological simulations are ad-hoc hindcasts that were specifically generated for this study using the operational versions of the GEM atmospheric model available in June 2018. We had better insist on this point in the revised manuscript (P 7 L 16-21).

P 7 L 7-9 Large set on international literature on this.
E.g.: Germann, U., Galli, G., Boscacci, M., & Bolliger, M. (2006). Radar precipitation measurement in a mountainous region. Quarterly Journal of the Royal Meteorological Society: A journal of the atmospheric sciences, applied meteorology and physical oceanography, 132(618), 1669-1692.

The reference provided in the initial manuscript was published in an international journal and was specific to the June 2013 flooding event. As suggested by Reviewer 2, we added the reference to Germann et al. (2006) in the revised manuscript (P 8 L 30).

P 8 L1 I accept your argumentations. This is for me a reason to ask nevertheless for simulation of other (smaller) events in order to evaluate the non-randomness of your analyses.

We did not add the simulations of additional events in the revised manuscript. Indeed, as mentioned before, this work has required the generation of new GEM hindcasts of the June 2013 Flood at different resolutions. This cannot be easily extended to other events. Instead, we add new hydrological simulations of the same event using a different soil texture database (see Table 1 and 4 in the revised manuscript). We also used a new set of 12 hydrometric stations for model evaluation (see our answer above to the second general comment). This new set of stations consist of the main unregulated stations located in the headwaters of the Oldman, Bow and Red Deer Rivers basins. This new selection of stations allowed us to derive less localised conclusion and improve the quality of our analysis regarding the sensitivity of the hydrological response across different mountainous basins.

P 8 L 23: Sevruk, B. (1983). Correction of measured precipitation in the Alps using the water equivalent of new snow. Hydrology Research, 14(2), 49-58.
Nešpor, V., & Sevruk, B. (1999). Estimation of wind-induced error of rainfall gauge measurements using a numerical simulation. Journal of Atmospheric and Oceanic Technology, 16(4), 450-464.

Savina, M., Schäppi, B., Molnar, P., Burlando, P., & Sevruk, B. (2012). Comparison of a tipping-bucket and electronic weighing precipitation gage for snowfall. Atmospheric Research, 103, 45-51.
The reference to Nespor and Sevruk (1999) has been included in the revised manuscript (P 10 L 8).

P 8 L 28-29
This study also uses alternatively row model conditions and SWE estimations at model initialization in the context of runoff and SWE forecasting in mountainous terrain. SWE estimation based on observations improve the forecast of discharge volume
Jörg-Hess, S., Griessinger, N., & Zappa, M. (2015). Probabilistic forecasts of snow water equivalent and runoff in mountainous areas. Journal of Hydrometeorology, 16(5), 2169-2186. https://doi.org/10.1175/JHM-D-14-0193.1
P9 L3: Figure 10 in  Jörg-Hess, S., Griessinger, N., & Zappa, M. (2015). Probabilistic forecasts of snow water equivalent and runoff in mountainous areas. Journal of Hydrometeorology, 16(5), 2169-2186. https://doi.org/10.1175/JHM-D-14-0193.1
We thank Reviewer 2 for this very interesting reference. This reference has been added in the introduction (P 3 L 24) and in the discussion (P 17 L 32),

P 10 L3-5 : Is such behavior to be expected also for less severe events?
An overall improvement of quantitive precipitation estimation with atmospheric models running at convective-permitting resolution compared to lower resolution models has been obtained in many studies in complex terrain (e.g. Rasmussen et al., 2011; Lundquist et al., 2019). These studies did not only focused on intense precipitation events. Therefore, we estimate that similar conclusions are expected for less severe events as well. These references are mentioned in the introduction and the discussion of the revised manuscript.
- Lundquist, J., Hughes, M., Gutmann, E., & Kapnick, S. (2019). Our skill in modeling mountain rain and snow is bypassing the skill of our observational networks. Bulletin of the American Meteorological Society, (2019).
- Rasmussen, R., Liu, C., Ikeda, K., Gochis, D., Yates, D., Chen, F., et al. (2011). High-resolution coupled climate runoff simulations of seasonal snowfall over Colorado: a process study of current and warmer climate. J. Clim. 24, 3015– 3048. doi: 10.1175/2010JCLI3985.1

P 10 L 6 Red Deer seem to be also fine at 10 km standard application. Why?
*CaPA 10 km Def* showed improved performances for the Red Deer River basin compared to the Bow and Oldman rivers basins as shown on Fig. 7a of the initial manuscript. However, a more detailed look at Fig. 5 of the initial manuscript shows that *CaPA 10 km Def* presented an underestimation of precipitation on the eastern side of the headwaters of the Red Deer River and an overestimation in the western side of the headwaters. This resulted from an error in the location of the precipitation in the 10-km background that was not corrected by the analysis since no station was used in the headwaters of the Red Deer River by *CaPA 10 km Def* (see Fig. 1 of the initial manuscript). The additional stations used in *CaPA 10 km New* corrected this error in the location of the precipitations (Fig. 6d of the initial manuscript), leading to improved performances of the analysis for the Red Deer Basin (Fig. 7b of the initial manuscript).
*CaPA 10 km Def* was removed from the revised manuscript so that its performances are not discussed anymore in the revised manuscript. However, it should be noted that *CaPA 10 km New* provided the best estimates of precipitation in terms of RMSE and correlation coefficient for the Red Deer basin compared to *CaPA 2.5 km* and *CaPA 1 km* (Table 5 of the revised manuscript). This results from better performance during the startiform phase of the heavy precipitation event.

P 10 L 17-18: Citation?
We added the following reference to the revised manuscript:
- Singh, V. P. (1997). Effect of spatial and temporal variability in rainfall and watershed characteristics on stream flow hydrograph. Hydrol. Proc. 11(12), 1649-1669

P 11 L 7 : Nice Figure, but difficult to see the black lines from the barcharts to the gauges.

Thanks for this comment. This figure has been removed from the revised manuscript due to the larger number of stations used for the evaluation of hydrological simulations. It has been replaced by Fig. 9. The location of the hydrometric stations is shown on Fig. 9.

P 11 L 9: So, 10km_Def always profits from SND instead of OPT. The other forcings partly.

The atmospheric forcing *10km_Def* used in the initial manuscript suffered from a systematic underestimation of precipitation which was systematically and partially compensated by the increase in initial SWE when using SND as the initial snow condition. The other precipitation forcing benefited from additional stations which greatly improved their quality in the headwaters, removing the underestimation of precipitation and removing the main source of uncertainty of the hydrological simulations. Therefore, hydrological simulations using these improved atmospheric forcing were more sensitive to errors in the initial snow conditions; in particular in the headwaters where SND tends to overestimate the initial SWE conditions and led to overestimated flood volume (see station 05BL019 for example in the initial manuscript).
The atmospheric forcing *10km_Def* is not used anymore in the revised manuscript. The sensitivity to the initial conditions taken from configurations SND and OPL is now presented in Sect. 3.3 (P 14 L 11 to P 15 L 13) and shown on Fig. 10 for the 12 headwater basins considered in the revised manuscript.

P 11 L 23: Same for at least two other creeks

These two creeks corresponded to the hydrometric stations 05BL027 and 05BL023 that were used in the initial manuscript. No snow in configuration OPL and SND was present in the drainage basins of these 2 stations on 18 June 2013. Note that these two stations were removed from the analysis in the revised manuscript.

P12 L 1-2: I don't see the point in filling the areas between OPT and SND in panels c) and f) of both Figures. You suggest here a range like for probabilistic forecasts with only two members.
Same in Figure 12.

See above our answer to the 3^rd general comment.

P 12 L 26-27 : We see the results, but we don't see the numbers for all Manning options. Table would be useful

We thank Reviewer 2 for this comment. In the revised manuscript, the results obtained with the different Manning options were removed from the section Results since they correspond to expected results with a lack of real novelty as detailed by Reviewer 3. Therefore, we decided to present the adjustment of Manning parameters as a calibration step in the revised manuscript. It is now mentioned in the section Methods (P 9 L 11-14) and describes in details in the supplementary material of the revised manuscript. In particular, Figure S2 in the supplementary materials shows the impact of adjusting the Manning's parameters on simulated peak flow timing and amplitude.

P 14 L 32: Is this really the first citation to a international study?

The paper by Bernhardt and Schulz (2010) describes the SnowSlide scheme which can be considered in the snow hydrology community as one of the reference schemes that simulate gravitational snow redistribution in a hydrological model. However, it is clear that it is not the first time that the importance of gravitational snow redistribution for mountain snow hydrology is mentioned in a paper. We can mention for example a study by Blösch and Kirnabauer (1992):

Blösch, G. and Kirnbauer, R. (1992), An analysis of snow cover patterns in a small alpine catchment. Hydrol. Process., 6: 99-109. doi:10.1002/hyp.3360060109

Note that the reference to the paper by Bernhardt and Schulz (2010) has been removed from the revised manuscript to limit the number of references used in this paper.

**Answer to Reviewer 3 HESS-2019-152**

We thank Reviewer 3 for his insightful comments. We provide here our responses to his comments and describe how we address his comments in the revised manuscript. The original review comments are in normal black font while our answers appear in blue font. The lines numbers given below refer to the line numbers in the version of revised manuscript without track-change mode.

**General Comments**

This paper examined the capability of a hydrological model (GEM-Hydro) in simulating the June 2013 flood event in Alberta, Canada. In particular, three sub-basins in the Bow River basin were selected to assess the impacts of spatial resolution, precipitation gauge density, and initial snow conditions on model ability in reproducing the flow volumes. Also, the model sensitivity to Manning coefficients in capturing the peak flow was investigated.

General Comments:
The objective of this paper is straightforward and this paper is well-written, easy to follow and well-structured. However, the creditability of the study has been reduced because the study is highly localized and reads more like a report of an application of a hydrological model to a specific flood event. There are little knowledge gain for the community. In general, two major concerns are needed to be addressed in this paper:

1) Novelty of the study

While the study would be a great contribution to the development of a Canadian hydrological forecasting system, it might not be novel enough as a scientific contribution for the international community. The effects of different spatial resolutions on model simulations have been previously and heavily studied in different hydrological models and it is well expected that finer resolution could provide better simulations because of its ability in capturing the fine-scale hydro-meteorological processes. It is also expected that the inclusion of additional information (e.g. increasing network density, inclusion of SWE information) would improve model performance because of the data-driven nature of the sophisticated models nowadays. Lastly, the Manning coefficients (both channel and floodplain) are well known to be one of the most sensitive routing parameters in any hydrological/hydraulic models. Adjusting such parameter will definitely improve the model ability in matching the peak flow timing and magnitude.

Furthermore, the Discussion section of the study did not provide innovative insights on modelling of extreme flood events. Although the Discussion section was well-written and fully supported by references, the major findings of this study were merely a confirmation of what had been shown in many previous studies (as repeatedly mentioned by the authors). Therefore, there is a lack of true novelty and scientific contribution in this study. The authors should vigorously address this critical issue by providing a better discussion on what new knowledge and information the international community could learn from this study.

We thank Reviewer 3 for this general comment. We made significant changes in the revised manuscript to make sure that this study brings enough novel contributions to the international community. These changes are detailed below.

First, some of the results that were presented in the initial manuscript were removed in the revised manuscript since they correspond to expected results with a lack of real novelty as detailed above by Reviewer 3. Therefore, we removed from the manuscript:

- the precipitation analysis at 10 km using the default stations (*CaPA 10 km Def*). This precipitation analysis used the same stations as the operational analysis at the time of the event but its quality is mediocre due the low density of the stations in complex terrain. In the revised manuscript, we only kept the precipitation analysis generated with all the station data for each horizontal resolution (10, 2.5, 1 km). Theses different precipitation data are now used in the revised manuscript as our best estimate of the precipitation at each horizontal resolution.
- the results corresponding to the different Manning options that were described in the section Results. The selection of the correction factor applied to the default values of the Manning coefficients is now detailed in the supplementary material and mentioned in the section Methods (P 9 L 11-14). Indeed, as mentioned by Reviewers 2 and 3, the adjustment of the Manning coefficient does not bring novelty to our study. Therefore, we decided to present it as a calibration step. The correction factor selected at this calibration step was then used in the rest of the analysis. Figure 2 in the supplementary material describe the impact of the correction factor. The challenges associated with the choice of the routing parameter for such extreme event are still mention in the section Discussion (P 18 L 28-31)

Removing these two components of the initial manuscript allowed us to focus the revised manuscript on the factors governing the predictions skill of late-spring flood in mountain basins. A new title for the revised manuscript is proposed ***Assessing the factors governing the ability to predict late-spring flooding in cold regions mountain basins.***

Late-spring flood in mountain basin represents a challenge for hydrological forecasting systems since they can combine intense precipitation with rapid melting of late-lying snowpack. Three sources of uncertainty, other than the hydrologic model processes and parameters, are considered in the revised manuscript: (i) the resolution of the atmospheric forcings, (ii) the snow and soil moisture initial conditions and (iii) the representation of the soil texture. For this purpose, an ensemble of twelve hydrological simulations with GEM-Hydro is used in the revised manuscript:

- three atmospheric forcings are used at 10, 2.5 and 1 km as in the initial manuscript. At each resolution, the precipitation forcing consists of the CaPA analysis including all the stations available in the region to obtain the best estimate of precipitation at a given resolution.
- for each atmospheric forcing, the same two initial snow and soil conditions are used as in the initial manuscript: the default GEM-Hydro simulation and snow information from the SNOw Data Assimilation System (SNODAS) that were used to modify peak snow accumulation in the GEM-Hydro simulation.
- finally for each atmospheric forcing and initial snow/soil conditions, two different soil database are considered: the Global Soil Dataset for Earth System Model (GSDE, (Shangguan et al., 2014) and the SoilGrid dataset (Hengl et al., 2017). These 2 database can be considered as reference soil databases for Earth system models (Dai et al, 2019).

As suggested by Reviewer 3 in a specific comment below, the evaluation of the hydrological simulations in the revised manuscript is carried out at 12 unregulated stations located in the headwaters of the Red Deer (2 stations), Bow (6 stations) and Oldman (4 stations) rivers basins. Among these 12 stations, 4 stations were used in the initial manuscript. The location of these stations is given on Figure 1 below and their main characteristics are reported in Table 1 below. In total, the river basins of these 12 stations cover 12700 km² and are characterised by different soil characteristics, total precipitation amount during the flooding event and initial snowpack conditions prior to the flood. This new set of the stations allowed use to propose a detailed analysis of the sensitivity of the hydrological response to the three sources of uncertainty considered in this study and to compare their importance for hydrometric stations, representing a large

variety of headwater catchments. To our knowledge, this is the first these three source of uncertainties are considered simultaneously for a ROS event across several mountain basins.

Major changes have been made in the text and in the figures compared to the revised manuscript. The results section has been extensively re-written:

- Section 3.1 evaluates the different precipitation forcings and a new Table (Table 5) contains detailed performance metrics. A new figure (Fig. 5) details the average accumulated precipitation over the 12 basins of interest.
- Section 3.2 quantifies the impact of the insertion of SNODAS data on the snowpack and soil moisture initial conditions prior to the flooding event. A new figure (Fig 8) gives the initial SWE and soil saturation for the different set of initial conditions considered in this study.
- Section 3.3 contains the evaluation of the hydrological simulations and presents the sensitivity of the hydrological response to the different source of uncertainties. It has been total re-written compared to the initial manuscript and new figures are used:
  - Figure 9 presents the distribution of error metrics for the different hydrological simulations considered in this study. Then,
  - Figure 10 quantifies the sensitivity of the error metrics to the 3 sources of uncertainty considered in this study
  - Figures. 11 and 12 provide example of hydrographs at different hydrometric stations to illustrate the variability of the sensibility of the hydrological responses across the different headwater basins considered in this work.
  - Figure 13 presents an analysis of the surface energy balance of the snowpack simulated by GEM-Hydro and its sensitivity to the resolution of the atmospheric forcing.

Based on these new results, the discussion section and the conclusions were modified accordingly.

2) Lack of connection with other worldwide extreme flood events
The Introduction section was again well-written and the rationale of the study was well presented, however, it was highly focused on the description of the June 2013 flood event and the previous works that were related to the 2013 flooding. The literature review did not discuss any research related to other worldwide flood events and any modelling works that address the current challenges of the modelling community in dealing with extreme flood induced by rain-on-snow events. This makes the study highly localized and event specific. The authors should provide a boarder discussion on similar research conducted in other regions, modelling strategy used in simulating such kind of flood events, and the research gaps this study could fill in for advancing the knowledge of the community.

The Introduction in the revised manuscript has been fully rewritten to put the study in a general context and to make it less localized. It is organised as follows:

- The 1st paragraph details the meteorological and hydrological conditions during the June 2013 flood in Alberta and details previous work on this extreme event.
- The 2nd paragraph mentions similarities between this event and several past flooding events in mountain basins and identifies the specificity of late-spring flooding events that can combine intense precipitation with rapid melting of late-lying snowpack.
- The 3rd paragraph gives an overview of the main source of uncertainties for operational flood forecasting in complex terrain and highlight the importance of the spatial and temporal accuracy of the rainfall forcing.
- The 4th paragraph details the uncertainties associated with initial soil moisture and the impact of the representation of the soil texture. It also mentions the influence of snowpack initial conditions and the benefit for runoff prediction of using snow information from a dedicated external snow monitoring system.

- The 5th paragraph describes the specific challenges associated with modelling of rain-on-snow event and refers to past studies focusing on hydrological prediction of specific rain-on-snow events.
- The 6th paragraph defines the scientific objective of this study and gives overview of the modelling strategy.

With this new structure, numerous new references have been added to the revised introduction to make it more general.

Specific Comments:
P1L28-29: How could the results of this study guide the development of the hydrological forecasting system worldwide?
This sentence has been removed from the Abstract in the revised manuscript. Following the major changes made in the paper, the abstract has been extensively rewritten.

P4L10: Could the authors comment on the consistency of these four different networks?
These four different meteorological networks have been developed for different purposes by different government agencies or universities. For example, the SYNOP and ABE stations are well distributed across Alberta to provide a permanent monitoring of weather conditions across the province. On the other hand, CRHO stations are distributed around targeted mountainous catchments and provide very valuable information in poorly observed areas by the other networks. CaPA has been designed to deal with different precipitation networks and insure the consistency of the precipitation data when they are included in the analysis. A spatial consistency test is applied in the Quality-control (QC) procedures of CaPA to identify and remove observations with large errors from the final analysis. The QC procedures are detailed in Lespinas et al. (2015). The revised manuscript mentions the QC procedures used in CaPA. (P 8 L 26-28).

P5L3-4: What are the drainage areas of these river basins?
The answer to this question is given in Table 1 shown below and in Table 1 in the revised manuscript.

P5L6-7: Could the authors provide the basic information of these 10 stations (e.g. station name, drainage area, name of tributary)?
As mentioned above in our answer to the first general comments and as proposed by Reviewer 3 in one of his following comment, a new set of 12 stations is used for model evaluation in the revised manuscript. This new set of stations consists of the main unregulated stations located in the headwaters of the Oldman River, Bow River and Red Deer River basins. Table 1 in the revised manuscript and below provides the main information for the new hydrometric stations used for model evaluation whereas Fig. 2 in the revised paper shows the locations of these stations.

*Table 1. Characteristics of the hydrometric stations used for model evaluation in the revised manuscript*

| Station Code | Station Name | Drainage Area (km²) | Included in the initial paper |
|---|---|---|---|
| 05CA004 | Red Deer River Above Panther River | 941 | No |
| 05CA009 | Red Deer River Below Burnt Timber Creek | 2246 | No |
| 05BB001 | Bow River at Banff | 2210 | No |
| 05BG010 | Ghost River Above Waiparous Creek | 484 | No |

| Station Code | Station Name | Drainage Area (km²) | Included in the initial paper |
|---|---|---|---|
| 05BH015 | Jumpingpound Creek at Township Road | 474 | Yes |
| 05BJ010 | Elbow River at Sarcee Bridge | 1189 | Yes |
| 05BL012 | Sheep River at Okotoks | 1494 | Yes |
| 05BL019 | Highwood River at Diebel's Ranch | 774 | Yes |
| 05AB041 | Willow Creek at Oxly Ranch | 833 | No |
| 05AA035 | Oldman River at Range Road | 1835 | No |
| 05AA008 | Crowsnest River near Franck | 403 | No |
| 05AA022 | Castle River near Beavers Mines | 820 | No |

P10L10-12: It would be better to provide the performance measures (Bias, RMSE, R2) of each basin in a table.
Table 5 in the revised manuscript provides the performances measures for each CaPA experiment and each main basin. Performances metrics were computed for the whole event and separately for the convective and stratiform periods as defined in Kochtubajda et al. (2016) and Li et al. (2017). The performances of the QPE products for these different periods and regions are now presented in the revised manuscript (P 11 L 18 to P 12 L 4). To limit the increase in the length of the manuscript, Table 5 in the revised version replaced the scatter plots that were shown on Figure 7 in the initial manuscript.

P10L13-14: Could the authors explain how the negative bias was removed by increasing the spatial resolution to 2.5km and 1km? Figures 7c) and 7d) do not show an obvious removal of the negative bias, instead, the points are still scattered above and below the 1:1 line.
Improvements in the precipitation analysis from 10 km to 2.5 km and 1 km resulted from improvements in the precipitation background provided by the Canadian NWP system GEM. GEM at 2.5 km and 1 km resolves explicitly part of the convection with the cloud microphysical scheme P3 whereas all the convection is parameterised in GEM at 10 km. Previous studies by Li et al. (2017) and Milrad et al. (2017) showed that atmospheric models at convection-permitting resolution performed best for this event. As detailed by Milrad et al. (2017), this is mainly due to (i) an improved representation of the orographic ascent that contributed to the magnitude of the extreme rainfall and (ii) an anchoring and increasing duration of the precipitation on the Eastern side of the Rockies. These explanations are clearly given in the revised manuscript. (P 11 L 30 to P 12 L1).

Compared to the analysis at 10 km, the bias was improved at 2.5 and 1 km since the points were better centred around the 1:1 line on Fig. 7 in the initial manuscript. However, as mentioned by Reviewer 3, these points were still scattered around this line with led to similar RMSE for the different experiments (see the last row of Table 5). Since Figure 7 has been removed from the revised manuscript, we did not describe anymore the scatter of the points for the different experiments. Instead, we estimate that Table 5 in the revised manuscript provides a better overview of model performances.

P10L15-16: Could the authors explain why there were overestimations of cumulative precipitation using CaPA 1.0km?

GEM at 1 km explicitly simulates most of the convection with its cloud microphysical scheme and is able to simulate localized high-precipitation structures that developed during this extreme rainfall event. However, the exact location of these precipitation cells was not necessarily well captured by the atmospheric model. This issue is frequently encountered when evaluating precipitation forecasts from convection-permitting models against rain-gauge observations (e.g. Clark et al. 2016). In terms of precipitation analysis, the error in terms of location of the precipitation patterns may not systematically corrected by CaPA during the leave-one-out evaluation. This limitation is mentioned in the revised manuscript (P 12 L 1-4).

*Clark, P., Roberts, N., Lean, H., Ballard, S. P., & Charlton-Perez, C. (2016). Convection-permitting models: a step-change in rainfall forecasting. Meteorological Applications, 23(2), 165-181.*

P11L1-2: While the analysis on the effects of spatial resolution, gauge network density, and initial snow condition were conducted across the Oldman, Bow, and Red Deer River basins, the hydrological simulations were evaluated on Jumpingpound Creek, Elbow River and Highwood River, which are all located within the Bow River basin. I think such selection of hydrological simulations could not fully reveal the impacts of those factors controlling the flood dynamics. A better experimental design could be selecting one or two headwater sub-basins from each river basin (Oldman, Bow, and Red Deer). The results could potentially provide more information than the current setting (10 stations all within the Bow River basin) especially when different responses were witnessed after the inclusion of additional information (precipitation and/or SWE) across the three basins (e.g. consistently underestimation of cumulative precipitation plus overestimation of SWE in the Bow River, a mixture of over-and under-estimation of cumulative precipitation in the Red Deer, and underestimation of cumulative precipitation plus fairly accurate SWE estimates in the Oldman River). I wonder why the authors only focused on hydrological simulations within the Bow River basin.

Only hydrometric stations within the Bow Rover basin were selected in the initial manuscript since we focused our evaluation of hydrological simulations over one of the area that was the most severely impacted during the flooding event and that was monitored by a relatively dense network of hydrometric stations. We agree with Reviewer 3 that this selection of stations limited our ability to analyse the impact of the different factors governing the flood dynamics.

Following this comment, a better experimental design is now used in the revised manuscript. 12 unregulated stations located in the headwaters of the Red Deer (2 stations), Bow (6 stations) and Oldman (4 stations) River basins were selected for the evaluation of hydrological simulations. Among these 12 stations, 4 stations were used in the initial manuscript. The location of these stations is given on the map below and their main characteristics are reported in the table on the previous page. In total, the river basins of these 12

stations cover 12700 km² and were characterised by different total precipitation amount, initial snowpack conditions and soil characteristics.

[Figure]

*Figure 1. Location of the 12 hydrometric stations used for the evaluation of hydrological simulations in the revised manuscript. The black lines represent the limits of the catchments associated with each station. The orange polygon delineates the area that was studied in the*

P15L30-P16L15: This study did not examine the model structure and the process representation in the model at all. Providing a list of potential reasons that might affect the model performance here becomes irrelevant unless concrete proof and result analysis are given to show the underlying causal relationship. This sub-section should be better re-written.
This section has been rewritten and simplified (P 18 L 30 to P 19 L 12). We kept a description of the limitations associated with SVS since it potentially affects the hydrological response simulated by the model and we strongly believe that the reader must be aware of these limits.

P17L19-20: What about the ground station data from the ABE, CHRO, and COOP networks? Are they publicly available to the international readers?
All the data from the ABE, CRHO and COOP networks are available to the international readers through requests on dedicated web-portals. The section *Data availability* of the revised manuscript includes these links (P 20 L 30).

Remarks:

P2L4-5, 10, and 12: could the authors check this reference please? Is this reference the correct one that the authors intended to use? I guess it should be Pomeroy et al. 2016a. Please correct me if I am wrong.

*Indeed we were using the wrong reference for lines 10 and 12 of the original introduction. The correct reference is Pomeroy et al (2016b) and we made the correction in the revised manuscript.*

*Pomeroy, J. W., R. E. Stewart, and P. H. Whitfield. The 2013 flood event in the South Saskatchewan and Elk River basins: Causes, assessment and damages. Can. Water Resour.. J. 41 (1-2), ,https://doi.org/10.1080/07011784.2015.1089190, 2016b*

P8L18: should be "similar to" not "similarly to" Correction included

P8L24 and 27: please spell out the full name first before using the abbreviation
Full names and abbreviations are now used in the right order.

P8L33: missing full stop after "(18 June 2013)" Correction included

P12L5: should it be "Fig. 10c, f and Fig. 11c, f" instead of " Fig. 10d, f and Fig. 11d,f"? This part of the text has been removed.

P12L23: better use "hypothesis" instead of "assumption" This part of the text has been removed.

P17L15: delete "are" This part of the text has been removed.

Figures 3 and 4: better use "boundaries" instead of "limits"
The legend of the two figures were modified according to this comment.

Figures 10-12: it is a bit misleading to use filled area to represent the results because essentially there are only two simulations using initial conditions from OPL and SND. They might not necessarily represent the upper and lower limits of the model performance, especially when more different initial conditions are used. We agree with Reviewer 3. The filled area has been systematically removed from the figures showing hydrographs in the revised manuscript to avoid any mis-interpretation of the results. See Figures 11 and 12 of the revised manuscript.

[revised manuscript text omitted]

Despite its severe hydrological consequences, this extreme weather event has received little attention from a hydrological modelling point of view. Teufel et al. (2017) discussed the impact of initial snow and soil conditions on the amplitude and timing of streamflows. However, their study only used daily discharges from one hydrometric station located in Calgary along the Bow River for model evaluation, and this station was heavily impacted by reservoir regulation making it unsuited for this purpose. Li et al. (2017) and Milrad et al. (2017) did not use their high-resolution WRF simulations to generate hydrological simulations of the flood. At the local scale, Fang and Pomeroy (2016) studied in detail the impact of antecedent soil moisture and snowpack conditions on runoff generation during the flood for a small (9.4 km$^2$) and well-instrumented mountain research basin. Pomeroy et al. (2016a) used the data from the same basin to better understand and simulate the complex rain-on-snowmelt runoff generation during the event. They showed that successful modelling of the event at small scales required detailed consideration of precipitation phase, rain-on-snow processes, slope/aspect impact on snowpack energetics, forest canopy-hydrological process interactions, antecedent soil moisture, sub-surface storage and routing of runoff water through both sub-surface and overland flow pathways. However, the modelling studies mentioned here did not evaluate the potential and limitations of large-scale hydrological models to simulate the dynamics of the flood at [1]

measurements from all observation networks available in the region using the Canadian Precipitation Analysis (CaPA) system (Lespinas et al., 2015; Fortin et al., 2018) to generate reference quantitative precipitation estimation (QPE) products for the flood at different resolutions. Uncertainties related to snowpack and soil moisture ICs were assessed by considering two estimates of snowpack conditions at peak snow accumulation: the default GEM-Hydro simulation and snow information from the SNOw Data Assimilation System (SNODAS; Barett, 2003). They were used in conjunction with two soil texture datasets in GEM-Hydro to generate a total of four different sets of snowpack and soil moisture ICs and their impact on flood modelling. The paper is organized as follows. Section 2.1 presents the study area and the different datasets used in our study. It also describes the GEM-Hydro modelling platform and the configurations of the different experiments carried out with this model. Section 3 evaluates the different hydrological inputs used in this study (initial snowpack conditions, QPE products) and examines the performances and the sensitivity of the resulting hydrological simulations. Section 4 contains a discussion of the main results of this study. Finally, concluding remarks are presented in Section 5.

**2. Data and Methods**

**2.1 Study area and data**

The study area covers the three main river basins in southern Alberta that were strongly impacted by the June 2013 flood: the Red Deer, Bow and Oldman river basins (Fig. 1). These rivers drain the Rocky Mountains and their foothills, flowing eastward towards the Canadian Prairies and eventually joining to form the South Saskatchewan River. Hourly streamflow time series were obtained from the National Hydrological Service of Environment and Climate Change Canada (ECCC) for 12 stations located in the headwaters of the Red Deer River (2 stations), Bow River (6 stations) and Oldman River (4 stations). These rivers were selected for the evaluation of GEM-Hydro simulations since they were strongly impacted by the flood and are not affected by regulation. The location of these stations is shown on Fig. 2 and their main characteristics are given in Table 1. Information on the soil texture (proportion of clay, sand and silt) for each basin were obtained from two different global datasets: the Global Soil Dataset for Earth System Model (GSDE; Shangguang et al. 2014) and the SoilGrid database (Hengl et al., 2017). The GSDE dataset has a resolution of 30 arcsec and combines existing regional and national soil information. The SoilGrid product has a resolution of 250 m and was obtained using digital soil mapping. Both datasets can be considered as reference datasets (Dai et al., 2019). The mean soil texture per basin is reported in Table 1, and Fig. S1 in the supplementary material shows the top 50-cm clay and sand content over southern Alberta for the GSDE and SoilGrid databases. SoilGrid provides grid-based spatially continuous estimation of soil texture, while GSDE provides estimations with abrupt changes at the boundaries of soil polygons. In both datasets, the soil contains more sand than clay for each basin (except for station 05BH015 with GSDE). SoilGrid reports generally more sand and less clay than GSDE, especially for the basins located in the central part of southern Alberta (from station 05BH015 in the north to station 05AA035 in the south; Table 1).

The primary objective of this study is to evaluate the ability of GEM-Hydro to reproduce the observed hydrology in hindcast mode for the June 2013 flood in southern Alberta. Another main objective is to assess the sensitivity of hydrological simulations to the main factors controlling flood dynamics, such as precipitation forcing accuracy, initial snowpack conditions and river routing parameters. For this purpose, a specific configuration of the Canadian NWP model GEM (Côté et al., 1998; Girard et al., 2014) was deployed over the region to produce atmospheric forcing at resolutions ranging from 1 to 10 km to drive GEM-Hydro. In particular, the precipitation simulated by GEM was combined with precipitation gauge measurements from different observation networks using the Canadian Precipitation Analysis (CaPA) system (Mahfouf et al., 2007; Lespinas et al., 2015; Fortin et al., 2018) to generate several quantitative precipitation estimation (QPE) products for the flood. These products were used to assess how QPE accuracy influences the uncertainty in simulating streamflow from the mountain headwaters downriver to cities that were impacted by the flood such as Calgary and High River. The role of initial snowpack conditions at high elevations prior to the flood and uncertainties in river routing parameters were also considered.¶

Precipitation data were obtained from four different networks (Fig. 1). Automatic synoptic stations (SYNOP network) maintained by Environment and Climate Change Canada and other organizations such as Alberta Agriculture and Forestry provide a good coverage over the prairies and the forested foothills but only a sparse coverage of the Canadian Rockies, particularly in Banff National Park where the stations are mainly located in the valleys (see for example the upper Bow river basin, Fig. 1). Therefore, to fill in data gaps, precipitation data were taken from Alberta Environment and Parks because of their good coverage of the higher elevations of the Canadian Rockies headwaters. Data from high elevation mountain stations of the Canadian Rockies Hydrological Observatory (CRHO, University of Saskatchewan, https://www.usask.ca/hydrology/CRHO.php) were also used in the upper Bow River Basin. SYNOP, ABE and CRHO data consist of hourly data that were cumulated to derive 6-h precipitation amounts for CaPA (Sect. 2.3.2). Finally, stations from the American Cooperative Observer Network (COOP) reporting 6-h precipitation amounts were included. These stations are mainly located in the US and on the western side of the study area and were referred to as stations from the SHEF network (Standard Hydrometeorological Exchange Format) in Lespinas et al. (2015). They were not included in the 6-h operational version of CaPA at the time of the flood.

Information on snow conditions before the flooding event and during the winter of 2012/2013 were obtained from 11 automatic snow pillows from Alberta Environment and Parks located in southern Alberta. These stations measure hourly snow water equivalent (SWE). Additionally, outputs from the SNOw Data Assimilation System (SNODAS) from the US National Operational Hydrologic Remote Sensing Center (NOHRSC) were included in this study. SNODAS estimates various snow properties (including SWE and snow depth) by merging satellite, airborne, and ground-based snow data with a numerical simulation of snowcover (Barrett, 2003). SNODAS data are available at 1-km spatial resolution and 24-hour temporal resolution and cover continental US as well as part of Canada (up to 54° N). In southern Alberta, the snow pillows from Alberta Environment and Parks are included in SNODAS since they provide relevant information for mountain snow conditions, including those affecting the Columbia River flowing from Canada to the US. The ECCC snow analysis (Brasnett, 1999) was not considered in our study since its spatial resolution at the time of the event (10 km) was too coarse to accurately represent snow conditions in the complex topography of the Canadian Rockies.

**2.2 The GEM-Hydro modelling platform**

GEM-Hydro is a distributed hydrological modelling platform developed at ECCC for hydrological forecasting and prediction across Canada (Gaborit et al., 2017). It includes two components: (i) the GEM-Surf surface prediction system (Bernier et al. 2011) and (ii) the WATROUTE routing scheme (Kouwen, 2010). A recent experimental GEM-Hydro version also includes the Canadian Land Data Assimilation system based on satellite data (CaLDAS-Sat, Carrera et al. 2019) for real-time forecasting purposes. However, CaLDAS was not used in this study. Instead, all GEM-Hydro simulations were performed in an open-loop manner, without any surface or hydrological data assimilation, which would hinder the effects of the external factors assessed here. GEM-Hydro has been extensively evaluated over the Great Lakes basin (Gaborit et al.,

Hourly streamflow time series were obtained from the Water Survey of Canada for stations located in the Jumpingpound Creek, Elbow River and Highwood River basins (Fig. 2). These rivers were selected for the evaluation of GEM-Hydro simulations since they were strongly impacted by the flood and are only slightly affected by regulation. Two stations were destroyed during the flooding event and removed from the study. Overall, 10 stations were kept for the evaluation of hydrological simulations (Fig. 2).¶

[revised manuscript text omitted]

The introduction of CRHO, COOP and ABE stations in the precipitation analysis (*CaPA 10 km New*) modified the pattern of cumulative precipitation (Fig. 6a and d), even as the precipitation background at 10-km grid spacing remained the same between the two simulations (Table 3). In agreement with Fig. 5b, the new stations led to an increase in the estimation of cumulative precipitation in the upper Elbow and Highwood river basins where local maxima were estimated to have exceeded 250 mm. Larger cumulative precipitation was also found in the Upper Bow River Basin and locally in the Oldman River Basin. The foothills of in the Red Deer River Basin are the only place in southern Alberta where the new analysis led to a decrease in cumulative precipitation. The resolution of the precipitation background also affects the precipitation analysis as shown in Fig. 6. 
[revised manuscript text omitted]
 gauged sites in three river basins, especially for the Elbow River (-20% in experiment *1.0km_OPL*) and the Highwood River (-28% in experiment *1.0km_OPL*). This may be explained by the underestimation of initial SWE in the basin headwaters in the *OPL* simulation as shown in Fig. 4. For this reason, the *SND* simulation, which incorporates SNODAS SWE to estimate peak snow accumulation, was considered as an alternative to obtain initial snow and soil conditions. Figure 9 shows that this substitution has no impact on hydrological simulations for Jumpingpound Creek as no snow was present in this drainage basin on 18 June 2013 in either simulation (Fig. 4). In contrast, the *SND* simulation with SNODAS ... [3]

[revised manuscript text omitted]
. This is in agreement with the overestimation of SWE close to peak accumulation reported using snow pillows observations at high elevations in these basins. Teufel et al. (2017) also found an overestimation of daily simulated streamflow when using initial snow conditions from SNODAS in their hydrological simulations of the June 2013 flood. In a recent study, Lv et al. (in review, WRR) reported a strong overestimation of SNODAS SWE in sub-alpine forest elevations in the Canadian Rockies. They have also found a poor agreement in open alpine environments due to missing wind-induced snow redistribution processes in SNODAS, similarly to the results obtained by Clow et al. (2012) in the Colorado Rocky Mountains. Our results show that there is a strong need for the … [7]

[revised manuscript text omitted]

---

## Referee Report (RR1)

In this work, the authors examine the sensitivity of the simulated hydrological processes to three external factors: atmospheric forcing resolution, snow and soil moisture initial conditions (ICs), and the representation of soil texture. They find the simulated hydrological processes are primarily affected by the resolution of the atmospheric forcing, followed by snow ICs, and then the soil texture. The article is well written, and can provide guidance for distributed hydrological simulation in other (mountainous) river basins. I therefore suggest minor revisions.

Minor comments:
1. Page 4 line 27: Remove the extra period.
2. Page 11 line 6: The minima is obtained? Either singular or plural.
3. Page 13 line 26: Missed period after the bracket.

---

## Author Response (AR2)

Answer to Reviewer 3 HESS-2019-152

We thank Reviewer 3 for his comments. We provide here our responses to his minor comments and describe how we address them in the revised manuscript. The original reviewer comments are in normal black font while our answers appear in blue font.

The authors had shown a tremendous effort in addressing the comments made from the previous round of review. I am satisfied with the revised version. There are several very minor issues to be addressed before publication.

• P5L31: replace the full name by ECCC Correction included

• P12L32: missing full stop between "Fig. 7e and 7f" and "Figure 8" Correction included

• Figure 2: please check the station codes. 05BG019 and 05BH010 do not appear in Table 1 and in the text. Thanks a lot for noticing this mistake. We corrected Figure 2 and replaced 05BG019 by 05BG010 and 05BH010 by 05BH015

Answer to Reviewer 4 HESS-2019-152

We thank Reviewer 4 for his comments. We provide here our responses to his minor comments and describe how we address them in the revised manuscript. The original reviewer comments are in normal black font while our answers appear in blue font.

In this work, the authors examine the sensitivity of the simulated hydrological processes to three external factors: atmospheric forcing resolution, snow and soil moisture initial conditions (ICs), and the representation of soil texture. They find the simulated hydrological processes are primarily affected by the resolution of the atmospheric forcing, followed by snow ICs, and then the soil texture. The article is well written, and can provide guidance for distributed hydrological simulation in other (mountainous) river basins. I therefore suggest minor revisions.

Minor comments:
1. Page 4 line 27: Remove the extra period. Correction included

2. Page 11 line 6: The minima is obtained? Either singular or plural. Correction included. We used "The minima of … are obtained …." in the revised manuscript.

3. Page 13 line 26: Missed period after the bracket. Correction included

[revised manuscript text omitted]